# A repeating fast radio burst associated with a persistent radio source

C.-H. Niu[1,21], K. Aggarwal[2,3,21], D. Li[1,4,11,21✉], X. Zhang[4,5], S. Chatterjee[6], C.-W. Tsai[1], W. Yu[5✉], C. J. Law[7,8✉], S. Burke-Spolaor[2,3,9], J. M. Cordes[6], Y.-K. Zhang[1,4], S. K. Ocker[6], J.-M. Yao[10], P. Wang[1], Y. Feng[1,4,11], Y. Niino[12,13], C. Bochenek[7], M. Cruces[14], L. Connor[7], J.-A. Jiang[15], S. Dai[16,17], R. Luo[16], G.-D. Li[1,4], C.-C. Miao[1,4], J.-R. Niu[1,4], R. Anna-Thomas[2,3], J. Sydnor[2,3], D. Stern[18], W.-Y. Wang[1,19], M. Yuan[1,4], Y.-L. Yue[1], D.-J. Zhou[1,4], Z. Yan[5], W.-W. Zhu[1] & B. Zhang[20]

The dispersive sweep of fast radio bursts (FRBs) has been used to probe the ionized baryon content of the intergalactic medium[1], which is assumed to dominate the total extragalactic dispersion. Although the host-galaxy contributions to the dispersion measure appear to be small for most FRBs[2], in at least one case there is evidence for an extreme magneto-ionic local environment[3,4] and a compact persistent radio source[5]. Here we report the detection and localization of the repeating FRB 20190520B, which is co-located with a compact, persistent radio source and associated with a dwarf host galaxy of high specific-star-formation rate at a redshift of 0.241 ± 0.001. The estimated host-galaxy dispersion measure of approximately $903^{+72}_{-111}$ parsecs per cubic centimetre, which is nearly an order of magnitude higher than the average of FRB host galaxies[2,6], far exceeds the dispersion-measure contribution of the intergalactic medium. Caution is thus warranted in inferring redshifts for FRBs without accurate host-galaxy identifications.

Fast radio burst (FRB) 20190520B was detected with the Five-hundred-meter Aperture Spherical radio Telescope (FAST)[7] in drift-scan mode as part of the Commensal Radio Astronomy FAST Survey (CRAFTS)[8] at 1.05–1.45 GHz in 2019. Four bursts were detected during the initial 24-s scan. Monthly follow-up tracking observations between April 2020 and September 2020 detected 75 bursts in 18.5 h with a mean pulse dispersion measure (DM) of 1,204.7 ± 4.0 pc cm⁻³. Assuming a Weibull distribution of the burst waiting time, we model the FRB burst rate to be $R = 4.5^{+1.9}_{-1.5}$ h⁻¹, for a fluence lower limit of 9 mJy ms and a burst width of 1 ms, which indicates that the FRB can episodically have a high burst rate. Similar to other repeating FRBs, this FRB shows a complex frequency–time intensity structure[9] with multicomponent profiles, sub-burst drifting and scattering (Fig. 1, Methods). No linear polarization was detected for FRB 20190520B from the FAST observations (see Methods), and the rotation measure (RM) for this source has been detected from the higher-frequency band[10,11]. The properties of FRB 20190520B are shown in Table 1.

We localized FRB 20190520B with the Karl G. Jansky Very Large Array (VLA) using the 'realfast' fast transient detection system[12]. Throughout the second half of 2020, we observed the source for 16 h and detected 3, 5 and 1 bursts in bands centred at 1.5 GHz, 3 GHz and 5.5 GHz, respectively. We measured a burst-source position in the International Celestial Reference Frame (ICRF) of (right ascension (RA), declination (dec.)) (J2000) = (16 h 02 min 04.272 s, −11° 17′ 17.32″) with a positional uncertainty (1σ) of (0.10″, 0.08″) dominated by systematic effects (Methods). Deep images using data from the same observing campaign revealed a persistent radio continuum counterpart at (RA, dec.) (J2000) = (16 h 02 min 04.261 s, −11° 17′ 17.35″) with 1σ position uncertainty of (0.10″, 0.05″) that is compact (less than 0.36″; Methods) and has a flux density of 202 ± 8 μJy averaged over a span of about 2 months from 30 August to 16 November 2020 at 3.0 GHz. Using the average flux density of each sub-band over the VLA campaign, we found that the radio continuum counterpart spectrum can be fit with a power-law spectral index of −0.41 ± 0.04 (Methods).

To identify the host galaxy, an optical image (R′ band) was obtained using the Canada-France-Hawaii Telescope/MegaCam. Figure 2 shows the FRB location compared with deep optical, near-infrared (NIR) and radio images of the field. The optical image (R′ band) reveals the galaxy J160204.31−111718.5 at the location of the FRB, the light profile of which peaks at about 1″ southeast. Given the measured offset of 1.3″ between the FRB and the peak of the galaxy light profile, and the sky surface density of galaxies with this magnitude, we estimate a chance coincidence probability of 0.8% (Methods), indicating that J160204.31−111718.5 is the host galaxy of FRB 20190520B. The NIR

[1]National Astronomical Observatories, Chinese Academy of Sciences, Beijing, China. [2]Department of Physics and Astronomy, West Virginia University, Morgantown, WV, USA. [3]Center for Gravitational Waves and Cosmology, West Virginia University, Morgantown, WV, USA. [4]University of Chinese Academy of Sciences, Beijing, China. [5]Shanghai Astronomical Observatory, Chinese Academy of Sciences, Shanghai, China. [6]Cornell Center for Astrophysics and Planetary Science, and Department of Astronomy, Cornell University, Ithaca, NY, USA. [7]Cahill Center for Astronomy and Astrophysics, California Institute of Technology, Pasadena, CA, USA. [8]Owens Valley Radio Observatory, California Institute of Technology, Big Pine, CA, USA. [9]Canadian Institute for Advanced Research, Toronto, Ontario, Canada. [10]Xinjiang Astronomical Observatory, Chinese Academy of Sciences, Urumqi, China. [11]Research Center for Intelligent Computing Platforms, Zhejiang Laboratory, Hangzhou, China. [12]Institute of Astronomy, Graduate School of Science, The University of Tokyo, Tokyo, Japan. [13]Research Center for the Early Universe, The University of Tokyo, Tokyo, Japan. [14]Max-Planck-Institut für Radioastronomie, Bonn, Germany. [15]Kavli Institute for the Physics and Mathematics of the Universe (WPI), The University of Tokyo, Kashiwa, Japan. [16]CSIRO Space and Astronomy, Epping, New South Wales, Australia. [17]School of Science, Western Sydney University, Penrith South DC, New South Wales, Australia. [18]Jet Propulsion Laboratory, California Institute of Technology, Pasadena, CA, USA. [19]Department of Astronomy, School of Physics, Peking University, Beijing, China. [20]Department of Physics and Astronomy, University of Nevada, Las Vegas, Las Vegas, NV, USA. [21]These authors contributed equally: C.-H. Niu, K. Aggarwal, D. Li. ✉e-mail: dili@nao.cas.cn; wenfei@shao.ac.cn; claw@astro.caltech.edu

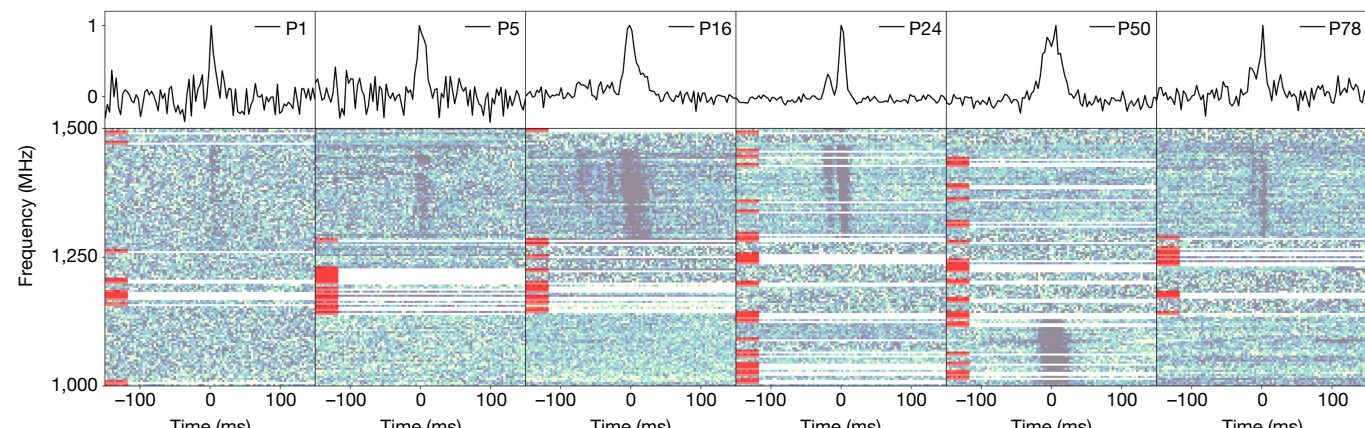

**Fig. 1 | Bursts from FRB 20190520B, shown as dynamic spectra and burst profiles.** Top: frequency-integrated burst normalized intensities that were detected during FAST observations. These six bursts are chosen from different observation epochs. The burst labels 'PXX' are in order of arrival time and the corresponding burst properties are shown in Supplementary Table 1. Bottom: de-dispersed dynamic spectra of the bursts, clearly showing their band-limited nature. The DMs are taken from Supplementary Table 1. The colour map is linearly scaled and darker patches represent higher intensities. The bad frequency channels are masked and labelled using red patches on the left. The time and frequency resolutions for the plots are downsampled to 0.786 ms and 3.91 MHz, respectively.

image (J band, 1.153–1.354 μm) obtained with Subaru and Multi-Object Infrared Camera and Spectrograph (MOIRCS)[13] most likely shows the stellar continuum emission of the host galaxy with an AB magnitude of 22.1 ± 0.1 in the J band, with FRB 20190520B and the radio continuum counterpart on the galaxy periphery.

We obtained an optical spectrum at the location of the FRB with the Double Spectrograph on the Palomar 200-inch Hale Telescope, which reveals the redshift of the putative host to be $z = 0.241 \pm 0.001$ based on a detection of strong Hα, [O III] 4,859 Å and [O III] 5,007 Å lines (Methods). A follow-up observation with the Low Resolution Imaging Spectrometer (LRIS) at the Keck I Telescope covering both the FRB location and the nearby Subaru J-band source along the extended R′-band structure indicates that the R′-band structure is dominated by the [O III] emission at the same redshift of $z = 0.241$. The Hα luminosity $L_{\text{Hα}} = 7.4 \pm 0.2 \times 10^{40}$ erg s$^{-1}$ after extinction correction implies a star-formation rate of about 0.41 $M_\odot$ yr$^{-1}$. On the basis of the J-band magnitude, we estimate the stellar mass of the host galaxy to be about $6 \times 10^8 M_\odot$. Thus, we characterize J160204.31−111718.5 as a dwarf galaxy with a relatively high star-formation rate for its stellar mass compared with local Sloan Digital Sky Survey (SDSS) galaxies[14]. At the luminosity distance of 1,218 Mpc implied by the redshift, the radio continuum counterpart has a spectral luminosity of $L_{\text{3GHz}} = 3 \times 10^{29}$ erg s$^{-1}$ Hz$^{-1}$.

The redshift of an FRB source is often estimated from the DM attributed to the intergalactic medium (IGM), $\text{DM}_{\text{IGM}} = \text{DM}_{\text{FRB}} - \text{DM}_{\text{host}} - \text{DM}_{\text{MW}}$, where MW is Milky Way. Theoretical calculations[15] and observations[1] have independently estimated the IGM contribution to FRB DM as a function of redshift (Fig. 3). For a DM contribution from the Milky Way of 113 ± 17 pc cm$^{-3}$ (including a ±40% range for the NE2001 model estimate[16] of the disk contribution and a uniform halo distribution from 25 pc cm$^{-3}$ to 80 pc cm$^{-3}$) combined with an assumed host-galaxy DM of only 50 pc cm$^{-3}$, the implied redshift range for FRB 20190520B is $z \approx 2.2$ to $z \approx 0.9$ for baryon fractions $f_{\text{IGM}}$ of 0.6 to 1 for the ionized IGM, much larger than the measured value (Methods).

Instead, using $\text{DM}_{\text{IGM}}(z = 0.241) \approx 195^{+110}_{-70}$ pccm$^{-3}$ (68% interval to account for the cosmic variance of $\text{DM}_{\text{IGM}}$ and 0.85 for the ionized baryon fraction[17]) combined with the Milky Way contribution, a very large host DM (disk + circumgalaxy + local to source contribution) of $\text{DM}_{\text{host}} = 903^{+72}_{-111}$ pccm$^{-3}$ is inferred from the posterior distribution. Over a broader range of ionized fractions from 0.6 to 1, $\text{DM}_{\text{host}}$ extends from 1,020 pc cm$^{-3}$ to 745 pc cm$^{-3}$.

In addition to the low chance coincidence probability, the measured DM and scattering properties of FRB 20190520B render it unlikely that J160204.31−111718.5 is a foreground galaxy with the true FRB host much farther in the background. The observed Hα emission implies a DM contribution from warm gas in the galaxy ranging from 230 pc cm$^{-3}$ to 650 pc cm$^{-3}$ (observer frame) for temperatures from $0.5 \times 10^4$ K to $5 \times 10^4$ K (Methods). The scattering contribution of a foreground galaxy depends not only on its DM contribution but also on the geometric leverage effect, which will increase the scattering by several orders of magnitude relative to scattering in the host galaxy. If J160204.31−111718.5 were a foreground galaxy, we estimate a fiducial range of scattering times from 0.6 s to 20 s at 1.25 GHz, orders of magnitude larger than the observed mean scattering time of 10 ± 2 ms. Although this estimate depends on parameters such as the ionized gas temperature, path length through the galaxy and the degree of turbulence in the gas, which are not well constrained, the observed scattering time is probably too small for the proposed host galaxy to lie in the foreground (Methods).

The large $\text{DM}_{\text{host}}$ inferred for FRB 20190520B demonstrates that the distribution of $\text{DM}_{\text{host}}$ values for the FRB population has a long tail, which may add considerable variance to estimates for the IGM contribution. It is conceivable that the $\text{DM}_{\text{host}}$ distribution may differ for repeating and non-repeating FRBs, which could make non-repeating FRB DMs more accurate proxies for redshift. To understand this further, a larger sample of precisely localized FRBs with measured host-galaxy redshifts is needed to statistically characterize host galaxies, their circumgalactic media and near-source environments along with the IGM[18]. Accordingly, searches need to accommodate large values of $\text{DM}_{\text{host}}$ as part of the DM budget for FRB sources. Large $\text{DM}_{\text{host}}$ contributions may also imply large scattering, which can also reduce search sensitivity.

The co-located radio continuum counterpart, the star-forming dwarf host galaxy and the high repetition rate make FRB 20190520B a clear analogue to FRB 20121102A, the first known repeating FRB[19] and the first to be identified with a compact, luminous persistent radio source (PRS)[5,20] ($L_{\text{1.6GHz}} \approx 3 \times 10^{29}$ erg s$^{-1}$ Hz$^{-1}$). Another repeating source, FRB 20201124A, was also associated with a radio continuum counterpart[21–23]; however, optical spectroscopy and radio interferometric measurements demonstrated that the persistent radio emission was spatially extended and consistent with an origin in star formation in the host galaxy[24]. By contrast, the continuum counterpart to FRB 20190520B appears to not be from star formation because its

## Table 1 | Properties of FRB 20190520B

| Burst parameters | |
|---|---|
| Right ascension (J2000) | 16 h 02 min 04.272 s (0.1″) |
| Declination (J2000) | −11° 17′ 17.32″ (0.08″) |
| Galactic coordinates ($l$, $b$) | 359.67°, 29.91° |
| Number of detections[a] | 88 |
| Mean total DM (pc cm$^{-3}$) | 1,204.7 ± 4.0 |
| Mean burst width (ms) | 13.5 ± 1.2[b] |
| Mean scattering timescale (ms) at 1.25 GHz | 10 ± 2[b] |
| Measured fluence (Jy ms) | 0.03 to 0.33 |
| $DM_{MW,disk}$, $DM_{MW,halo}$ (pc cm$^{-3}$)[c] | 60, 50 |
| $DM_{host}$ (pc cm$^{-3}$) | $903^{+72}_{-111}$ |
| Luminosity distance (Mpc) | 1,218 |
| Isotropic equivalent energy (10$^{37}$ erg) | 3.6 to 40 |
| **Persistent radio source** | |
| Right ascension (J2000) | 16 h 02 min 04.261 s (0.1″) |
| Declination (J2000) | −11° 17′ 17.35″ (0.05″) |
| Flux density at 3.0 GHz (μJy) | 202 ± 8[d] |
| Luminosity at 3.0 GHz (erg s$^{-1}$ Hz$^{-1}$) | 3 × 10$^{29}$ |
| Size (3 GHz) | <0.36″ (<1.4 kpc) |
| **Host galaxy** | |
| Redshift ($z$) | 0.241 ± 0.001 |
| Stellar mass ($M_\odot$)[e] | ~6 × 10$^{8}$ |
| Hα luminosity (erg s$^{-1}$) | 7.4 × 10$^{40}$ ± 0.2 × 10$^{40}$ |
| Star-formation rate ($M_\odot$ yr$^{-1}$)[f] | ~0.41 |

[a]Including the FAST and VLA observations.
[b]These are the mean values from FAST bursts fitted with a Gaussian pulse convolved with a one-sided exponential (Methods).
[c]The MW electron density model from NE2001 and YMW16.
[d]Averaged over a span of about 2 months from 30 August to 16 November 2020.
[e]Based on the J-band magnitude.
[f]Based on the Hα luminosity.

luminosity would imply a star-formation rate of about 10 $M_\odot$ yr$^{-1}$, a factor of 25 larger than that measured for the host galaxy and a factor of five larger than the highest observed star-formation rate for galaxies of this mass[14]. Given its extreme luminosity, unresolved structure in

VLA observations and offset from the peak optical emission in the host galaxy, we conclude that the radio continuum counterpart is a compact source (less than 1.4 kpc) physically connected to FRB 20190520B, a PRS like that associated with FRB 20121102A (which has a very long baseline interferometry (VLBI) confirmed size of less than 0.7 pc (ref. [20])).

More than a dozen FRBs were localized before FRB 20190520B, including five repeating sources[25,26], but only FRB 20121102A had been associated with a compact PRS. FRB 20121102A also demonstrates a sporadically large burst rate[27] (for example, a peak burst rate of 122 h$^{-1}$), a substantial RM that varies over both short (burst to burst) and long (year) timescales[3,4] and $DM_{host}$ as large as about 300 pc cm$^{-3}$, suggesting that burst activity may be correlated with both relativistic plasma emitting synchrotron radiation and the presence of thermal plasma in the local FRB environment. In addition to a PRS, FRB 20190520B shows a $DM_{host}$ that is almost three times larger than that of FRB 20121102A, and may have a comparably large RM (Methods). If a sizeable fraction of $DM_{host}$ is from thermal plasma in the circum-source medium, then perhaps the presence of a PRS and a dynamic magneto-ionic environment are correlated with FRB formation, and repeating FRBs like FRB 20121102A and FRB 20190520B are younger sources that still reside in their complex natal environments.

However, until further VLBI observation constraints become available, it is unclear how much of the large $DM_{host}$ for FRB 20190520B is attributable to ionized gas in the circum-source medium versus the host galaxy's interstellar medium. Moreover, other repeating FRBs have very deep limits on PRS counterparts[28,29], which complicates the connection between burst and magneto-ionic activity and PRS luminosity. For example, FRB 20200120E has an upper limit on persistent radio luminosity of $L_{1.5GHz} < 3.1 \times 10^{23}$ erg s$^{-1}$ Hz$^{-1}$ (ref. [29]). It is associated with an old stellar population and has a modest $DM_{host} \lesssim 50$ pc cm$^{-3}$, which is in sharp contrast to FRB 20121102A and FRB 20190520B. One possibility is that there are multiple kinds of sources that can emit FRBs, a point that has been argued based on burst rate and phenomenology[30,31]. Alternatively, the source properties may evolve as the source ages, the PRS fades, the event rate drops and the surrounding plasma dissipates[32]. The similarity of FRB 20190520B to FRB 20121102A suggests a potential connection between burst activity, the presence of a PRS and $DM_{host}$ for at least some FRBs.

Various methods have been used to argue that repeating and non-repeating FRBs comprise different subclasses[3,9,30], either at different evolutionary phases or entirely different physical scenarios. Although the observed burst repetition and morphology can be time

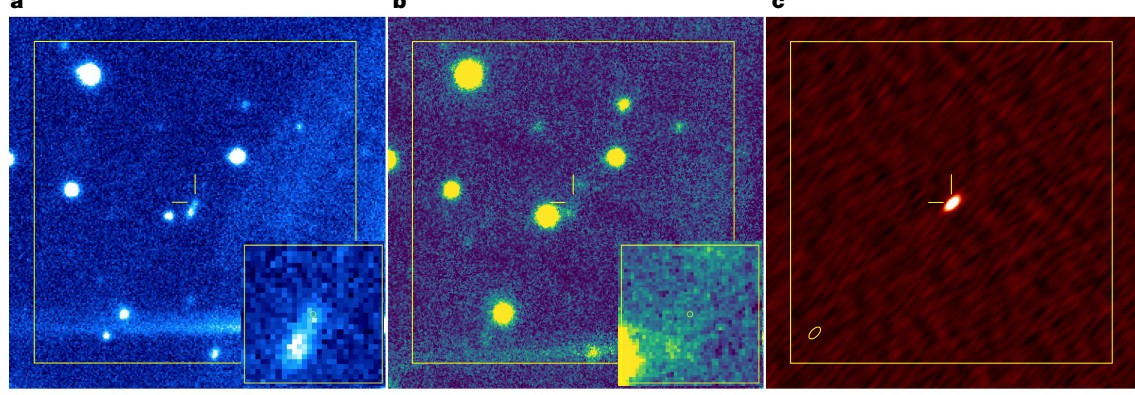

**Fig. 2 | Optical, infrared and radio images of the field of FRB 20190520B.** In each case, the box is 40″ in size and the 2″ crosshairs indicate the best FRB position at RA = 16 h 02 min 04.272 s, dec. = −11° 17′ 17.32″ (J2000). North is up and east is to the left. **a**, An optical R′-band image obtained by CFHT MegaCam covers 5,427–7,041 Å, including redshifted Hβ 4,861 Å, [O III] 4.959 Å and [O III] 5,007 Å emission lines from the host galaxy. The bright streak in the lower left is an artefact caused by a bright star outside the field of view. The inset shows the host galaxy in a 5″ region centred on the FRB position, as indicated by the yellow circle (0.1″ radius, corresponding to the 1σ position uncertainty). **b**, The infrared J-band image by Subaru/MOIRCS shows emission only at the location of the peak of the optical light profile of the host galaxy. The inset is a 5″ region matching the inset in **a**. **c**, The radio VLA image (2–4 GHz) shows a compact persistent source at the FRB location. The synthesized beam is shown as an ellipse of size (0.92″ × 0.47″) in the left corner.

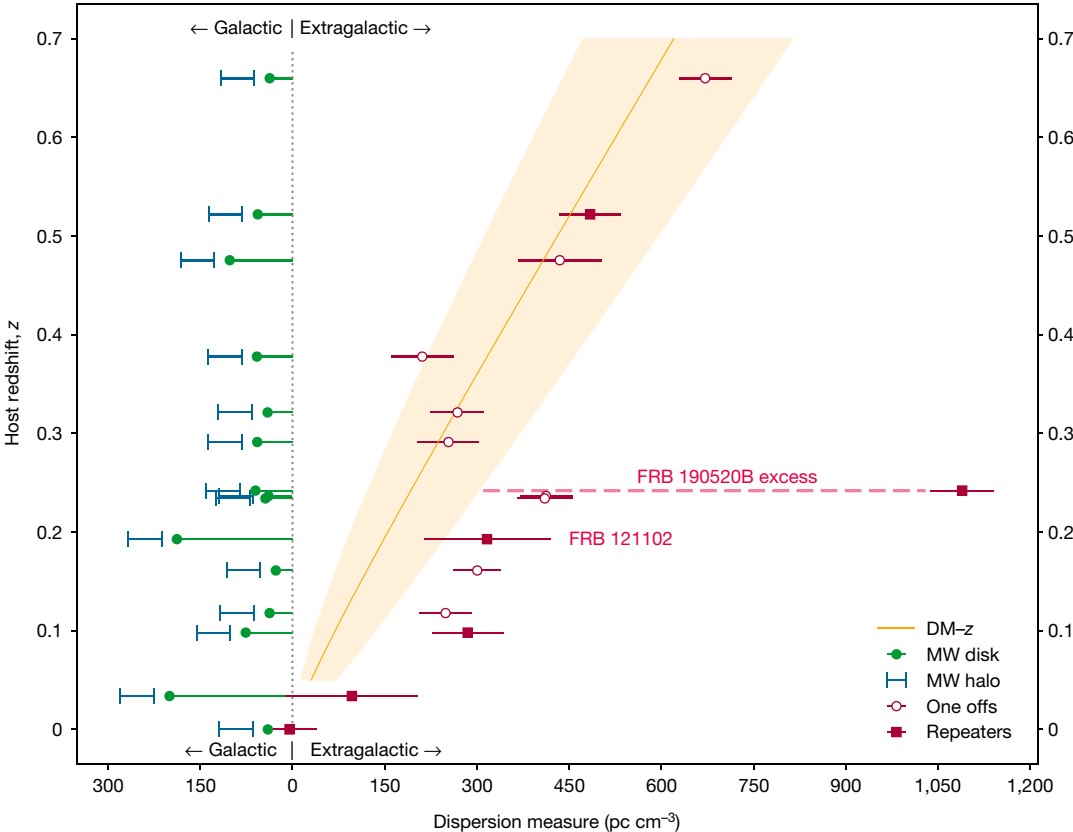

**Fig. 3 | Galactic and extragalactic contributions to the DM observed for 15 FRBs.** Theses 15 FRBs have firm host-galaxy associations and redshifts. Sources localized on initial detection ('one off' bursts) and repeating FRBs are identified separately. Galactic disk contributions are estimated from NE2001[16] with ±20% uncertainties, along with an additional halo contribution of 25–80 pc cm$^{-3}$ (full range). The red error bars on each extragalactic DM estimate represent a conservative full range uncertainty derived by adding the disk and halo ranges. The expected median DM contribution of the IGM and the inner 68% confidence interval are shown by the orange line and shaded region (Methods). The host-galaxy contributions DM$_{host}$ shift observed values to the right of the band of extragalactic DM predicted for the IGM alone. FRB 20190520B is a clear outlier from the general trend, with an unprecedented DM contribution from its host galaxy. FRB 20121102A is identified for comparison.

dependent owing to various mechanisms[33], PRS emission and DM$_{host}$ may reflect more persistent aspects of the FRB environment and thus may be more reliable tracers of any putative subclasses. Further progress will result from better characterization of the full range of host-galaxy DMs, which will also help mitigate biases in the DM–redshift relation[1].

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

## Methods

### Observations

**FAST.** CRAFTS is a multi-purpose drift scan survey conducted with FAST using a 19-beam receiver operating at 1.05–1.45 GHz, deployed in May 2018 and conducting blind FRB searches using multiple pipelines[34]. FRB 20190520B was discovered on 16 November 2019 in archived CRAFTS data, which are in 1-bit filterbank format with 196-μs time resolution and 0.122-MHz frequency resolution. In this first discovery observation, 3 bursts were detected in 10 s, and another burst was detected 20 s later. These 4 bursts from the drift scan survey gave a preliminary location for the source within an approximately 5-arcmin-diameter region. Taking the pointing location from FAST, 2 follow-up observations were performed with FAST on 25 April and 22 May in 2020 with 19-beam mode in which 15 bursts were detected. A monthly observation campaign was then conducted by FAST using the approximately 100-mas localization from the VLA (next section). After some regular telescope maintenance, 10 observations were performed spanning from 30 July 2020 to 19 September 2020 in which 60 more bursts were detected.

Bursts were detected from FRB 20190520B in each FAST monitoring observation. We list the properties of those bursts detected by FAST in Supplementary Table 1. The burst arrival time is in modified Julian date (MJD) format and has been transformed to the arrival time at the Solar System barycentre at 1.5 GHz with the DM values from Supplementary Table 1. The observed $DM_{obs}$ is measured using the method from ref. [9] and the code from ref. [35]. We use the DM value that maximizes the structure of the highest signal-to-noise ratio (S/N) burst on a given date, and report this same DM for all other bursts detected on that date. Apparent epoch-to-epoch DM variations may simply be the result of a combination of variable effects, including intrinsic pulse structure, time–frequency drift, flux distribution in frequency and scattering (which affects bursts chromatically).

The FAST data were searched using a Heimdall-based pipeline[34,36]. For the FRB blind search in the 19-beam drift scan survey, the polarizations were added and only the total intensity (Stokes I) was recorded. The trial DM range was from 100 pc cm$^{-3}$ to 5,000 pc cm$^{-3}$ and we matched the pulse width using a boxcar search up to 200 ms. The candidates with S/N > 7 and present in fewer than 4 adjacent beams were manually examined for further inspection. After we identified FRB 20190520B as a new FRB source, the follow-up burst search was done with a narrower DM range (100–2,000 pc cm$^{-3}$). Following the VLA localization, the tracking observations only recorded data from the central beam but with all the Stokes parameters and higher time resolution (about 50 μs). Candidates with S/N > 7 correspond to a fluence threshold of 9 mJy ms for a burst width of 1 ms.

The pulse widths were estimated by a Gaussian profile fit if the burst showed no evidence of scattering (see section on scattering below). A subpulse is recognized if the bridge between two closely spaced peaks drops more than 5$\sigma$ below the higher peak. If the burst shows a scattering tail, the pulse width is derived from a combined fit for the Gaussian width and scattering time. We roughly estimate the bandwidth of each burst by dividing the whole bandpass into 50-MHz subbands and identifying the subbands containing burst emission. The burst fluences were determined using the bandwidth, temporal width and amplitude of each burst.

**VLA.** Following the FAST detection of bursts from FRB 20190520B, VLA observations were performed from July–November 2020 with director's discretionary time (DDT) project 20A-557 at the most reliable position determined using FAST (good to 5 arcmin). Most of the observations were in the B array configuration (with a maximum baseline of 11.1 km), with the exceptions that the array configuration was BnA on MJD 59,161 and BnA->A on MJD 59,167 and MJD 59,169. In total, 11.4 h were spent on the source at multiple bands. The total bandwidths for the L (1.5 GHz), S (3 GHz) and C (5.5 GHz) bands were 1,024 MHz,

2,022 MHz and 2,022 MHz, respectively, with 1,024 channels, corresponding to channel bandwidths of approximately 1 MHz, 2 MHz and 2 MHz, respectively.

The details of the observations are given in Extended Data Table 1. The telescopes were pointed at the field centred at (RA, dec.) (J2000) = (16 h 02 min 01 s, −11° 17′ 28″). We used the 'realfast' search system at the VLA to search for bursts from FRB 20190520B in our VLA observations. Observations on MJD 59,169 contributed to the imaging at S band, but the 'realfast' system was not run on this day owing to a system error.

The 'realfast' search system has been described in detail in refs. [12,37], but here we discuss it briefly. Using a commensal correlator mode, the visibilites are sampled with 10-ms resolution and distributed to the 'realfast' graphics-processing-unit cluster to search for transients. The search pipeline rfpipe[38] then applies the online calibration, de-disperses the visibilities and forms images at a set of different temporal widths. The 8$\sigma$ fluence limit of a 10-ms image is 0.29 Jy ms, 0.18 Jy ms and 0.13 Jy ms for the L, S and C bands, respectively. Candidates with an image S/N greater than the threshold triggered the recording of fast sampled visibilities in windows ranging from 2 s to 5 s centred on each candidate. For each candidate, time–frequency data corresponding to the maximum pixel in the image are then extracted and post-processed. The candidate is then classified by fetch, a graphics-processing-unit-based convolutional neural network[39].

Promising candidates selected visually by the 'realfast' team go through an offline analysis to refine the candidate parameters and to improve its detection significance, if possible. Several methods are tried within this analysis: offline search for the transient using a finer DM grid, varying radio frequency interference (RFI) flagging thresholds, changing image size (as the real-time search is done on non-optimal image sizes owing to computational limitations), subband search and so on (see section 2.4 of ref. [37] for more details). The pixel sizes of images formed by the real-time system for the L, S and C bands were 0.9″, 0.48″ and 0.27″, respectively, with an image size of 2,048 × 2,048 pixels corresponding to field sizes of 0.5°, 0.27° and 0.15°, respectively. During refinement, we searched the data with smaller pixel sizes: 0.5″ × 0.75″ for the L band, 0.38″ × 0.28″ for the S band and 0.27″ × 0.18″ for the C band. The significance of an astrophysical transient improves when the data is de-dispersed at a DM closer to the true DM of the candidate (see section 2 of ref. [40]). Therefore, we re-ran the search with a finer DM grid at 0.1% fractional sensitivity loss, compared with 5% used by the real-time system. Noise-like events or RFI are sensitive to RFI flagging and image gridding parameters, and so they cannot be reproduced on refinement and are discarded. Sometimes, the transient signal is present only in a fraction of the whole frequency band, so we also re-run our search only using the relevant frequencies, which further improves the detection significance. We applied these techniques on all the candidates selected from these observations to refine their parameters.

### Localization of bursts

**Calibration.** The 'realfast' system makes several assumptions during calibration and imaging to improve the computational efficiency for the real-time imaging and search. Moreover, the point spread function (PSF) of the interferometer is not deconvolved from the image, so the real-time system forms 'dirty' images. The PSF shape makes the images more difficult to visually interpret and model.

To address these issues, we used a top-hat function to select the raw, de-dispersed, fast-sampled visibilities containing only the burst signal to re-image the burst data using the Common Astronomy Software Application package (CASA v5.6.2-3)[41]. Observations of 3C 286 (acquired before the FRB observations) were used to calibrate the flux density scale, bandpass and delays. Complex gain fluctuations over time were calibrated with observations of the calibrator J1558−1409. We performed phase-reference switching at intervals

of 16 min, 13 min and 12 min for the L, S and C bands, respectively, consistent with the nominal phase-coherence timescales for the VLA, so that after calibration any short-timescale phase variations are negligible. Therefore, the systematic errors for the burst positions on short timescales are of the same magnitude as those for the deep imaging, as discussed below.

**Determining properties of individual bursts.** After calibration, the CASA task tclean was used to generate an image for each burst and estimate the S/N. Most of the bursts are spectrally confined, so for each burst we select the spectral window range that produces the highest image S/N. We then use the CASA task imfit around the FRB position to fit an elliptical Gaussian to the source in the radio image and measure the centroid location, peak flux density and $1\sigma$ image-plane uncertainties. Hereafter, we refer to these image-plane uncertainties as statistical errors on burst positions.

The average burst positions are calculated for each frequency band separately, by weighting each detection by the inverse of the position fit errors (statistical errors) reported by CASA. Statistical errors are inversely proportional to the S/N, and therefore high significance detection is expected to have smaller fit errors. The total positional error at each frequency band is obtained by adding this statistical error in quadrature with the systematic error in that band (determined using the deep radio images described below). The total positional error (RA error, dec. error) is (0.25″, 0.32″) for the L band, (0.28″, 0.17″) for the S band and (0.12″, 0.09″) for the C band. In the final analysis, the errors are dominated by systematic uncertainties at each frequency band (Extended Data Tables 2, 3). The final burst position was obtained by taking a weighted average of the burst positions at each of the three frequency bands, using the inverse of the total error at each frequency band as the weight. The best estimate of the burst position is RA = 16 h 02 min 04.272 s, dec. = −11° 17′ 17.32″ (J2000). We estimate the error on this position to be 0.10″ and 0.08″. We calculated the reduced chi-square value of individual burst positions with respect to this best estimate of the FRB position. The reduced chi-square was given by $\chi^2 = (1/8) \sum_i (\theta_i - \theta_{\mathrm{best}})^2/\sigma_i^2$, where $\theta_i$ refers to the coordinates of a burst, $\theta_{\mathrm{best}}$ refers to the best estimate of the FRB position, $\sigma_i$ is the total error on a burst (statistical and systematic errors added in quadrature) and the sum is over all the 9 localized bursts. We obtained a reduced chi-square value of 0.53 and 0.77 for RA and dec. respectively.

We confirmed that there were no short-timescale phase errors by performing intermediate-timescale imaging of the continuum data. We imaged segments of 5–30 s, such that at least 1–3 sources could be detected at each frequency, and inspected the position variations of those sources over time. At all frequencies, the positions were stable and consistent with the statistical uncertainty (radiometer noise). Offsets of the sources were within the range of the systematics quoted for the PRS in the next section.

Owing to the low time resolution of the VLA data (10 ms), we do not perform any sophisticated modelling of the burst properties. The properties of the VLA bursts, including the flux densities, were obtained using CASA and the 'realfast' system, and are given in Extended Data Table 4. Here the reported widths should be considered as upper limits. The frequency extent of the burst signal was visually determined and reported in the last column. We also could not estimate a structure maximizing DM for the VLA bursts because of the coarse time resolution. The DMs reported in this table are the values that maximize the S/N. This also explains the apparent variability seen in the DM values.

## Persistent radio source
**Deep radio images and PRS.** The VLA campaign obtained two epochs at 1.5 GHz and six epochs at 3 GHz and 5.5 GHz, which resulted in an on-source time of about 3 h for 1.5 GHz, and about 4 h for both 3 GHz and 5.5 GHz.

Along with the 'realfast' output, the VLA visibilities with 3 s or 5 s sampling times were saved and analysed to search for persistent radio emission. This is done in parallel with saving the high-time-resolution (10 ms) data around the burst that was used in the previous section. We used the same data reduction, flagging and calibration approach that was used for burst localization. We then performed further flagging on the target and then subsequently imaged its Stokes I data using the CASA deconvolution algorithm tclean. To balance sensitivity while reducing sidelobes from a nearby bright source, we imaged with a Briggs weighing scheme (robust=0). In addition, self-calibrations were performed for all observations to correct considerable artefacts from the close-by bright sources in the field. We made use of the CASA task imfit to measure source flux densities by fitting an elliptical Gaussian model in the image plane.

We stacked observations at each central frequency in the $u$–$v$ plane and then imaged the Stokes I intensity, resulting in deep images at 1.5 GHz, 3 GHz and 5.5 GHz with root-mean-square (r.m.s.) noise of 9.0 μJy per beam, 4.5 μJy per beam and 3.0 μJy per beam, respectively. At 1.5 GHz, 3 GHz and 5.5 GHz, the upper limits of the deconvolved sizes of the PRS are as large as 1.4″ × 0.89″, 0.51″ × 0.14″ and 0.36″ × 0.1″, respectively. Thus, a conservative upper limit of the size of the PRS from VLA observations is 0.36″, which corresponds to 1.4 kpc at the angular diameter distance of 809 Mpc. The obtained positions for the PRS are shown in Extended Data Table 2 and the positions of the bursts and PRS are shown in Extended Data Table 1. The systematic offsets on these positions are estimated in the next section.

**Systematic offsets.** To determine the systematic errors on the coordinates of the PRS that we determined from the deep images, we ran the PyBDSF (https://www.astron.nl/citt/pybdsf/index.html) package to extract radio sources from the deep images. We then cross-matched the detected point sources in the deep images with the sources listed in the optical PanSTARRS survey DR1[42]. We identified the radio point sources using the following criteria: (1) the peak intensity (Jy per beam) of a source should be 0.7, 0.5 and 0.5 times higher than its integrated flux (Jy) for the 1.5-GHz, 3-GHz and 5.5-GHz images, respectively; (2) the S/N (peak intensity/local r.m.s. noise) of a source should be greater than 5.

In total, we detected 375, 113 and 43 sources in the 1.5-GHz, 3-GHz and 5.5-GHz deep images, respectively. The selected sources were also visually checked to make sure that they are 'point like' sources in the deep images. We searched for matching optical sources within radii of 0.5″, 1.0″ and 2.0″. Going from 1.0″ to 2.0″, the additional cross-matched sources at each band are consistent with chance coincidence. Therefore, we adopt a 1.0″ cross-match radius, finding 136, 31 and 9 sources with optical counterparts in the PanSTARRS DR1 catalogue at 1.5 GHz, 3 GHz and 5.5 GHz, respectively. These cross-matched sources were used to determine the astrometry for FRB 20190520B and its PRS. By subtracting PanSTARRS coordinates from VLA coordinates, we estimate the systematic offsets in the 1.5-GHz, 3-GHz and 5.5-GHz positions, as listed in Extended Data Table 2. The systematic offsets are consistent with zero mean, and their uncertainties dominate the uncertainties of the PRS position.

**Determining the position of the PRS.** To determine the position of the PRS in the three frequency bands, we followed a procedure similar to what was used to determine the burst positions (see 'Localization of bursts'). The best estimate for the PRS position is (RA, dec.) (J2000) = (16 h 02 min 04.261 s, −11° 17′ 17.35″). The error on this position is estimated to be 0.10″ and 0.05″ for RA and dec., respectively.

**Variability and spectrum of the PRS.** The flux density of the source measured at each epoch is shown in Extended Data Table 1. The measured flux densities show variations that are mostly consistent with measurement errors, but there are roughly $2\sigma$ variations in the 2.5-GHz subband that, if real, could be refractive interstellar scintillation or

intrinsic variations, or both. To study the spectrum of the PRS, we split each of the observations into two 0.5 GHz/1 GHz subbands. Then we measured the average flux density at each of the subbands over the campaign. The multiband data were fit with a power-law model ($S_\nu \propto \nu^\alpha$, where $S_\nu$ is the observed flux density at the frequency $\nu$, and $\alpha$ is the spectral index), yielding an average spectral index for the PRS of $-0.41 \pm 0.04$ (Extended Data Fig. 2).

**Chance coincidence association of the PRS.** In the 1.5-GHz deep image, we detected 8 'point like' sources, including the PRS, within 5 arcmin of the phase centre and with a flux density no less than that of the PRS (at 260 μJy) based on our point-source selection criteria described earlier. There is an additional bright source with a flux density of a few tens of millijansky in the region, but it was not classified as a 'point like' source based on our criteria.

To estimate the chance coincidence probability of the PRS with the bursts, we compared the solid angle corresponding to the uncertainty of burst localization and the average solid angle occupied by each of the eight 'point like' sources in the FRB field of view. The solid angle corresponding to each of the 8 'point like' sources is roughly estimated as $S_{source} = \pi(5/60)^2/8$ steradians (Sr). The offset between the average position of the nine bursts and the position of the PRS at 5.5 GHz, which is best constrained when taking both statistical and systematic errors into account, is about 0.06 arcsec. This, along with a statistical error of 0.01 arcsec and a conservative estimate of the systematic error of 0.12 arcsec, can be used to estimate the offset between the PRS and the FRB position. We conservatively estimate this offset to be 0.19 arcsec. The solid angle corresponding to the offset therefore can be estimated as $S_{offset} = \pi(0.19/60/60)^2/8$ Sr. The ratio between $S_{offset}$ and $S_{source}$ gives the chance of coincident association of the PRS with the FRB position to be about $3 \times 10^{-6}$.

**Galaxy photometry and redshift determination.** The deep R′-band (5,427–7,041 Å) images obtained by CFHT/MegaCam were stacked from archival observation data taken in 2014–2015 by the Canada France Hawaii Telescope (CFHT) archival pipeline MEGAPIPE, with a total of about 3.6 h on the field. The level 3 (flux calibrated) images were retrieved for our analysis.

NIR J-band images of the FRB−20190520B field were taken under a relatively poor seeing condition (about 1.3″) through a Subaru target-of-opportunity observation on 5 August 2020. A total of 1.4 h of observations were used for the final combined J-band image shown in Fig. 2. A J-band source of $22.07 \pm 0.14$ mag (AB) was detected at about 1″ southeast of the burst location, and is possibly stellar emission from the host galaxy. A faint source 2.5″ north has $22.87 \pm 0.26$ mag in the J band. Neither of these two sources were detected in the Ks-band image, with a $5\sigma$ limit of 21.74 mag (1.1 h).

An optical spectrum was obtained with the Double Spectrograph (DBSP) on the Palomar 200-inch telescope on 24 July 2020 using a 1″ slit width. This observation was executed before the CFHT/MegaCam archival data on FRB 20190520B field were found, and only PanSTARRS images were used for observation planning. The slit of DBSP was set to cover the PRS emission at RA = 16 h 02 min 04.27 s, dec. = −11° 17′ 17.5″ detected by the VLA in L-band on 22 July 2020. No clear optical counterpart was detected in any of the five band images of PanSTARRS from DR1[42]. The slit was guided by the nearby M-star at RA = 16 h 02 min 04.48 s, dec. = −11° 17′ 19.1″ as reported in the PanSTARRS DR1 catalogue, with $i = 20.4$ mag, and 3.4″ due east of the PRS coordinates. The slit was set to a position angle of 108.5°, ensuring that both the PRS and the M-star fell within the slit. The observations with $2 \times 900$ s exposures were carried out under photometric sky conditions and subarcsecond seeing. The two-dimensional spectrum was generated using the image reduction and analysis facility (IRAF), including bias removal, flat-fielding and reduction of other instrumental effects. The one-dimensional spectrum was extracted from a 1.5″ window.

The standard star BD+284211 was used for telluric correction and flux calibration. The DBSP one-dimensional spectrum is shown in Extended Data Fig. 3. The flux scale of the spectrum does not include the slit loss and registration error of PanSTARRS coordinates of the M-type star. The [O III] 5,007 Å line and the Hα line are both well detected (>5σ). The two emission lines are narrow, each with a full-width at half-maximum of about 10 Å. The redshift derived from these two spectral lines is $z = 0.241 \pm 0.001$.

A follow-up Keck LRIS spectroscopic observation was carried out on 25 August 2020 under reasonable weather and seeing conditions (1.1″). The 1.5″ slit was set at a position angle of 160° to the extended optical emission seen in the MegaCam R′-band image around the FRB 20190520B location. A total exposure of 3,600 s was obtained. The emission lines Hα, Hβ, [O III] 4,859 Å, and [O III] 5,007 Å are well detected, indicating that the extended R′-band structure has the same redshift of $z = 0.241$. After the spectrum was corrected for Galactic foreground extinction, the line flux ratio between Hα and Hβ was used to estimate an extinction of $A_V = 0.80$ mag, assuming a case-B line ratio of 2.86, yielding an Hα flux $F_{H\alpha} = (42.0 \pm 0.5) \times 10^{-17}$ erg s$^{-1}$ cm$^{-2}$. The extinction-corrected Hα luminosity is $L_{H\alpha} = 7.4 \pm 0.2 \times 10^{40}$ erg s$^{-1}$.

**Chance association probability of the host galaxy.** We use the approach of ref. [43] to estimate the chance coincidence probability of the association between the galaxy (J160204.31−111718.5) and FRB 20190520B. Assuming a uniform surface distribution of galaxies, the probability of chance coincidence follows a Poisson distribution; that is, $P = 1 - \exp(-n_i)$, where $n_i$ is the mean number density of galaxies brighter than a specified R′-band magnitude in a circle of given radius, determined by the half light radius of the galaxy and the burst localization error region. This number density is estimated using the results from ref. [44]. From previous discussion, we assume the localization error on the FRB position to be about 0.1″, and the size of the host galaxy to be 0.5″. The R′-band magnitude of the possible host is difficult to estimate as it is significantly affected by the emission lines. Thus, we conservatively estimate the R′-band magnitude to be ≲23.3″ mag assuming a flat spectral energy distribution in $\nu L_\nu$ between the R′ band and the J band, where $\nu$ and $L_\nu$ are frequency and luminosity, respectively. The probability of a chance association between FRB 20190520B and J160204.31−111718.5 is estimated to be about 0.8%, independent of constraints from the association with a rare PRS or the observed FRB scattering. The next nearest galaxy is 6.5″ away from the FRB and has a chance coincidence probability of >20%.

## FAST burst sample analysis

**Repetition rate.** FRB 20190520B was found to be regularly active with two or more bursts detected in each monitoring observation with FAST. Owing to the possible clustering behaviour of FRB emission, a Weibull distribution was used for the waiting time $\delta$ (the separation between adjacent detected bursts)[45]

$$\mathcal{W} = \frac{k}{\lambda}\left(\frac{\delta}{\lambda}\right)^{k-1} e^{-(\delta/\lambda)^k}, \qquad (1)$$

where $k$ is the shape parameter and $\lambda$ is the scale parameter of the distribution. The Weibull distribution reduces to a Poisson distribution when $k = 1$. The mean waiting time, $E(\delta) = \lambda \Gamma(1 + 1/k)$, where $\Gamma$ is the gamma function, implies a burst rate $r = 1/E(\delta)$. Extended Data Fig. 4 shows the inferred parameter distributions obtained with a Markov chain Monte Carlo. We find the burst rate of FRB 20190520B is $r = 4.5^{+1.9}_{-1.5}$ h$^{-1}$ with shape parameters $k = 0.37^{+0.04}_{-0.04}$ when all 79 bursts above 9 mJy ms are used (left panel in Extended Data Fig. 4). Waiting times shorter than 1 s include substructure in individual bursts, so we also analyse only longer wait times $\delta \geq 1$ s, yielding $r = 5.3^{+1.1}_{-1.0}$ h$^{-1}$ with shape parameters $k = 0.76^{+0.09}_{-0.08}$ (right panel in Extended Data Fig. 4).

**Short- and long-timescale periodicity search.** For a range of trial periods ($P$) and period derivatives ($\dot{P}$), the arrival times of the 79 FAST bursts were folded to calculate the phase of each burst for a given trial period. For the short-timescale periodicity search, the arrival times were folded at $P$ between 1 ms and 1,000 s and $\dot{P}$ between $10^{-12}$ s s$^{-1}$ and 1 s s$^{-1}$. For the long-timescale periodicity search, the trial range of period $P$ was from 2 d to 365 d and the range for period derivative (to account for spindown) was $\dot{P}$ from $10^{-12}$ d d$^{-1}$ to 1 d d$^{-1}$. The longest contiguous phase segment without bursts was calculated for each fold trial as a signature for possible periodicity.

For the short-timescale periodicity search, the bursts spread to a phase window larger than 60% of one period, which indicates no periodicity pattern in the trial range. For the long-timescale periodicity search, the longest phase segment without bursts showed two maxima about 0.6 near 67 d and 169 d. These two maxima are reproduced by folding the MJDs of all observation sessions, which indicates that the maxima reflect the sampling window rather than any burst periodicity. Thus, no long or short period of FRB 20190520B was detected.

**Energy distribution.** A 1-K equivalent noise calibration signal was injected before each observation session to obtain a high-quality flux density and energy calibration measurement for each detected burst. The injected noise calibration signal was used to scale the data into temperature units, yielding a nearly constant r.m.s. radiometer noise to within 6%. We then converted from units of kelvin to jansky using the zenith-angle-dependent gain curve, provided by the observatory through quasar measurements[46]. For most observations, the zenith angle is less than 20°, which corresponds to a stable gain of about 16 K Jy$^{-1}$.

Assuming bursts have spectral index of about 0, we calculated their isotropic equivalent burst energies, $E$, following equation (9) of ref.[17]:

$$E = 10^{39} \text{erg} \times \frac{4\pi}{1+z}\left(\frac{D_L}{10^{28}\text{cm}}\right)^2\left(\frac{F_\nu}{\text{Jy ms}}\right)\left(\frac{\nu_c}{\text{GHz}}\right), \qquad (2)$$

where $F_\nu = S_\nu \times W_{eq}$ is the specific fluence in units of erg cm$^{-2}$ Hz$^{-1}$ or Jy ms, $S_\nu$ is the peak flux density, which has been calibrated with the noise level of the baseline, giving the flux measurement for each pulse at a central frequency of $\nu_c = 1.25$ GHz, $W_{eq}$ is the equivalent burst duration, and the luminosity distance $D_L = 1,218$ Mpc corresponds to a redshift $z = 0.241$ for the source of FRB 20190520B, using standard cosmological parameters[47]. The fluence-width distribution at 1.25 GHz for FRB 20190520B bursts can be seen in Extended Data Fig. 5. The histogram of burst energies (Extended Data Fig. 6) exhibits a bump that we fit with a log-normal function.

**DM inventory analysis.** The observed DM can be separated into four primary components (all in the observer's frame):

$$\text{DM}_{obs} = \text{DM}_{MW} + \text{DM}_{MW,halo} + \text{DM}_{IGM} + \text{DM}_{host} \qquad (3)$$

where $\text{DM}_{MW}$ is the contribution from the Milky Way's interstellar medium, $\text{DM}_{halo}$ is the contribution from the Milky Way halo, $\text{DM}_{host}$ the contribution from the host galaxy including its halo and any gas local to the FRB source, and $\text{DM}_{IGM}$ is the contribution from the IGM.

We use the NE2001 model[16] to evaluate $\text{DM}_{MW} = 60$ pc cm$^{-3}$ (compared with 50 pc cm$^{-3}$ from the YMW16 model[48]) as the mean of a uniform distribution with a generous ±40% width to conservatively represent the uncertainty in $\text{DM}_{MW}$ for the direction of FRB 20190520B. For the MW halo contribution, we use a uniform distribution from 25 pc cm$^{-3}$ to 80 pc cm$^{-3}$. Together, the Milky Way disk and halo components yield a total range from 61 pc cm$^{-3}$ to 164 pc cm$^{-3}$ with a mean of 113 pc cm$^{-3}$ and r.m.s. uncertainty of 17 pc cm$^{-3}$.

For the IGM, we use the $\Lambda$ cold dark matter model ($\Lambda$CDM, where $\Lambda$ is the cosmological constant) to calculate the mean DM contribution[49]

$$\text{DM}_{IGM}(z) = \frac{3cH_0\Omega_b f_{IGM}}{8\pi G m_p}\int_0^z \frac{\chi(z)(1+z)\text{d}z}{[\Omega_m(1+z)^3 + \Omega_\Lambda]^{\frac{1}{2}}}, \qquad (4)$$

where the free electron number per baryon in the Universe is $\chi(z) \approx 7/8$ (which is assumed to be constant for FRB redshifts), the normalized matter density is $\Omega_m = 0.315 \pm 0.007$, the dark-energy fraction is $\Omega_\Lambda \simeq 1 - \Omega_m$, the baryonic fraction is $\Omega_b = h^{-2} \times (0.02237 \pm 0.00015)$, the fraction of baryons in the IGM is $f_{IGM}$, and the Hubble constant is $H_0 \equiv h \times 100$ km s$^{-1}$ = $67.36 \pm 0.54$ km s$^{-1}$ Mpc$^{-1}$ (ref.[47]). Using values for the speed of light $c$, the gravitational constant $G$ and the proton mass $m_p$, the resulting expression

$$\text{DM}_{IGM}(z) \approx 978 \text{ pc cm}^{-3} f_{IGM}\int_0^z \frac{(1+z)\text{d}z}{[\Omega_m(1+z)^3 + \Omega_\Lambda]^{\frac{1}{2}}}, \qquad (5)$$

yields $\text{DM}_{IGM}(0.241) = 248 f_{IGM}$ pc cm$^{-3}$.

Using $\text{DM}_{IGM}(z)$ as the mean, we calculate the range of values for a given redshift using a log-normal distribution with variance $\sigma^2_{\text{DM}_{IGM}}(z) = \text{DM}_{IGM}(z)\text{DM}_c$, where $\text{DM}_c = 50$ pc cm$^{-3}$ is chosen to provide cosmic variance consistent with published simulations. This gives $\sigma_{\text{DM}_{IGM}}(0.241) = 111\sqrt{f_{IGM}}$ pc cm$^{-3}$. The parameters for the log-normal distribution are then $\sigma_{\ln\text{DM}_{IGM}} = \{\ln[1 + (\sigma_{\text{DM}_{IGM}}(z)/\text{DM}_{IGM}(z))^2]\}^{1/2}$ and $\mu_{\ln\text{DM}_{IGM}} = \ln[\text{DM}_{IGM}(z)] - \sigma^2_{\ln\text{DM}_{IGM}}/2$. It is noted that the log-normal distribution is asymmetric. Equation (5) and the log-normal distribution are also used to estimate the median $\text{DM}_{IGM}$ and its inner 68% uncertainty range shown in Fig. 3.

To constrain the host-galaxy DM, we combine the MW and IGM estimates and their uncertainties with the measured DM averaged over all bursts, which is $\text{DM}_{obs} = 1,204.7 \pm 4.0$ pc cm$^{-3}$. The cumulative distribution function (CDF) of the log-normal distribution for the IGM contribution yields the range $\text{DM}_{IGM} = 195^{+110}_{-70}$ pc cm$^{-3}$. We then calculate the posterior distribution of $\text{DM}_{host}$ after marginalizing over the IGM and Milky Way distributions and using a flat, uninformative prior for $\text{DM}_{host}$. The median and probable ranges are calculated from the corresponding CDF. Using $f_{IGM} = 0.85$ for the baryon fraction in the IGM[17,50], we obtain a median value for $\text{DM}_{host}$ and 68% probable interval $\text{DM}_{host} = 903^{+72}_{-111}$ pc cm$^{-3}$. When we vary $f_{IGM}$ from 0.6 to 1, the total range of values for $\text{DM}_{IGM}$ is about 80 pc cm$^{-1}$ to 350 pc cm$^{-1}$ and for $\text{DM}_{host}$ is about 1,020 pc cm$^{-1}$ to 745 pc cm$^{-1}$, where we have taken the median values resulting from each value of $f_{IGM}$ and extended the ranges using (half of the) corresponding 68% probable ranges for the smallest and largest values of $f_{IGM}$.

For comparison, we also give redshift estimates for different values of $f_{IGM}$ if a fixed value of $\text{DM}_{host} = 50$ pc cm$^{-3}$ is used (as is often found in the literature) along with the above quoted mean value for the MW contribution. For $f_{IGM}$ ranging from 0.6 to 1, we obtain a range of redshift values from 2.2 to 0.9 (including the 68% redshift interval derived from the redshift CDF for each value of $f_{IGM}$), much larger than for the redshift of 0.241 for the identified host galaxy.

**Constraints on $\text{DM}_{host}$ from H$\alpha$ measurements.** The DM of the host galaxy is independently estimated from its H$\alpha$ emission by converting the extinction-corrected H$\alpha$ flux, $F_{H\alpha} = (42.0 \pm 0.5) \times 10^{-17}$ erg s$^{-1}$ cm$^{-2}$, to a H$\alpha$ surface density of 224 ± 3 Rayleighs in the source frame at $z = 0.241$, assuming host-galaxy dimensions of 0.5 × 0.5 arcsec (ref.[51]). These host-galaxy dimensions are only a rough estimate based on the size of the H$\alpha$-emitting region in the Keck image, but because the image is seeing limited, the assumed dimensions may be even smaller, which would only serve to increase the H$\alpha$ surface density in the following calculations. The H$\alpha$ flux and estimated surface density are similar to those found for the host galaxy of FRB 20121102A[51]. The H$\alpha$ surface

density in the source frame $S(H\alpha)_s$ is related to the Hα surface density in the observer frame as $S(H\alpha)_s = (1+z)^4 S(H\alpha)_o$, and is related to the emission measure (EM) in the source frame by

$$EM_s = 2.75 \text{ pc cm}^{-6} T_4^{0.9} S(H\alpha)_s$$
$$\approx 616 \pm 7 \text{ pc cm}^{-6} \times T_4^{0.9} \left[ \frac{S(H\alpha)_s}{224 \pm 3 \text{ R}} \right], \tag{6}$$

where we have used the redshift $z = 0.241$ and $T_4$ is the temperature in units of $10^4$ K. The EM is related to the DM in the ionized cloudlet model[51] by $EM = [\zeta(1+\epsilon^2)/f]DM^2/L$, where $\zeta$ represents cloud–cloud variations in the mean density, $\epsilon^2$ represents the variance of density fluctuations in a cloud, $f$ is the filling factor and $L$ is the path length through the gas sampled by the FRB. Using this relation, we obtain the corresponding source frame DM

$$DM_s \approx 392 \pm 3 \text{ pc cm}^{-3} \times T_4^{0.45} \left( \frac{L}{5 \text{ kpc}} \right)^{1/2} \left[ \frac{f/0.1}{\zeta(1+\epsilon^2)/2} \right]^{1/2}$$
$$\left[ \frac{S(H\alpha)_s}{224 \pm 3 \text{ R}} \right]^{1/2}, \tag{7}$$

where we have adopted fiducial values of $f = 0.1$ and $\zeta = \epsilon^2 = 1$, which are typical for the warm ionized medium in the Milky Way. For the path length through the gas, we adopt a fiducial value of $L = 5$ kpc, which is based on the apparent size of the Hα-emitting region in the host galaxy. However, given that the optical image is seeing limited, and that the orientation of the galaxy relative to the FRB line of sight is not known, this path length could be as large as 10 kpc or as small as 0.1 kpc. For a range of $L$ from 0.1 kpc to 10 kpc, we find that $DM_s$ could be as small as 55 pc cm$^{-3}$ or as large as 560 pc cm$^{-3}$, for the same fiducial values of $T_4$, $f$, $\zeta$ and $\epsilon^2$. The estimated DM is also highly sensitive to the temperature: for a range of $T_4$ from 0.5 to 5, $DM_s$ could range from 290 pc cm$^{-3}$ to 810 pc cm$^{-3}$, keeping all other parameters fixed at their fiducial values. In the observer's frame, the measured DM contribution of the host galaxy is smaller by a factor of $1/(1+z)$, yielding a nominal value of $DM_{host,coeff} \approx 316 \pm 2$ pc cm$^{-3}$ for the coefficient in equation (7). The quoted errors account for only measurement errors in the Hα flux. To match the inferred value of $DM_{host} \approx 900$ pc cm$^{-3}$ requires that the three factors in equation (7) involving $T_4$, $L$ and $f$ combine to a factor of about 3, which may easily be explained by a higher temperature or the broad range of possible values for $L$. Regardless, the large Hα EM affirms that the FRB DM receives a significant contribution from ionized gas in the host galaxy, but it is unclear whether the diffuse, Hα-emitting gas can account for the entire host-galaxy contribution or whether the FRB's local environment contains significant ionized gas content that is not seen in Hα.

**Extragalactic scattering.** Scatter broadening manifests as a frequency-dependent temporal asymmetry in the burst dynamic spectrum, which is typically modelled as a one-sided exponential pulse broadening function convolved with a Gaussian pulse. Some bursts from FRB 20190520B have a burst structure that is suggestive of scattering: leading edges that are aligned across the radio frequency band coupled with temporal asymmetries that broaden at lower observing frequencies. However, a number of bursts are symmetric across the radio frequency band and vary significantly in burst width. When the burst S/N is low, it is also difficult to distinguish scattering from other burst substructure, such as the time–frequency drift of intensity islands[9] (often referred to as the 'sad trombone' effect). Examples of bursts with and without evidence of pulse broadening are shown in Extended Data Fig. 7.

To measure the mean burst width and scattering timescale of FRB 20190520B, we first integrate the dynamic spectrum of each burst along the frequency axis and normalize the resulting burst profile. Each burst profile is fit with a model composed of a Gaussian component convolved with a one-sided exponential. Out of the 79 fitted scattering times, all scattering times with fractional uncertainties >50% are excised, leaving a subset of 26 bursts. The scattering times of these 26 bursts are then compared with their dynamic spectra and one-dimensional burst profiles to verify that the burst temporal asymmetry is broadly consistent with a $\nu^{-4}$ frequency scaling. Supplementary Table 1 shows the scattering times re-scaled to 1.25 GHz assuming a $\nu^{-4}$ frequency dependence. The remaining 53 bursts, whose scattering times have fractional uncertainties >50%, are then re-fit with a Gaussian-only model for the one-dimensional burst profile. The full-width at half-maximum burst widths are also shown in Supplementary Table 1. This provisional approach yields a mean scattering time and burst width of 10 ± 2.0 ms and 13.5 ± 1.2 ms at 1.25 GHz, respectively. A more detailed study investigating the burst widths in separate frequency subbands is in progress, and corroborates the scattering interpretation presented here.

The observed mean scattering time of 10 ± 2 ms at 1.25 GHz is probably too small for the FRB's host galaxy to lie behind our proposed host-galaxy association. In this alternative scenario, the host galaxy would lie at a redshift $z_h > 0.241$ that depends on the foreground galaxy's contribution to the total DM budget. The FRB would pass through the putative intervening galaxy at a redshift $z_l > 0.241$ at an impact parameter of about 4 kpc (based on the observed offset of the FRB localization in the optical images). The scattering contribution of the intervening galaxy lens is related to its DM contribution by[52,53]

$$\tau(DM, \nu, z) \approx 48.03 \text{ μs} \times \frac{\widetilde{F} G_{scatt} DM_l^2}{(1+z_l)^3 \nu^4}, \tag{8}$$

where $DM_l$ is the contribution of the intervening galaxy in pc cm$^{-3}$ in its rest frame, $\nu$ is the observing frequency in GHz, $z_l$ is the intervening galaxy redshift, and $\widetilde{F} = \zeta\epsilon^2/f(l_o^2 l_i)^{1/3}$ quantifies the electron density fluctuations in the scattering layer for $\widetilde{F}$ in units of pc$^{-2/3}$ km$^{-1/3}$, where $\zeta$, $\epsilon^2$ and $f$ describe the density fluctuation statistics and filling factor, $l_0$ is the outer scale of turbulence and $l_i$ is the inner scale. The fluctuation parameter $\widetilde{F}$ in the Milky Way varies from ~$10^{-3}$ pc$^{-2/3}$ km$^{-1/3}$ in the thick disk to $\geq 10^2$ pc$^{-2/3}$ km$^{-1/3}$ near the inner Galaxy, and is typically about 0.1–1 pc$^{-2/3}$ km$^{-1/3}$ in the thin disk, but it is not generally known for other galaxies[52]. The geometric factor $G_{scatt}$ is unity for scattering in a host galaxy but it can be very large for an intervening galaxy, in which case $G_{scatt} \approx 2d_{sl}d_{lo}/d_{so} L$, where $d_{sl}$, $d_{so}$ and $d_{so}$ are the angular diameter distances between the source and lens, lens and observer, and source and observer, respectively, and $L$ is the path length through the lens. The numerical pre-factor in equation (8) is for all angular diameter distances in Gpc and $L$ in Mpc.

For the nominal DM contribution implied by the Hα emission, $DM_l \approx 392$ pc cm$^{-3}$, and a path length through the intervening galaxy $L \approx 5$ kpc, the implied scattering time is $\tau \approx 20$ s at 1.25 GHz, orders of magnitude larger than the observed mean scattering time. For a smaller gas temperature $T_4 \approx 0.5$, yielding a DM contribution $DM_l \approx 290$ pc cm$^{-3}$, the implied scattering time is still $\tau \approx 18$ s at 1.25 GHz, assuming $\widetilde{F}$ 0.1 ≈ pc$^{-2/3}$ km$^{-1/3}$ and $L \approx 5$ kpc. However, there is significant latitude in this estimate, depending on the assumed gas temperature, path length sampled by the FRB and the value of $\widetilde{F}$. Although we have assumed that the FRB traces the entire $DM_l$ implied by the Hα emission, the FRB line of sight probably only traces a fraction of the Hα-emitting gas. If $DM_l$ is as small as 50 pc cm$^{-3}$, then $\tau$ could be as small as 0.6 s for $\widetilde{F}$ 0.1 ≈ pc$^{-2/3}$ km$^{-1/3}$ and $L \approx 5$ kpc. Although the implied scattering time could be two orders of magnitude smaller or larger depending on the combination of $\widetilde{F}$, the path length $L$ and the parameters used to estimate the Hα DM, our fiducial estimates suggest that the observed scattering time is significantly smaller than the scattering expected from an intervening galaxy.

**Rotation measure search from FAST observations.** A search for RM was performed with the FAST data. The polarization was calibrated by correcting for differential gains and phases between the receptors through separate measurements of a noise diode injected at an angle of 45° from the linear receptors. We searched for the RM from $-30 \times 10^5$ rad m$^{-2}$ to $3.0 \times 10^5$ rad m$^{-2}$. No significant peak was found in the Faraday spectrum. The observed lack of polarization could be due to the intrinsic low linear polarization, a depolarization process within the source or from intra-channel Faraday rotation.

## Data availability

The FAST data are available at https://doi.org/10.11922/sciencedb. o00069.00004. The VLA data can be accessed at https://doi. org/10.7910/DVN/C5CEEI.

## Code availability

The data analysis code used for FAST observations is available at https:// github.com/peterniuzai/FRB190520B_discovery_paper_Code_avail-ability.git. For code relevant to the analysis of the VLA observations, see https://github.com/realfastvla/rfpipe. The publicly available packages CASA, Heimdall and DM_phase can be found on their respective websites.

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

**Acknowledgements** This work was supported by National Natural Science Foundation of China (NSFC)programme numbers 11988101, 11725313, 12041303, U1731238, U2031117, U1831131 and U1831207, and by the Cultivation Project for FAST Scientific Payoff and Research Achievement of CAMS-CAS. C.-H.N. acknowledges support from the FAST Fellowship. S.C., J.M.C. and S.O. acknowledge support from the National Science Foundation (AAG 1815242) and are members of the NANOGrav Physics Frontiers Center, which is supported by NSF award PHY-1430284. C.J.L. acknowledges support from the National Science Foundation under grant number 2022546. K.A., S.B.-S. and R.A.-T. acknowledge support from NSF grant AAG-1714897. P.W. thanks the YSBR-006, CAS Project for Young Scientists in Basic Research. S.B.-S. is a CIFAR Azrieli Global Scholar in the Gravity and the Extreme Universe programme. Y.N. acknowledges support from JSPS KAKENHI grant number JP20H01942. J.-A.J. acknowledges support from the Japan Society for the Promotion of Science (JSPS) KAKENHI grant JP19K23456 and JP18J12714. S.D. is the recipient of an Australian Research Council Discovery Early Career Award (DE210101738) funded by the Australian Government. We thank the FAST key science project for supporting follow-up observations; and the FAST collaborations and realfast team for their technical support. Some data presented herein were obtained at the W. M. Keck Observatory, which is operated as a scientific partnership among the California Institute of Technology, the University of California and the National Aeronautics and Space Administration. The observatory was made possible by the generous financial support of the W. M. Keck Foundation. This study is based in part on data collected at the Subaru Telescope, which is operated by the National Astronomical Observatory of Japan. The National Radio Astronomy Observatory is a facility of the National Science Foundation operated under cooperative agreement by Associated Universities, Inc. This work was supported by the China Science and Technology Cloud (CSTCloud) and China Environment for Network Innovations (CENI). We thank the staff of CSTCloud/CENI for their support during data processing.

**Author contributions** C.H.N. discovered the source FRB 20190520B. D.L., C.J.L., S.C., C.-W.T., W.Y. and C.-H.N. initiated the follow-up projects. D.L., C.-H.N., B.Z. and W.-W.Z. led the follow-up FAST observations. C.-H.N., J.-M.Y., Y.-K.Z., P.W., D.-J.Z. and Y.F. searched and processed the FAST data. K.A., C.J.L., S.C., X.Z., S.B.-S., Z.Y., W.Y. and L.C. contributed to the VLA burst detection and localization, identification and measurements of the associated persistent radio source. C.-W.T., S.C., D.S., Y.N., J.-A.J., C.B. and G.-D.L. contributed to the optical/NIR follow-up observations and analysis. S.K.O., J.M.C., S.C., J.-M.Y. and C.-H.N. measured the burst scattering and analysed the propagation effects. J.M.C., B.Z. and W.-Y.W. contributed to the DM$_{host}$ estimation. K.A., Y.-K.Z., J.-R.N., R.L., W.-W.Z. and C.-H.N. contributed to the periodicity and burst rate analysis. P.W. and Y.-K.Z. helped with energy calibration and M.Y. contributed to the radio frequency interference removal on FAST data. Z.Y., W.Y., M.C., S.D. and Y.-L.Y. contributed to other follow-up observations. S.B.-S., D.L., K.A., C.-H.N., S.C., J.M.C., S.K.O. and C.J.L. had major contributions to the preparation of the manuscript. All of the authors have reviewed, discussed and commented on the presented results and the manuscript.

**Competing interests** The authors declare no competing interests.

**Additional information**
**Correspondence and requests for materials** should be addressed to D. Li, W. Yu or C. J. Law.

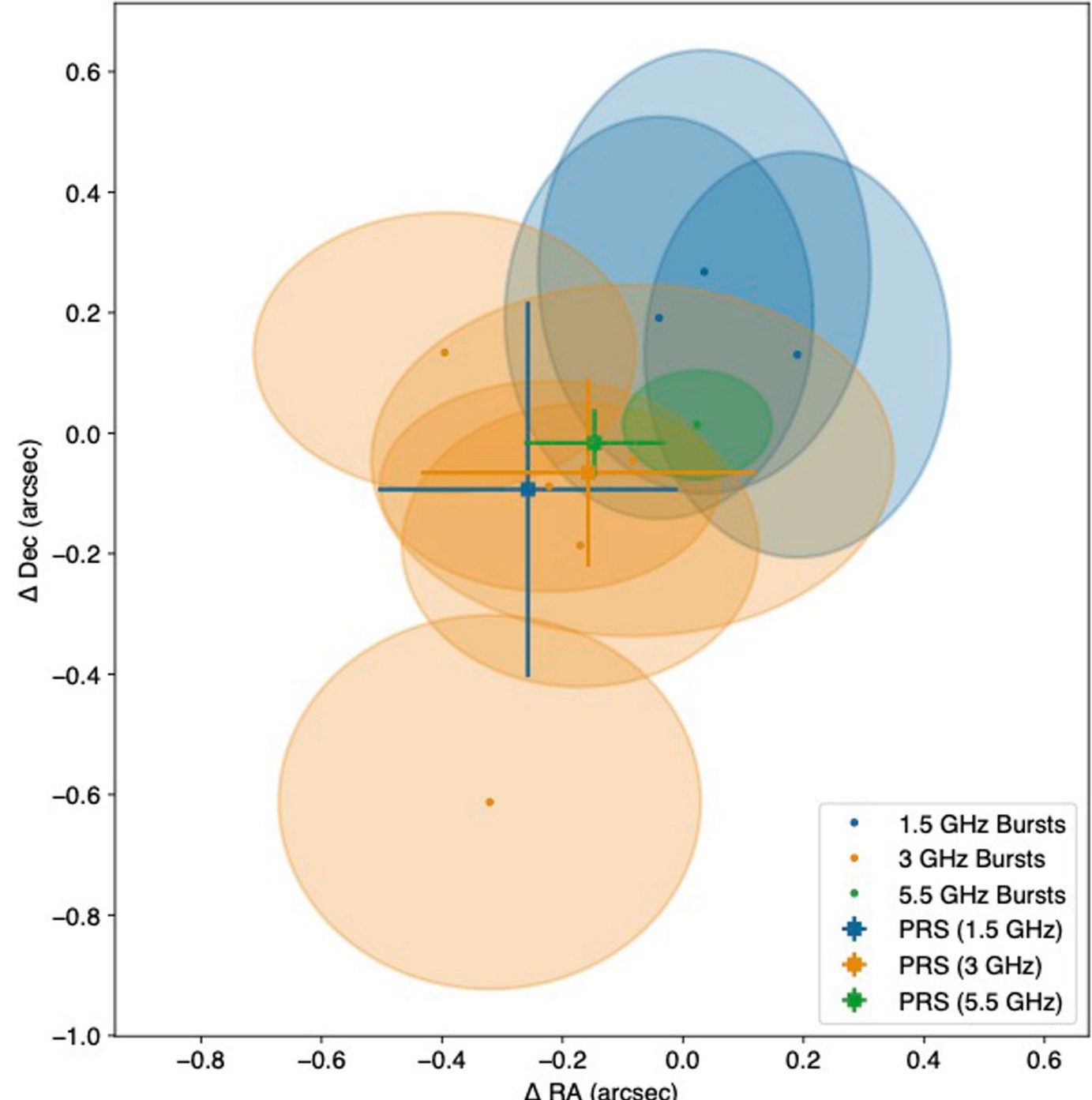

**Extended Data Fig. 1 | Positions of the bursts and persistent radio source identified with VLA observations.** The observation were performed at 1.5, 3, and 5.5 GHz. They were shown as offsets from the best-fit position of the ensemble of bursts, at RA = 16h02m04.272s, dec. = −11° 17′ 17.32″ (J2000). The uncertainties on the positions of the bursts are indicated with shaded ellipses, and those for the PRS are shown with error bars. These uncertainties include $1\sigma$ statistical errors and estimates for systematic errors added in quadrature.

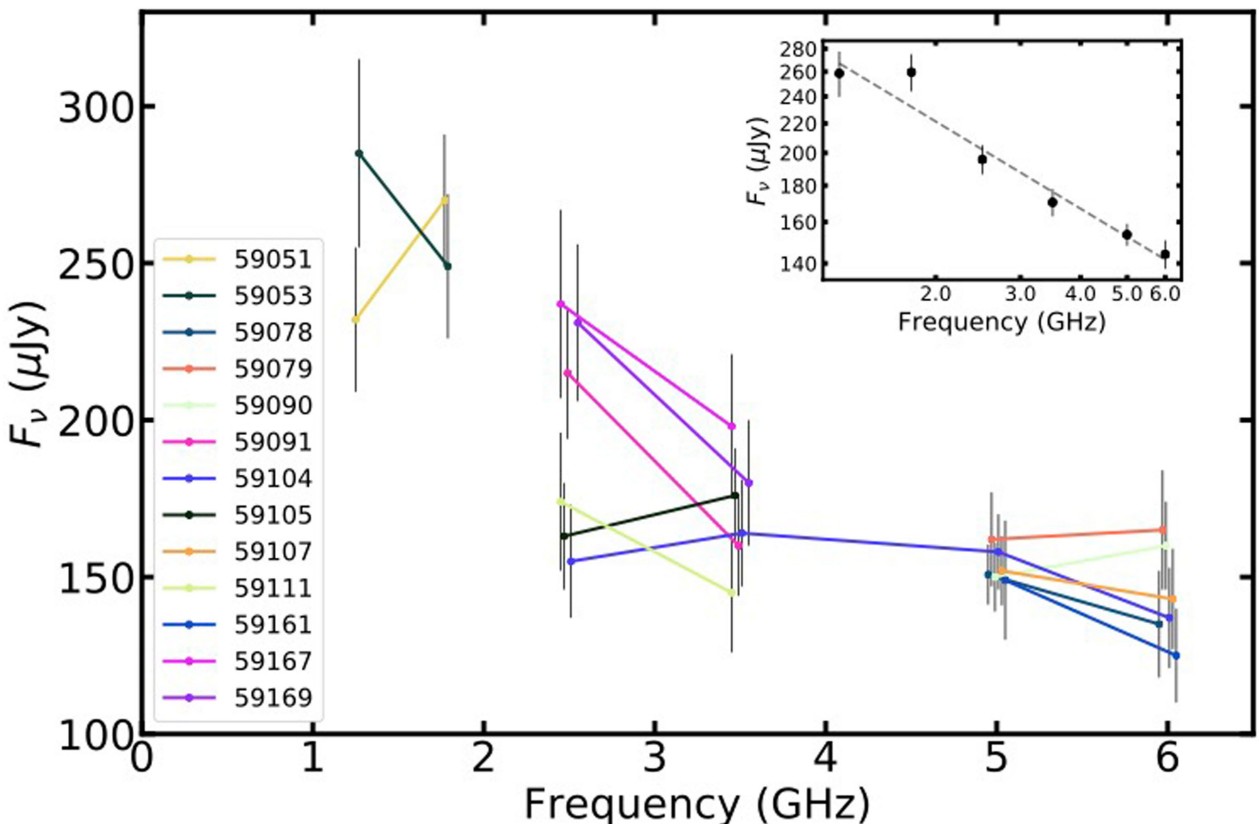

**Extended Data Fig. 2 | Spectrum of the PRS associated with FRB 20190520B.** The bandwidth is split into two subbands for all the observations and the corresponding radio flux densities are shown. The observation dates in MJD are shown in the legend. The frequencies for each subband are shifted slightly in order to show the flux density error bars. The inset plot shows the average flux at each subband. A power-law model fit to these measurements is shown by the black dashed line and yields a spectral index of −0.41± 0.04 for the PRS.

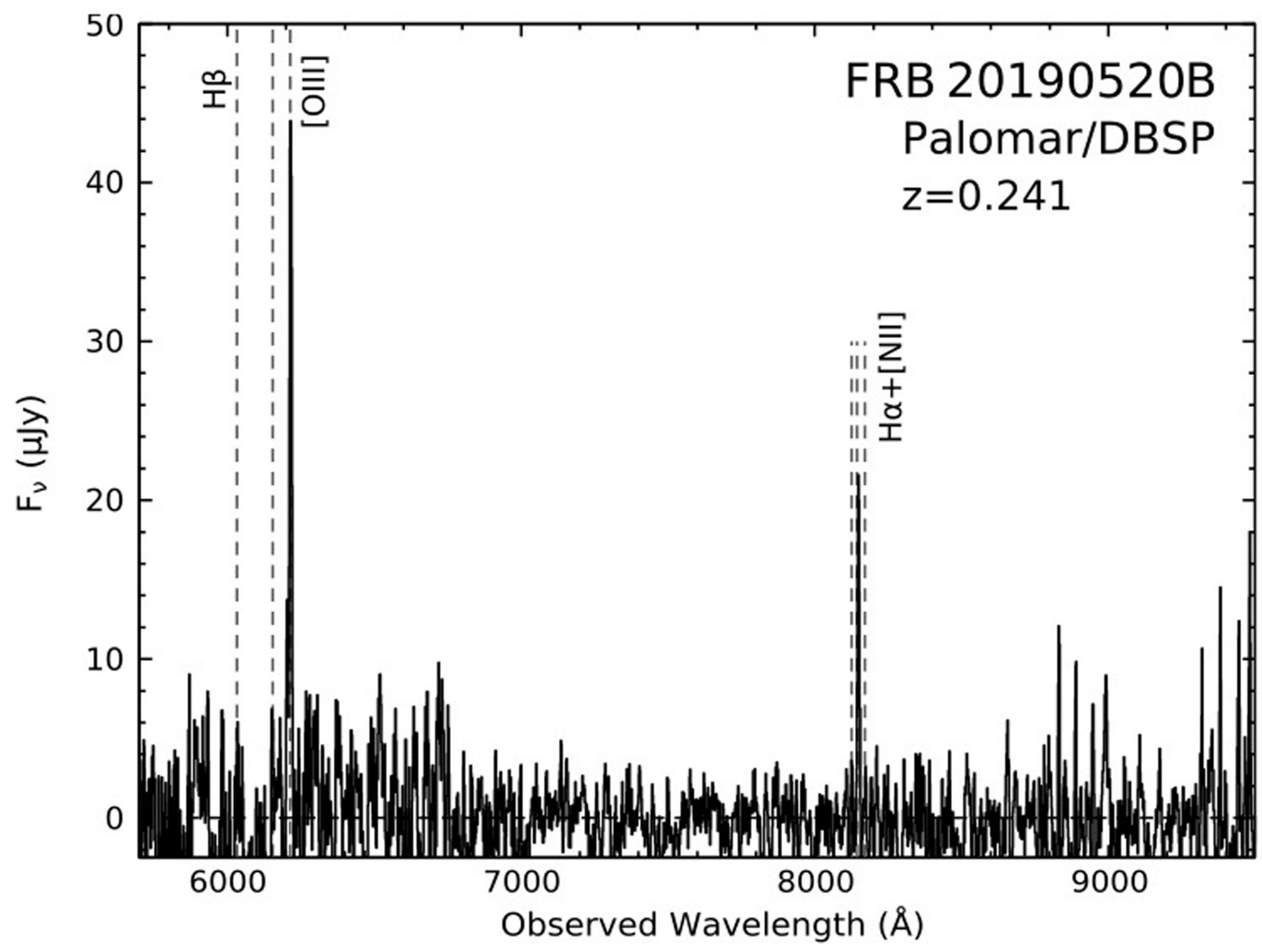

**Extended Data Fig. 3 | Optical spectrum of the FRB 20190520B host galaxy obtained at Palomar.** The redshift of $z$ = 0.241 was determined using the [OIII]-5007Å line and H$\alpha$ line (>5$\sigma$ detections). These two emission lines are narrow, with FWHM approximately 10 Å. The flux scale of the spectrum is not corrected for slit loss or extinction.

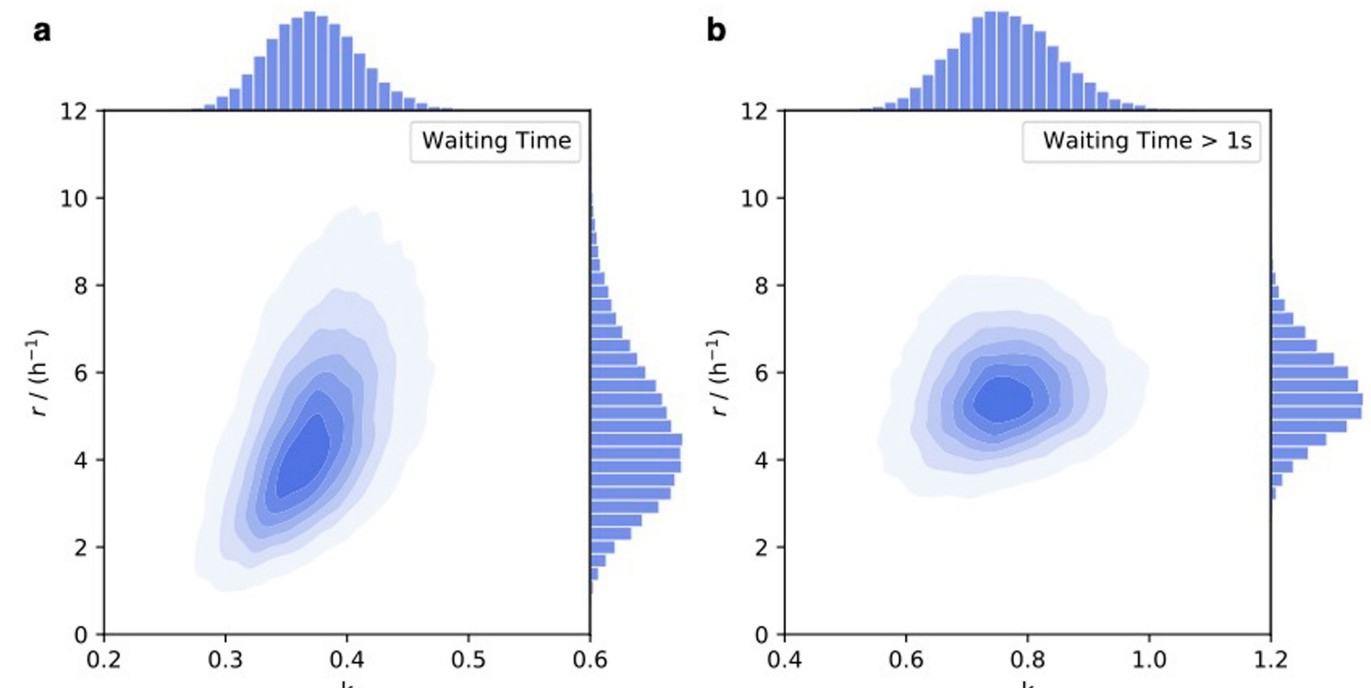

**Extended Data Fig. 4 | Posterior probability distributions.** The probability distribution are for the shape parameter $k$ and event rate $r$ of the Weibull distribution. **a**, Results from a fit to all burst waiting times of FRB 20190520B detected by FAST. **b**, Results from a fit to all burst waiting times longer than 1 s. These bursts are at fluences higher than 9 mJy ms.

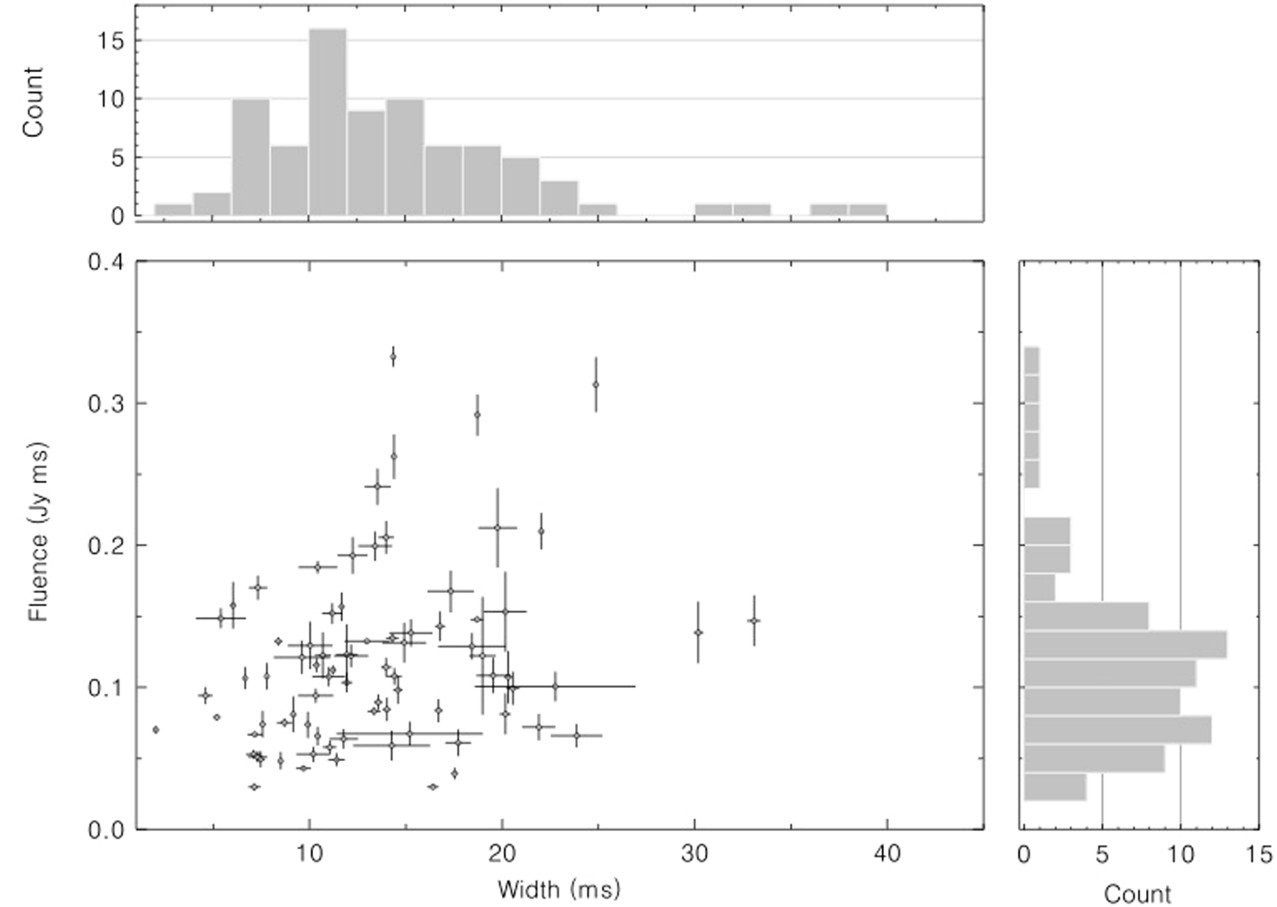

**Extended Data Fig. 5 | The fluence-width distribution at 1.25 GHz for FRB 20190520B.** The widths and fluences of the 79 bursts detected by FAST are also shown in Supplementary Table 1.

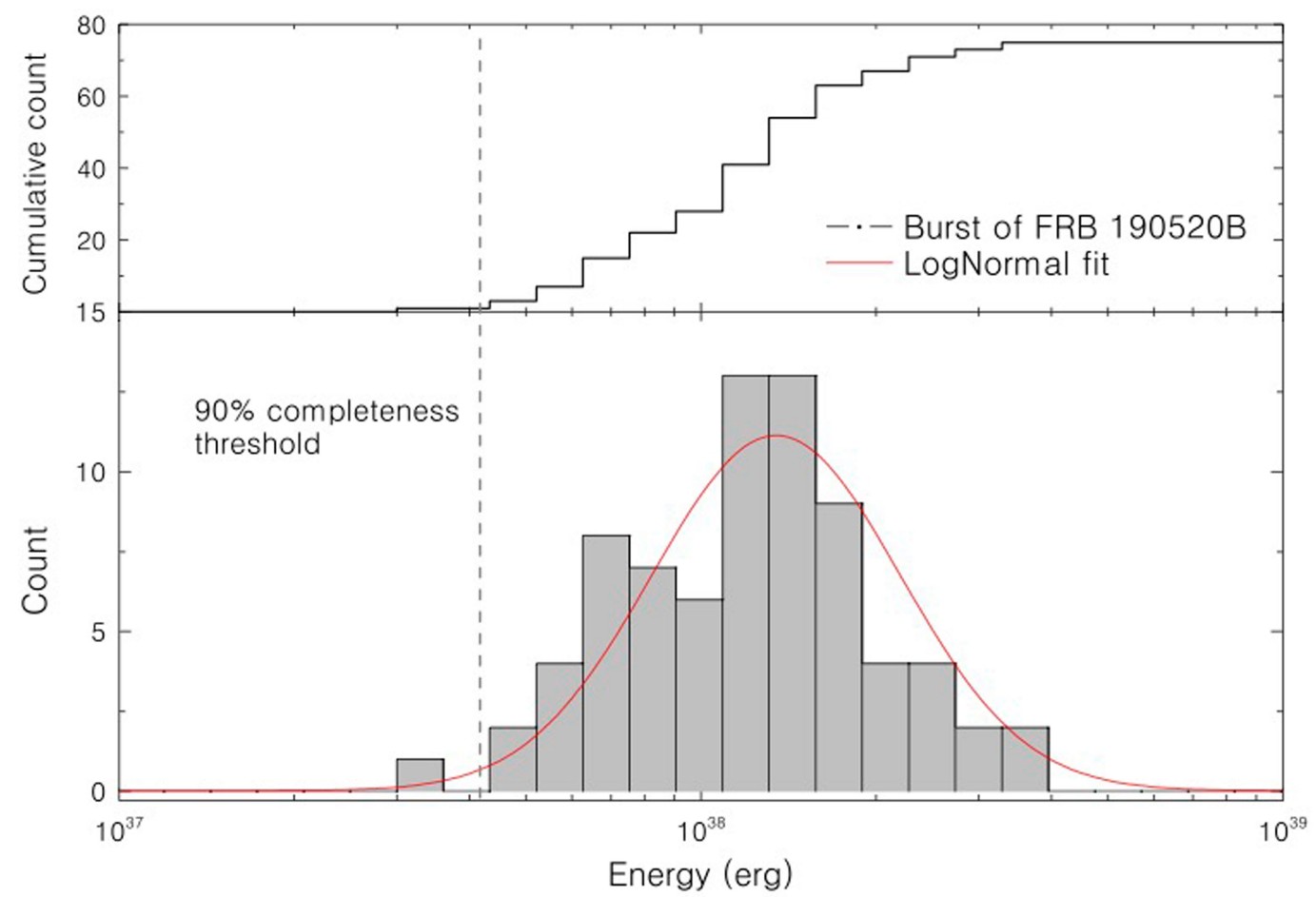

**Extended Data Fig. 6 | The distribution of burst energies for FRB 20190520B.** The bursts are from FAST observations. The red line and black dashed line are the log-normal fit and 90% completeness threshold, respectively. The 90% completeness threshold uses 0.023 Jy ms, which is the simulation result from ref. [27].

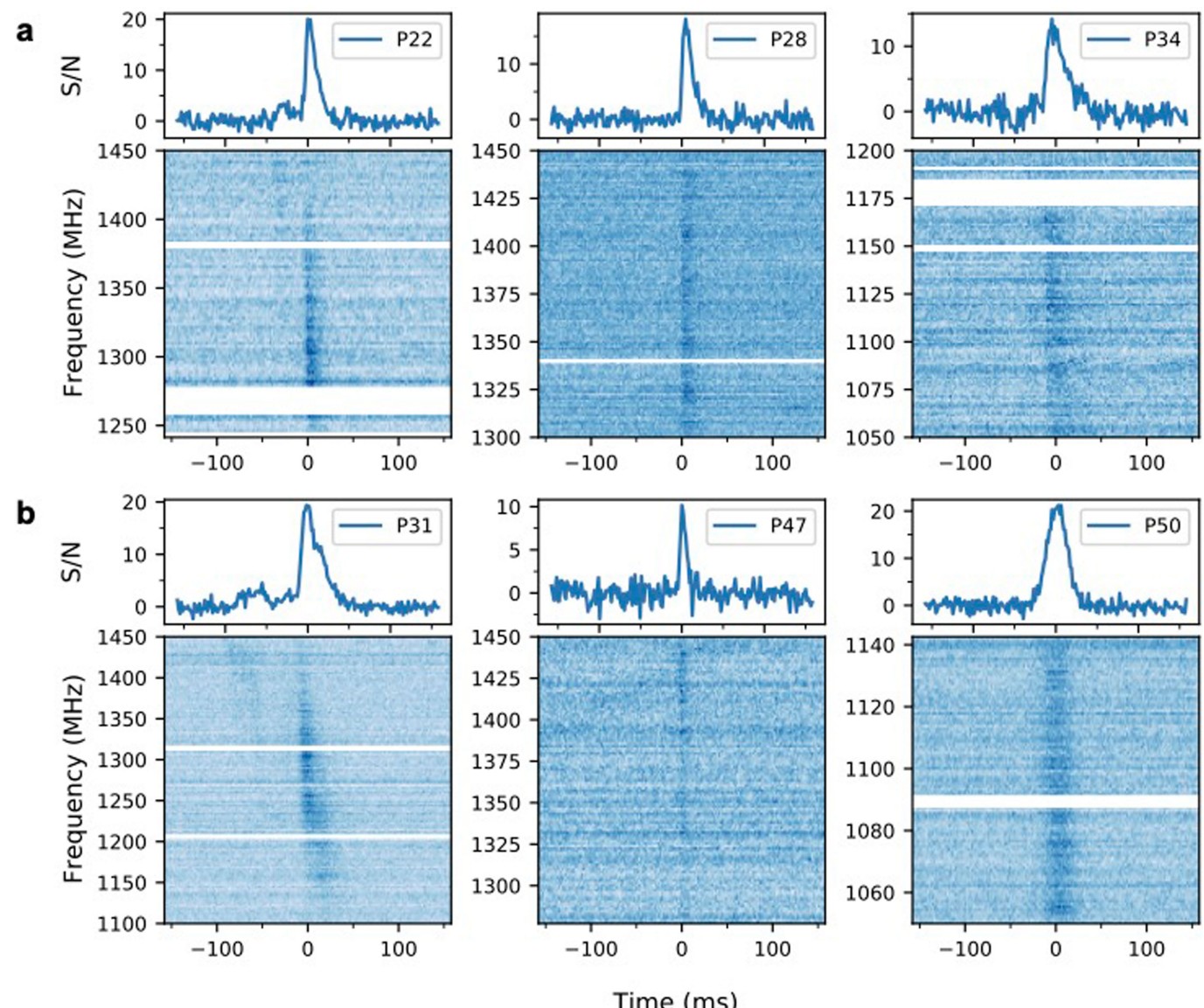

**Extended Data Fig. 7 | Dynamic spectra of bursts with and without scattering. a**, Frequency-time dynamic spectra of bursts with significant evidence of pulse broadening, along with 1D burst profiles that have been averaged across the entire frequency band. **b**, Examples of bursts without significant evidence of pulse broadening. White lines indicate excised radio frequency interference. All 1D burst profiles are shown in units of the signal-to-noise ratio (S/N). For P31, there is a potential combination of scattering, time–frequency drift, and multiple unresolved burst components; in this case we report a scattering time that should be interpreted as an upper limit. Bursts with scattering time constraints are shown in Supplementary Table 1.

**Extended Data Table 1 | VLA observations of the PRS and number of bursts detected in each observation**

| Start Time (MJD) | Frequency (GHz) | Duration (min) | Beam size (",") | Flux density ($\mu$Jy) | Bursts |
|---|---|---|---|---|---|
| 59051.033815 | 1.5 | 96 | 4.78×2.90 | 258±16 | 1 |
| 59053.046790 | 1.5 | 96 | 4.63×2.90 | 273±23 | 2 |
| 59079.004955 | 5.5 | 41 | 1.38×1.02 | 145±10 | 0 |
| 59079.971533 | 5.5 | 41 | 1.44×1.05 | 164±11 | 0 |
| 59090.941498 | 5.5 | 41 | 1.45×1.04 | 158±9 | 0 |
| 59091.938768 | 3 | 41 | 2.43×1.85 | 195±14 | 5 |
| 59104.867225 | 3 | 41 | 2.76×1.76 | 160±13 | 0 |
| 59104.908789 | 5.5 | 41 | 1.43×1.04 | 151±9 | 0 |
| 59105.976837 | 3 | 41 | 2.44×1.60 | 186±15 | 0 |
| 59107.915546 | 5.5 | 41 | 1.38×1.02 | 153±10 | 1 |
| 59111.116215 | 3 | 41 | 5.00×1.45 | 176±16 | 0 |
| 59161.677240 | 5.5 | 41 | 1.92×0.47 | 139±13 | 0 |
| 59167.637761 | 3 | 41 | 2.76×0.51 | 233±17 | 0 |
| 59169.640486 | 3 | 41 | 2.47×0.48 | 211±14 | - |

Frequency refers to the centre of the observing band and duration represents the total length of on-source observations. Note that the realfast system was not run on the last observation owing to a system error.

**Extended Data Table 2 | Localized positions of the PRS from 1.5, 3, 5.5 GHz VLA deep images**

| Frequency (GHz) | R.A. | Decl. | Statistical Error | | Systematic Offsets | |
|---|---|---|---|---|---|---|
| | | | R.A. ('') | Decl. ('') | R.A. ('') | Decl. ('') |
| 1.5 | 16h02m04.2543s | −11d17m17.4146s | 0.0386 | 0.0713 | 0.0373±0.2460 | 0.0144±0.3035 |
| 3 | 16h02m04.2611s | −11d17m17.3869s | 0.0176 | 0.0168 | −0.0696±0.2772 | 0.0382±0.1552 |
| 5.5 | 16h02m04.2618s | −11d17m17.3375s | 0.0075 | 0.0095 | −0.0947±0.1161 | 0.1069±0.0554 |

The first column shows the observing central frequency. The second and third columns report the coordinates of the PRS from the deep images. The remaining columns show the 1σ statistical errors and the cross-matched systematic offsets in right ascension and declination.

**Extended Data Table 3 | Localized positions of the VLA bursts**

| Band | S/N | R.A. | Decl. | Statistical Error | | Systematic Error | |
|------|-----|------|-------|----------|----------|----------|----------|
| | | | | R.A. (″) | Decl. (″) | R.A. (″) | Decl. (″) |
| L1 | 16 | 16h02m04.2742s | −11d17m17.0535s | 0.124 | 0.208 | 0.2460 | 0.3035 |
| L2 | 27 | 16h02m04.2847s | −11d17m17.1912s | 0.060 | 0.145 | 0.2460 | 0.3035 |
| L3 | 15 | 16h02m04.2691s | −11d17m17.1299s | 0.070 | 0.139 | 0.2460 | 0.3035 |
| S1 | 19 | 16h02m04.2567s | −11d17m17.4093s | 0.044 | 0.080 | 0.2772 | 0.1552 |
| S2 | 17 | 16h02m04.2449s | −11d17m17.1874s | 0.153 | 0.173 | 0.2772 | 0.1552 |
| S3 | 15 | 16h02m04.2602s | −11d17m17.5078s | 0.103 | 0.176 | 0.2772 | 0.1552 |
| S4 | 10 | 16h02m04.2661s | −11d17m17.3666s | 0.333 | 0.247 | 0.2772 | 0.1552 |
| S5 | 7 | 16h02m04.2500s | −11d17m17.9337s | 0.214 | 0.269 | 0.2772 | 0.1552 |
| C1 | 19 | 16h02m04.2734s | −11d17m17.3078s | 0.041 | 0.071 | 0.1161 | 0.0554 |

The label in the first column shows the frequency band of observation, followed by the burst number. DM represents the S/N maximizing DM obtained using offline refinement of the bursts and S/N reports the maximum image S/N. The remaining columns show the burst positions (and errors) obtained using CASA calibration and image fitting. The systematic errors reported here were estimated using deep radio images generated using all the observations at the respective frequency band.

**Extended Data Table 4 | Properties of the VLA bursts**

| Name | MJD | DM | Flux density (Jy) | Width (ms) | Frequency (GHz) |
|------|-----|-----|-------------------|------------|-----------------|
| L1 | 59051.0476715 | 1209.60 | 0.113±0.006 | 20 | 1.33–1.52 |
| L2 | 59053.1045803 | 1227.15 | 0.275±0.010 | 20 | 1.39–1.52 |
| L3 | 59053.0973272 | 1236.60 | 0.189±0.009 | 10 | 1.78–1.97 |
| S1 | 59091.9604318 | 1291.30 | 0.075±0.004 | 20 | 3.12–3.50 |
| S2 | 59091.9560792 | 1193.40 | 0.100±0.009 | 20 | 2.63–2.88 |
| S3 | 59091.9635008 | 1222.75 | 0.093±0.007 | 20 | 2.50–2.76 |
| S4 | 59091.9435287 | 1216.80 | 0.037±0.005 | 10 | 2.88–3.24 |
| S5 | 59091.9668476 | 1276.50 | 0.022±0.003 | 10 | 2.50–3.50 |
| C1 | 59107.9272138 | 1267.50 | 0.068±0.004 | 10 | 4.49–4.88 |

The label in the first column shows the frequency band of observation, followed by the burst number. MJD is referenced to the solar system barycentre (in the TDB scale) and corrected for dispersion delay at infinite frequency. Width values here should be considered as upper limits. Frequency represents the range of frequencies in which the burst signal is prominently seen, and was visually determined. The DM that maximizes the S/N of the respective burst is given in the third column. The apparent DMs may reflect time–frequency structure, as opposed to bona fide DM variations.