## [Peer Review File · Nature]

Manuscript Title: A repeating fast radio burst in a dense environment with a compact persistent radio source

Reviewer Comments & Author Rebuttals

Reviewer Reports on the Initial Version:

Referees' comments:

Referee #1 (Remarks to the Author):

The authors have presented a study of the newly discovered and highly active repeating fast radio burst FRB 190520. This study has shown a number of remarkable features that are best explained by a dense and potentially highly magnetized ionized plasma surrounding the FRB source, including a high excess dispersion measure, a co-located continuum radio source, and a non-detection of linear polarization that could be due a high rotation measure. Along with the first repeating FRB121102, which is also a highly active source and which also possesses similar characteristics, this this suggests that FRB activity levels may be correlated with their local environment. Whether this hints at a distinct source population or a different evolutionary phase compared to the bulk of the FRB population, this is a highly significant result and well worthy of publication. The impact of large local contributions to the FRB DM on the use of FRBs as probes of intergalactic baryons is also important, as noted, although the manuscript does not specifically highlight that if FRB repeat rate is correlated with local environmental properties, this may offer an opportunity to calibrate local contributions to the FRB DM, or at least identify potentially problematic sources for exclusion from IGM studies. Overall, the original and significant results presented in this manuscript will certainly be of interest to the large and growing community of FRB researchers and those that hope to use FRBs as tools to study the Universe.

I have identified a number of places in the manuscript where additional information is warranted or other improvements are required to aid understanding for the reader. In some cases, this may require additional analysis, or else the authors should justify in the text why this additional analysis has not been undertaken. I have identified these in a (rather lengthy) numbered list below, which also includes some comments on the presentation of uncertainties and other statistical matters.

Before I enumerate specific suggestions and criticisms, though, I will comment on the clarity, context, and conclusions of the paper as a whole. First, I should note that there are very many minor and not-so-minor typographical errors throughout the paper: missing articles, plural/singular disagreement, italicised units from the use of math mode in latex, symbols not defined, incorrect numbers of significant digits, etc. I have highlighted some of them in my enumerated list below but this list is definitely not exhaustive – I suggest the authors give the entire manuscript another, much more thorough proof-reading. These issues were especially prevalent in the Methods section.

A second important point regarding clarity refers to the usage of the terms “Persistent Radio Source” and “persistent radio emission. I would encourage the authors to consider carefully their use of

“PRS” vs “persistent radio emission”; the former is used to imply compact emission co-located with the FRB itself, while the latter could be emission associated with other host galaxy properties such as star formation that is not directly associated with the FRB progenitor. Many FRB papers are guilty of muddying the waters here, but I would encourage a cleaner separation between the two, perhaps by making it clear that PRS refers to something definitely associated with the local FRB environment, using an alternative such as “continuum radio emission” when discussing galaxy-wide radio emission, and using “potential PRS” when continuum radio emission may or may not be associated with the local FRB environment.

Final comment on clarity: I encourage the authors to consider the use of TNS naming designations for the FRBs. I understand that the historical, non-TNS name for FRB121102 is in widespread use, but the field seems to be moving towards adoption of TNS names, and the use of the non-TNS designator “201124A” for FRB 20201124A sticks out.

In terms of conclusions, the primary conclusion of the paper, that FRB repeat activity level may be correlated with local source environment, is well supported. As I noted in the first paragraph of the review, I think that a logical extension of this connection is that outlier sources that might otherwise bias the Macquart relation between DM and redshift may be identifiable (and potentially even calibrate-able.). It would seem logical to comment on this directly in the manuscript. I note that the final sentence of the main text, “Either possibility has direct impact on calibrating the FRB DM as a probe of cosmic baryons and on understanding FRB origin(s)”, does touch on this point, but at no stage in the text was the link made to *how* this calibration would be done.

In the second last sentence of the conclusion, however, I disagree with the statement that more detections will distinguish between the two scenarios presented in the near future. It is certainly plausible that a correlation may be found between burst repetition rate and the presence of a PRS and/or a large DM_{host}, and that this may be related to source age, as we have already established. But I don’t see how this can then be connected to apparently non-repeating sources. Most *repeaters* already don’t have a detectable PRS, and less active repeaters already have unremarkable host DM contributions. So any such correlation between repetition rate and PRS properties, if it exists, would presumably become undetectable (since the PRS will not be detectable, and any excess contribution to DM will become negligible) well before you get to the apparently non-repeating FRBs that form the bulk of the FRB population. So I believe that this statement needs adjustment or additional support.

Final overarching comment on the manuscript: I am not sure why so much space is dedicated to the exclusion (based on scattering) that the host is a chance alignment in the main text. This can be adequately covered in the Methods and simply referred to, as it is not of general interest. On the other hand, the extremely important result that the RM may be very high (like 121102) is referred to nowhere in the main text, despite this being (if confirmed) an extremely important additional link between the two sources in terms of their local environment. If the results of the analysis are not referred to in the main text, why is this in the Methods?

Enumerated comments in order of appearance in the text:

1. In the abstract, a number of quantities are used without context or insufficiently defined. Fast Radio Bursts themselves are introduced with no context. The term “repeater” is introduced without context, as it “persistent radio source” (noting my earlier point about defining PRS precisely). “The estimated host galaxy contribution DM_{host}” is similarly imprecise: at this stage, DM (Dispersion Measure) has not been defined, so it would not be clear to most readers what this “contribution” is. Grammatical errors abound: Fast Radio Bursts (FRB) should have “FRBs” within the parentheses, line 44 should read “...to be associated with **a** persistent radio source”, no space before “Here” on line 45, line 51/52 should read “...with **a** confirmed association between **an** FRB and **a** compact PRS”, etc.
2. Line 68: using an uncertainty in arcseconds for right ascension right next to the position (in seconds) is rather confusing. I suggest that either giving the uncertainty in seconds, or quoting the full position (RA+Dec) followed by an uncertainty ellipse in arcseconds, would be less prone to misinterpretation. Moreover, it is stated that the uncertainty is dominated by systematic considerations, but this is never shown for the FRB in the Methods (it is shown for the PRS). Finally, I disagree with the uncertainty presented for the FRB – how can the systematic uncertainty on a position which is obtained from a handful of milliseconds of exposure time have a smaller systematic uncertainty than the PRS, which has hours of exposure time? In a best case, they would be identical, but as I discuss in the methods comments below, I think the systematic contribution to the FRB position uncertainty has been given insufficient attention.
3. Line 71: too many significant digits in the Declination position, and too many significant digits in both RA and Decl. uncertainty.
4. Lines 78-80: While the association between the galaxy and the FRB is quite secure, a chance probability calculation is a cruder measure than a Bayesian inference (e.g., Aggarwal+2021) that also takes into account other nearby galaxies and their properties. The authors may wish to consider evaluating the probability of association using a Bayesian method such as the one indicated.
5. Line 87/88 – both R and R’ band are referred to. Which was used? I note that R’ and R are again both used later in the Methods section. Line 89: How is that Halpha luminosity derived? It is not derived anywhere in the Methods.
6. Line 101: There is no reference to the Methods section where the DM_IGM number is calculated. I also am unable to reproduce the reasoning that yields the estimated value and range for DM_IGM, but we shall return to that in due course.
7. Line 107: “The measured DM and scattering properties exclude” – need to say what is being referred to here after “properties” (i.e., “of the FRB”)
8. In this same paragraph, the predicted scattering time in the event of a chance background source is only two orders of magnitude greater than observed. In the Milky Way, scattering times show a very large degree of variability. I would suggest that the (lack of) scattering on its own is not conclusive evidence to rule out a background source, but combined with the low P_{chance} of the host galaxy association, it further strengthens the already very strong case.
9. Line 126: Is it not worth also commenting on the similarities in terms of an apparent star formation peak at the location of the FRB and PRS for both 121102 (Bassa+2017) and 190520?
10. Line 139/140: DM_{host} can also be observationally biased, as it contributes to an observational quantity (the total DM) to which different FRB searches have different sensitivities.
11. Line 142: To be fair, this sentence should note that not all FRBs have comparably deep limits on

the presence of a PRS.

12. Figure 2: I think that either a zoom in (for one of the bands) and/or some annotations of other nearby sources would be valuable. This goes to point 4 above – are any of the other relatively nearby sources larger galaxies that may have a reasonable likelihood of harboring an FRB in their outskirts?

13. Figure 3: why is there no uncertainty on the MW contribution to DM from NE2001? The authors could consider the difference between NE2001 and YMW16 as a crude estimate of the uncertainty. No reference is given for the MW halo contribution and uncertainty, and likewise no reference is given for the relationship used for the expected median extragalactic DM contribution or its uncertainty.

14. Line 283: Why would some bursts show scattering and others not? Surely the simplest assumption is that the scattering is time independent, which would enable this parameter to be fixed across all bursts and constrained with good precision, which would then lead to a better precision on the (deconvolved) gaussian burst widths for all bursts? Non-gaussian intrinsic structure will surely complicate the analysis, but it will also be complicating the present analysis (where the choice of scattering vs no scattering is apparently made by eye) already, and the assumption of constant scattering is at least physically justifiable.

15. Line 284: what is “the sub-pulse”? presumably “a sub pulse”, but this needs to be much more carefully defined. What level of significance is used to define a sub-burst that should be fitted (currently “the noise baseline” is stated)? At what width?

16. Line 298: why is Heimdall (searching) described after the burst modelling? This is backwards, surely it would make more sense to introduce the burst detection prior to the burst modelling.

17. Line 329: How is FETCH applied to imaging data, which has multiple spatial pixels, each of which contains time-frequency data? Is the time-frequency data from a single pixel extracted and fed into FETCH? Further clarification is needed (the supplied references do not explain.)

18. Line 334: More details of the refined imaging of candidates is appropriate. What is the pixel size used (and how does this differ to the real time search)? What is the step size of the DM grid? What RFI flagging parameters are changed? All these details should be presented. The supplied reference (Section 2.4 of Law+2020) does not go into these details.

19. Line 348: The description of the impact of not performing deconvolution are somewhat imprecisely described in the first paragraph of Section 2. For a point source, deconvolution should not normally affect the peak of the map substantially (dirty vs clean). What it does is remove the PSF shape from the surrounding pixels and replace it with a smoother shape. After all, once a source has been deconvolved, the final image is obtained by convolving the model with a gaussian approximation of the PSF before added it to the residual, but this does not “spread the signal” out – at least, not in a significantly different way to way the PSF does. What deconvolution does do is enable a higher S/N when performing a gaussian fit in the image plane, since the weighted average of all the pixels that contain some signal are used, rather than only looking at the peak pixel. This is particularly helpful when the PSF has only been sparsely sampled (few pixels per beam) and the source lies near a pixel boundary; in this case, the S/N can be a bit higher than simply the peak pixel value divided by the off-source rms. All that was a long way of saying: the final sentence of this paragraph should be revised.

20. Line 351: How is the time selection of the de-dispersed visibilities achieved? Is any weighting applied based on the fitted width of the burst, or is a top-hat selection function applied?

21. Line 364-372: it is naive to take the statistical image-plane fit uncertainty as the sole source of uncertainty for an individual burst, as there will surely be calibration errors leading to systematic

position uncertainties. This likelihood should be acknowledged and estimated (the recent papers on FRB20201124A mostly include reasonable discussions of systematic uncertainties on position estimates). It is then even more naïve to take a simple weighted mean of all of the burst positions (with statistical-only uncertainties) estimated to arrive at a final FRB position and uncertainty. An estimate of systematic uncertainty in the VLA position must be presented at the same time as that final statistical uncertainty.

22. Line 391-393: Why was the PRS fit only with a point source model? If the source size is consistent with being unresolved then this is reasonable, but surely a gaussian fit should also be attempted to place an upper limit on size (at each frequency)?

23. Line 410: Which direction are the systematic offsets? VLA-PanSTARRS or vice versa? Has the correction been applied to the final PRS position that was reported, or not?

24. Line 433: Why is 0.1" considered a conservative estimate to the systematic uncertainty, when the uncertainty in the VLA-PanSTARRS offset is >0.12" at all frequencies? (And indeed, 0.12" is used as the systematic uncertainty in the PRS position in the main text?)

25. Line 438: It is odd that this section is titled "Optical Redshift Determination" when it also includes a lot of photometry.

26. Line 442: The spectral range of the J band observations is not given.

27. Line 452: "The slit of DBSP was set to cover the FRB optical counterpart" – but the preceding text says that only PanSTARRS was used, and subsequent text says that the host galaxy is not seen in PanSTARRS. What optical counterpart is thus being referred to here?

28. End of the same sentence, line 454-455: "which later is found to coincide with the location of the pulsation emission from FRB 190520 in 0.18": this sentence needs revising, as the grammar is incorrect and it is not exactly clear what the authors mean.

29. Equation 1 and surrounds: Neither lower-case lambda nor upper case gamma are defined. All symbols used should be completely defined.

30. Line 486: The writing in this section is quite poor and makes it hard to understand exactly what has been done. It needs a comprehensive re-write. Line 492 refers to a figure – what figure? What is the contiguous inactivity fraction? (I can guess, but it is not a standard term, and hence should be defined.) What step sizes were used in period and period derivative, and what is the maximum phase error that could result? Latex errors (italicized units) abound in this section.

31. Line 506: why is there no uncertainty for the MW contributions?

32. Line 507: Reference for the chosen value and uncertainty of Halo DM?

33. Line 509: The text and references do not actually justify the range quoted for the IGM DM. The quoted reference considers f_{IGM} in the range 0.6 +/- 0.1, not 0.8, and in any case provides no hints as to how cosmic variance was translated into this DM range. This justification should be strengthened. Also, throughout this section (and indeed, throughout the entire paper) confidence intervals should be specified (or if are typically 68%, then this should be stated somewhere).

34. Equation 3: Half of the symbols used in this equation are not defined.

35. Line 514: the uncertainties presented for DM_{host} are clearly inaccurate, having presumably neglected the uncertainties on the MW halo (and made no attempt to estimate an uncertainty on the MW contribution). The number of significant digits is also incorrect. At no point in this section is the derivation of the final value of $912 + 69 - 108$ that is quoted in the main text shown. Whatever the number is in the main text should be supported by the analysis here, and it is not. How were the multiple sources of uncertainty combined?

36. Line 517: Where does the extinction-corrected H α flux come from? Which spectrum? What was

the extinction correction? The caption of extended data figure 4 says that the Palomar spectrum that is plotted was not corrected for slit loss, if it comes from this spectrum, was that correction applied, if so what was the result?

37. Line 518: How was the size of $0.5 \times 0.5''$ for the host galaxy estimated? At optical wavelengths, the situation is confused – there is enhanced emission at the site of the FRB, but the centroid of the galaxy is quite close by. It is not immediately clear to me from the description of the DBSP observations how much light the slit should be catching from the enhanced emission at the FRB location vs the bulk of the galaxy. This should be clarified.

38. Equations 4 and 5: again, many symbols are not defined: ϵ , ζ , T_4 , ...

39. Line 531: the logical conclusion from the lengthy analysis in this paragraph, that there is a local contribution to DM rather than the host contribution coming solely from the diffuse host galaxy, is not stated explicitly. And no attempt is made to say how well constrained **any** of these parameters are. This lack of rigor is relevant to the scattering analysis later.

40. Line 538 (and elsewhere): there is no justification to quote the DM to 4 significant digits.

41. Equation 6: m is not defined.

42. Line 544: what does “fold the frequency” mean?

43. Line 542-550: As already noted in point 14, surely the simplest assumption possible would be for no time dependence to scattering, and it would make more sense to fit a single scattering time to the entire ensemble of bursts? That would probably give a better estimate of the scattering time, and then also a better estimate of the distribution of widths. Otherwise, the widths of the bursts for which no scattering time was estimated (and which were then fit by an unscattered gaussian) are very likely to be biased high. Since the main purpose of this analysis is to estimate a scattering time in order to rule out the observed galaxy being a foreground source, obtaining a mean value this way, by fixing it across all bursts, must surely be preferred.

44. Equation 7, **again** half the symbols are not defined (considering both this equation and the unlabeled one on line 558).

45. Line 568: I agree with the analysis but not the firmness of the conclusions reached. Many other parameters than those mentioned at the end of the paragraph contribute to the estimated scattering time, and no attempt was made to estimate their potential ranges. To be definitive here, the authors should rule out the possibility of a DM from this galaxy (the putative host, which would actually be an intervening galaxy in this scenario) being sufficiently low given the uncertainties, and they do not do so at present. I want to stress that I agree with the likely conclusion, but it needs to be far better justified – or else the possibility needs to be left open (and some estimate made of the confidence level).

46. Line 569: nowhere is it stated (and referenced) that almost all FRBs, especially repeaters, are highly linearly polarized – because otherwise, why couldn't the observed polarization be due to an intrinsic low linear polarization, rather than depolarization? This should be stated at the outset. For that matter, the observed polarization is not well described. What are the upper limits on linear polarization and circular polarization? This is particularly important if these results are to be referred to in the main text.

47. Extended Data Figure 1: Why does this appear out of sequence (before other extended data figures that are referred to earlier in the Methods?). It took me quite some time to realize that the values printed in red are the uncertainty in the **peak** of the plotted distribution, as opposed to some kind of confidence interval. This should be made clear. Also, the fitted distribution for both τ and width seem to have non-zero constant components, which seems like a very strange thing to

allow in a gaussian fit to a distribution that should tend to zero at negative values (and positive infinity). What is the value of fitting a gaussian at all, as opposed to simply quoting e.g. the median value? In short, this needs a complete overhaul, but could be improved while also addressing my comments 14 and 43.

48. Extended Data Table 4: nowhere is it made clear where the systematic uncertainties in the final columns have come from. By cross-referencing, it is apparent that they are simply the overall systematic uncertainty estimated for the PRS from the stacked deep images at each frequency. But this is a lower bound to the systematic uncertainty for a given burst, since it assumes no time dependence at all – the radio positions used to estimate these offsets and corresponding uncertainties have been obtained from an image averaged over hours of integration (spread across days in the case of the L band bursts). If there are any residual calibration errors at the time of a given burst, they will lead to a systematic position offset that goes unappreciated here. If the authors provided more details of the solution interval and S/N threshold of their gain calibration step, this might provide some reassurance that the potential maximum size of any time dependent systematics is not too large. A better approach would be to select a time range centered on the burst and use positions of the continuum sources from this subset of the data to estimate the systematic offset in each case. If that is deemed too much work, then alternative methods to take this potential underestimate into account should be used – for instance, by looking at the chi squared of the final weighted mean. Doing this, and then taking the correlation between systematic position uncertainty for bursts at the same frequency, would be a more correct way to derive a final FRB position and uncertainty.

49. Extended Data Table 5: I'm curious how an apparently 20 sigma burst (burst C5) can be identified to have a width of ~0.1 ms with an uncertainty of 70 microsecond, given the 10ms sampling of realfast? All of the other bursts have a width uncertainty of 1-2ms, as I would expect. Perhaps this is a typo? Other numbers in these tables are curious: e.g. burst L1 has fluence 4(6) Jy-ms. Why is the fractional uncertainty >100%? Why is the localization of burst S1 more than twice as precise than any of the other S band bursts, despite burst S2 and S3 having comparable S/N? I see that it is much wider-band than the other two, but that doesn't really matter given that the image S/N is the final result of the width+fluence+bandwidth of the burst combined. Looking at the beam sizes and the S/N values quoted, it seems more like the statistical uncertainty is too large for the other bursts rather than too small for burst S1. This might indicate that the fitted source size was >> the synthesized beam size in those cases, which would be indicative of a concerning systematic error (as the FRB itself must be a point source, and hence should have a size approximately equal to the synthesized beam size).

50. Reference 41: why is this the arxiv reference rather than the published ApJ article?

Referee #2 (Remarks to the Author):

A highly active repeating fast radio burst in a complex local environment Niu et al.

General:

In this paper, the authors present the discovery of a repeating fast radio burst (FRB) source, whose host galaxy they identify. This source, FRB 190520, is similar to the first-known repeating FRB source,

FRB 121102, in that it is hosted in a dwarf galaxy and has an associated compact, persistent radio source (PRS).

The most novel aspect of the paper is that the authors show – assuming that the host galaxy association is correct - that the local electron density (DM) of the FRB 190520 must be quite large compared to other known repeating FRBs.

They argue that this may mean that it is younger compared to other sources. That may be, but I think it is also possible that the source is in a galactic-centre-like environment, near an accreting massive black hole (one of the hypotheses for FRB 121102, and as we see for the Galactic centre magnetar PSR J1745-2900 in our own Milky Way).

The authors also claim that the large local DM suggests that caution is needed in using FRB DMs to estimate redshift. Given that only a few percent of FRBs are known to be repeaters, and given that several repeaters have been shown to have much lower local DM contributions, I think this claim isn't well substantiated. In other words, does this one exceptional source really suggest that DM is a poor proxy for distance for FRBs in general?

Overall, I found the paper to be quite interesting (great that there is now an FRB 121102 twin!) and potentially suitable for Nature, but I would like the authors to address the previous comments, as well as those below.

Sincerely,

Jason Hessels
University of Amsterdam & ASTRON

Other general comments:

- Is the source actually exceptionally active compared to other well-studied repeaters? The claim of a link between "high activity" and the presence of a PRS doesn't fit with observations of FRB 20201124A.
- The paper should discuss how some repeaters *don't* have a PRS, down to very low luminosity limits (e.g. FRB 20180916B, Marcote et al. 2020 and FRB 20200120E, Kirsten et al. 2021).
- What are the actual constraints on the properties of the PRS (size and offset from FRB source and host galaxy)? How does this quantitatively compare to FRB 121102 (Chatterjee et al. 2017, Tendulkar et al. 2017, Bassa et al. 2017, Marcote et al. 2017)?
- The paper should describe the burst properties (e.g. wait times distribution, energy distribution, etc.) in order to compare with other repeaters. For example, are the bursts of the same average energy compared to FRB 121102?

- The apparent lack of DM variations should be discussed, since one might expect variations if the source is exceptionally young.

- There is not much discussion of the host galaxy properties. This is only the 2nd known dwarf host for an FRB, but repeaters have also been found in a wide range of galaxies. How does that fit with the author's interpretation?

- The chance coincidence with the host galaxy is not discussed in proper detail (there is only one sentence in the paper on this, which simply states that the chance association probability is low, without describing any of the assumptions or giving exact numbers). Given that many of the results depend on a robust host galaxy association, this deserves a proper treatment.

- Some of the analyses - e.g. short and long-term periodicity, scintillation bandwidth, etc. - could have a related figure in the Methods.

Detailed comments (some are minor typos!):

Title:

- Line 1: "A highly active repeating fast radio burst in a complex local environment"

- I would suggest that the title focuses on the most novel aspect of the paper, which is the high local electron density (DM). The source activity is not unprecedented, and "complex local environment" is vague.

Summary paragraph:

- Line 41:

- "central engine" is jargon and may not be clear to a wider readership

- Given current FRB observations, I would tend to talk about multiple possible source types for their origin.

- Line 42:

- "the dispersion sweep of FRBs provides": awkward mixing of singular and plural in this sentence. There are other examples of this in the Summary paragraph and elsewhere in the paper.

- Line 43:

- The dispersive sweep is also probing the ionised material in the host galaxy, as well as the Milky Way.

- Line 43:

- "Active repeaters has been shown": *some* repeaters have shown this, while other repeaters have not.

- Line 47:
- "host galaxy of high star formation": high *specific* star formation, I assume.

- Line 50:
- "suggesting caution in inferring redshifts for FRBs without accurate host galaxy identifications": but only a few percent of FRBs have been shown to be repeaters. So the paper does not present evidence that this is true for a significant fraction of FRBs.

- Line 53:
- "may point to a distinctive origin or an earlier evolutionary stage for highly active repeating FRBs": there hasn't been any quantification yet about whether this source is actually more active than other well-studied repeaters. That's necessary to justify this statement.

Main:

- Line 59:
- "detected 75 bursts in 18.5 hrs": FRB 121102 has sometimes been seen to produce as many bursts in ~1 hour of observations. So I would not say that FRB 190520 is exceptionally active.

- Line 59:
- "a mean pulse dispersion measure (DM) of $1202 \pm 10 \text{ pc cm}^{-3}$ ": why is the uncertainty on the DM so large?

- Line 72:
- "Using averaged flux density at each"  "Using the average flux density of each"

- Line 73:
- "we find a PRS flux density spectrum can"  "we find that the PRS flux density spectrum can"

- Line 74:
- "index -0.41"  "index -0.41 " (negative sign)

- Line 78:
- "Given the measured offset of the FRB from the galaxy": what is the measured offset found to be?

- Line 79:
- "we estimate a chance coincidence probability": I assume that there is a more detailed demonstration presented in Methods.
That should be cited here.

- Line 79:
- "a chance coincidence probability of less than 1%": why not quote the actual value?

- Line 79:
- "supporting J160204.31–111718.5's being the host galaxy of FRB 190520"  "supporting the claim

that J160204.31–111718.5 is the host galaxy of FRB 190520"

- Line 84:

- "that revealed the"  ", which revealed the"

- Line 84:

- "to be $z = 0.241$ ": include uncertainties

- Line 92:

- "a relatively high star-formation rate for its stellar mass": the Summary paragraph simply says "high". What does "relatively" mean, more quantitatively? Is the rate somewhat above average, in the top 10%, etc., compared to other dwarf galaxies?

- Line 92:

- "At the luminosity distance implied by the redshift": would be better to also state what that distance is.

- Line 95:

- "of a FRB source"  "of an FRB source"

- Line 98:

- "For nominal DM contributions from the Milky Way (100 pc cm^{-3} for MW disk and MW halo)": is this what electron density models would predict in this direction? Would be good to state that this source is well off the Galactic plane.

- Line 99:

- "and host galaxy (50 pc cm^{-3})": why is the host galaxy contribution assumed to be smaller than that from the Milky Way?

- Line 99:

- "and also assuming baryon fractions of 0.6 to 1 for the ionized IGM": citation needed.

- Line 117:

- "observed scattering time of 10.8 ms": uncertainty should be included, as well as a reference to Methods for more detail on how this was determined.

- Line 122:

- "FRB 190520 shows that the distribution of DM_{host} values can have a long tail, which adds considerable variance to estimates for the IGM.": this is only true if a significant number of FRBs are like this source. Given that $>95\%$ of FRBs appear to be non-repeaters, it seems to me that one can still accurately estimate the IGM contribution in the vast majority of cases - unless non-repeaters are, in the future, shown to often inhabit dense local environments.

- Line 128:

- "Another repeating source, FRB 201124A, was also associated with persistent radio emission.

However, through...": I don't think that it's important to discuss this in the Main part of the paper, since the upshot is that FRB 20201124A (note correct name) doesn't have a PRS like that of FRB 121102 or FRB 190520. Much more important, is to comment on the fact that other repeaters have *no* PRS, down to very constraining limits. In particular, FRB 20180916B's potential PRS is constrained to be $\sim 300x$ less luminous than FRB 121102 (Marcote et al. 2020), and the odd FRB 20200120E in a globular cluster provides even tighter constraints on PRS luminosity because of its exceptional proximity (Kirsten et al. 2021). Actually, it's also important to discuss that FRB 20201124A doesn't have a compact persistent radio counterpart (the preamble explanation isn't necessary in the Main part of the paper though) because that FRB has been (at times!) extremely active. That doesn't fit with the idea that more active sources are associated with PRSs.

- Line 132:

- "the PRS luminosity would imply a star-formation rate of $\sim 10 M_{\odot} \text{ yr}^{-1}$ ": how would that compare to what has been observed from dwarf galaxies? Is it known to be possible to have such a high rate in a dwarf?

- Line 133:

- "Given the extreme PRS luminosity, its unresolved structure in VLA observations": it would be good to state what the constraints on the physical size (and offset) actually are, and to compare these to FRB 121102 (Chatterjee et al. 2017; Marcote et al. 2017).

- Line 134:

- "and its offset from the center of the optical emission of the host galaxy": I think it's important to note, however, that FRB 121102 and its persistent radio source are very close to a dominant knot of star formation (Bassa et al. 2017) in its dwarf host galaxy (though in HST observations ever so slightly offset from it...).

- Line 136:

- "as found for FRB 1211022.": here I would also cite Marcote et al. (2017), since that provides the most precise quantification of the physical size of FRB 121102's PRS, as well as its maximum possible offset from the FRB source itself.

- Line 137:

- "Burst repetition and spectral structure have been used to argue": also Faraday rotation measure (e.g. Michilli et al. 2018) and time-frequency structure (e.g. Hessels et al. 2019) have been used to distinguish repeaters from apparent non-repeaters.

- Line 139:

- "The observed burst properties are subject to observational biases": this seems to imply that the current study isn't also affected by observational biases, which is of course not true. Rephrase.

- Line 139:

- "but PRS emission and DM_{host} reflect different aspects of the FRB environment.": difference between repeaters and apparent non-repeaters? Difference compared to the aforementioned burst properties? I'm unclear on what point this short paragraph is trying to make.

- Line 143:

- "two FRBs associated with PRSs are among the most active": is this a robust statement? FRB 121102 has still only shown one burst in CHIME/FRB observations, whereas FRB 20180916B has been detected many dozens of times by CHIME/FRB and shows no PRS. FRB 20201124A has also been extremely active (at times!) at 1.4 GHz, but shows no PRS.

- Line 144:

- "and have large DM_host values": the DM_host of FRB 121102 hasn't been mentioned yet (see Tendulkar et al. 2017 and Bassa et al. 2017). The FRB 121102 constraint of $55 \text{ cm}^{-3} < \text{DM}_{\text{host}} < 225 \text{ pc cm}^{-3}$ isn't necessarily "large", like is being claimed for FRB 190520.

- Line 153:

- "The discovery of FRB 190520 and its high similarity to FRB 121102 demonstrate that some FRBs have very large local DM and PRS counterparts.": I'm surprised that the main part of the paper hasn't said anything explicit about the extremely high and variable RM of FRB 121102 and how FRB 190520 compares.

Figure 1:

- What time and frequency resolutions are being used for plotting?

- What DM is being used for plotting?

- Indicate that some frequency channels have been removed to excise radio frequency interference. Also, it is best practice (I think) to mark these with ticks at the side of the plot.

- Is the colour scaling of the dynamic spectra linear? Any clipping at the low/high end of the value distribution? Why the blue colour instead of greyscale?

- Indicate the typical observation durations here.

Figure 2:

- Indicate what is causing the bright artefacts in the optical images.

- Could be useful to have a zoom-in at the source position in the optical images.

Figure 3:

- "The expected DM contribution of the intergalactic medium (orange line) is": citation to Macquart et al. 2020 needed.

- Mark FRB 121102 in this diagram.

Author contributions:

- Line 259:
- "Energy"  "energy"

- Line 261:
- "Parks observations"  "Parkes observations"

Methods:

1 Observations:

- Line 271:
- "In this first discovery observation, 3 bursts were detected in 10 seconds, and another burst was detected 20 seconds later.": the paper would benefit from describing some basic statistics of the bursts, like wait time distribution and energy distribution. This is to inform the comparison to FRB 121102 and other repeaters.

- Line 274:
- "follow-up observations were performed with FAST on April 25th and May 22nd": what year?

- Line 276:
- "using the ~ 100 mas localization from VLA": point to the relevant section that describes this.

- Line 279:
- "monitor observation"  "monitoring observation"

- Line 281:
- "has been transformed to the arrival time at the solar system barycentre (SSB) at 1.5GHz": using what DM(s)?

- Line 284:
- "The sub-pulse is recognized if the profile peak does not fall behind the noise baseline.": I'm not sure what this means. Needs to be explained more clearly.

- Line 287:
- "Radio Frequency Interference": don't capitalise.

- Line 300:
- "stokes I"  "Stokes I"

- Line 300:
- "and the": missing space before "and"

- Line 300:

- "the pulse width is adapted by a boxcar in the search": reword, this is an awkward and unclear description.

- Line 303:

- "DM range": missing space after "range"

- Line 304:

- "from central beam"  "from the central beam"

- Line 305:

- "stokes parameters"  "Stokes parameters"

- Line 318:

- "-11"  "\$-11\$"

- Line 322:

- "detail in"  "detail in Refs."

- Line 332:

- "goes through"  "go through" (because "candidates" is plural)

- Line 336:

- "etc"  "etc."

- Line 336:

- "section 2.4 of"  "Section 2.4 of Ref."

- Line 338:

- "see section 2 of"  "see Section 2 of Ref."

2 Localization of bursts:

- Line 357:

- "J1558-1409"  "J1558\$-\$1409"

- Line 367:

- "to signal to noise ratio"  "to S/N" (for consistency with rest of paper)

- Line 371:

- "We estimate statistical error to be": the scatter in positions is ~3-4 larger than this. Would be good to comment on that.

- Line 371:

- "-11"  "\$-11\$"

- Line 371:
- "0.023''": would be better to use " $0.023^{\prime\prime}$ " (here and elsewhere in the paper where arc-minute/second is meant)

- Line 374:
- "discussion of"  "discussion of Ref."
- "been described in"  "been described in Ref."

- Line 374:
- "We model the pulse profile": is the fitted DM from maximizing the peak S/N? Would be good to comment on what these DMs mean compared to the analysis method used for the FAST bursts. The FAST burst DMs show little-to-know scatter between epochs when the structure-maximizing assumption/technique is used.

- Line 376:
- "Following"  "Following Ref."

3 Persistent Radio Source:

- Line 383:
- "The VLA visibilities with 3 or 5 s sampling time were saved...": I would clarify to the reader that the data around bursts are saved at high time resolution (~10ms), and that the entire observing span (tens of minutes / hours) are in parallel saved at this lower time resolution.

- Line 386:
- "J1558-1409"  "J1558 $\hat{}$ -1409"

- Line 388:
- "its Stokes I"  "its Stokes I data"

- Line 394:
- "The VLA campaign obtained two-epochs": in this sub-section, I think it would make more sense to first describe the observations and thereafter the analysis, as opposed to the opposite order that's currently used.

- Line 414:
- "We report the PRS coordinates based on the measurements at 3 GHz": and what is that position?

- Line 419:
- "Power-Law(PL)"  "Power-Law (PL)"

- Line 420:
- "-0.41"  " -0.41° "

- Line 421:
- "image, We"  "image, we"

- Line 422:
- "sources , including"  "sources, including"

- Line 423:
- "with a flux density higher than 260 μJy ": make it clear to the reader why this particular flux density is relevant.

4 Optical Redshift Determination:

- Line 442:
- "of FRB 190520 field"  "of the FRB 190520 field"

- Line 443:
- "August 05th"  "August 5th"

- Line 454:
- "which later is found to coincide with the location of the pulsation emission from FRB 190520 in 0.18".": this is confusingly worded.

- Line 455:
- "pulsation emission"  "burst emission"

- Line 460:
- "in to the slit"  "into the slit"

- Line 462:
- "reduction on other"  "reduction of other"
- "instrumentation effects"  "instrumental effects"

- Line 467:
- "The corresponding redshift derived based on these two spectral lines is $z = 0.241$ ": with what uncertainty?

- Line 473:
- "indicating the extended R'-band structure has the same redshift of $z = 0.241$ ": what is the actual measurement and uncertainty in this additional analysis?

5 FAST Burst Sample Analysis:

- Line 484:
- "mJy · ms"  "mJy ms"

- Line 485:
 - "for excluding waiting time"  "when excluding waiting times"

- Line 486:
 - "A period search was conducted from the total 75 bursts from FAST.": this is confusing because Table 1 reports 79 bursts.

- Line 491:
 - "by traversing the period (P) among"  "by searching for period (P) from"

- Line 492:
 - "Almost all values in this figure are less than 0.4.": what "figure" is being referred to? Also, why is 0.4 the threshold?
This is confusing.

- Line 494:
 - "range of period (P) among"  "range of period (P) from"

- Line 495:
 - "2 – 365 d": "d" shouldn't be italicise here (or later in the same sentence)

- Line 495:
 - "and period derivative (Pdot) among"  "and period derivative (Pdot) from"

- Line 496:
 - "the mjds"  "the MJDs"

- Line 498:
 - "observing session mjds"  "observing session MJDs"

- Line 510:
 - "discussion in"  "discussion in Ref."

- Line 513:
 - "The averaged DM_obs from all bursts is 1202 ± 10 pc cm⁻³": the paper should be more clear about whether the 10 pc cm⁻³ uncertainty here is due to measurement uncertainty or scatter in the measurements.

- Line 514:
 - " 921.1 ± 10 "  " 921 ± 10 " (overquoting significant digits)

- Line 515:
 - "for NE2001"  " for NE2001" (missing space before "for")

- Line 515:
 - "for NE2001 and YMW16 model separately"  "for the NE2001 and YMW16 models, respectively"

- Line 525:
 - "inferred value of DMh"  "inferred value of DM_host" (for consistency with the rest of the paper; use the same name for the same quantity throughout)

- Line 536:
 - "Using Equation": be consistent in use of "Eq." vs. "Equation"

- Line 542:
 - "Among the detected 79 pulses of FRB 190520, 28 pulses show frequency-dependent temporal width that is consistent with scattering.": is it also potentially related to an un-resolved time-frequency drift (sad-trombone effect)? What is the evidence that it must be solely due to scattering?

- Line 544:
 - "we first fold the frequency": unclear what this means.

- Line 547:
 - "The upper panels of Figure 1"  "The upper panels of Figure ED1"

- Line 581:
 - "Such a large RM is even larger than that of FRB 121102": the text here should be more clear that it's also possible that the source is intrinsically depolarized at low frequencies, or that generalised Faraday rotation is prohibiting detection of an RM. I think that's just as plausible a hypothesis as intra-channel Faraday rotation and a large RM.

- Line 582:
 - "Large RM also indicates that the FRB 121102 could be very young": not necessarily. It has also been hypothesised that the source is in a galactic-centre-like environment, in the vicinity of an accreting massive black hole (which in turn is the PRS). The Galactic centre magnetar in our galaxy has a high RM, but we don't have reason to believe that that is because it is very young compared to other magnetars.

- Line 583:
 - "similarly, FRB 190520 could also be very young": quantify.

- Line 585:
 - The references in Main go up to and including #31, so the first reference in Methods should be #32.

- Line 626:
 - Ref. 50 also appears as Ref. 4. Double check that there aren't more redundancies between the two reference lists in the Main and Methods sections.

Figure ED1:

- Given that the inferred scattering timescales are comparable to the burst widths, how can one know that this isn't actually scattering, but rather due to an unresolved time-frequency drift (sad trombone effect)? Can the authors provide some figures of dynamic spectra and fits that support this interpretation and disfavour (or rule out) time-frequency drift?

Figure ED2:

- No comments.

Figure ED3:

- "corresponding radio flux are shown"  "corresponding radio flux densities are shown"

- "the flux error"  "the flux density error"

- "as -0.41"  "as $-0.41 \pm$ "

- I did not see a discussion of the apparent variability of the flux density of the PRS. What is the amplitude of variability (~20%), how does that compare with FRB 121102, and is this consistent with being due to scintillation in the Milky Way ISM?

Figure ED4:

- Are there any constraints on the metallicity of the host galaxy?

Figure ED5:

- No comments.

Table ED1:

- Given how close the DMs are to each other, from day to day, it appears like the DM uncertainties are overestimated. A figure of DM and apparent scattering measure vs. time would be valuable.

- Include information on the reference system used for the burst times.

- Include note on how the DMs were determined (and why it's the same for each day).

- Emphasise that the "Energy" is assuming isotropic emission, which is unlikely to be true.

Table ED2:

- Are these "on source" observation durations, i.e. not including phase calibrator (etc.) scans? Clarify

in a table note.

Table ED3:

- "in ra. and dec."  "in RA and Dec" (to match table labels)
- Values in table have hyphens instead of negative signs (\$-\$).

Table ED4:

- "The alphabet in the first"  "The label in the first"
- "image signal to noise ratio"  "image S/N"
- The DMs reported here are wildly different. Explain why, and what these correspond to. Were these the discovery trial DMs?

Table ED5:

- The DMs quoted here were determined using different assumptions compared to the FAST burst analysis. Why? Also, the DMs appear to vary, but there is not discussion of this.
- Some of these bursts are ~10x brighter than the brightest bursts seen by FAST. Isn't that surprising?

Author Rebuttals to Initial Comments:

Response for 1st referee

Referee #1 (Remarks to the Author):

The authors have presented a study of the newly discovered and highly active repeating fast radio burst FRB 190520. This study has shown a number of remarkable features that are best explained by a dense and potentially highly magnetized ionized plasma surrounding the FRB source, including a high excess dispersion measure, a co-located continuum radio source, and a non-detection of linear polarization that could be due a high rotation measure. Along with the first repeating FRB 121102, which is also a highly active source and which also possesses similar characteristics, this suggests that FRB activity levels may be correlated with their local environment. Whether this hints at a distinct source population or a different evolutionary phase compared to the bulk of the FRB population, this is a highly significant result and well worthy of publication.

###Response: Thanks!

The impact of large local contributions to the FRB DM on the use of FRBs as probes of intergalactic baryons is also important, as noted, although the manuscript does not specifically highlight that if FRB repeat rate is correlated with local environmental properties, this may offer an opportunity to calibrate local contributions to the FRB DM, or at least identify potentially problematic sources for exclusion from IGM studies. Overall, the original and significant results presented in this manuscript will certainly be of interest to the large and growing community of FRB researchers and those that hope to use FRBs as tools to study the Universe. I have identified a number of places in the manuscript where additional information is warranted or other improvements are required to aid understanding for the reader. In some cases, this may require additional analysis, or else the authors should justify in the text why this additional analysis has not been undertaken. I have identified these in a (rather lengthy) numbered list below, which also includes some comments on the presentation of uncertainties and other statistical matters. Before I enumerate specific suggestions and criticisms, though, I will comment on the clarity, context, and conclusions of the paper as a whole.

First, I should note that there are very many minor and not-so-minor typographical errors throughout the paper: missing articles, plural/singular disagreement, italicized units from the use of math mode in latex, symbols not defined, incorrect numbers of significant digits, etc. I have highlighted some of them in my enumerated list below but this list is definitely not exhaustive – I suggest the authors give the entire manuscript another, much more thorough proof-reading. These issues were especially prevalent in the Methods section.

### Response

Thank you for pointing these out! We have gone through the Methods section thoroughly.

A second important point regarding clarity refers to the usage of the terms “Persistent Radio Source” and “persistent radio emission. I would encourage the authors to consider carefully their use of “PRS” vs “persistent radio emission”; the former is used to imply compact emission co-located with the FRB itself, while the latter could be emission associated with other host galaxy properties such as star formation that is not directly associated with the FRB progenitor. Many FRB papers are guilty of muddying the waters here, but I would encourage a cleaner separation between the two, perhaps by making it clear that PRS refers to something definitely associated with the local FRB environment, using an alternative such as “continuum radio emission” when discussing galaxy-wide radio emission, and using “potential PRS” when continuum radio emission may or may not be associated with the local FRB environment.

Response

The terminology is indeed muddy in the literature and in our initial submission. We have modified the text to use more clear terminology. We now initially refer to the persistent emission only as “the radio continuum counterpart”. We then justify its association, compactness, and differentiation from star formation. Only then we refer to it as “a PRS”, in comparison for that of FRB 121102.

Final comment on clarity: I encourage the authors to consider the use of TNS naming designations for the FRBs. I understand that the historical, non-TNS name for FRB121102 is in widespread use, but the field seems to be moving towards adoption of TNS names, and the use of the non-TNS designator “201124A” for FRB 20201124A sticks out.

Response

Thanks, we registered the TNS name for FRB 190520 as “FRB 20190520B”. In the first instances, we note “FRB 121102 (corresponding TNS name FRB 20121102A, same hereafter)” and “FRB 190520B (corresponding TNS name FRB 20190520B, same hereafter)”, and use FRB 190520B and FRB 121102 consistently thereafter. For other sources, we use all TNS names.

In terms of conclusions, the primary conclusion of the paper, that FRB repeat activity level may be correlated with local source environment, is well supported. As I noted in the first paragraph of the review, I think that a logical extension of this connection is that outlier sources that might otherwise bias the Macquart relation between DM and redshift may be identifiable (and potentially even calibrate-able.). It would seem logical to comment on this directly in the manuscript.

Response

Thanks for pointing this out. We add the comment to the conclusion:

“Further study of such correlations may help identify outliers to the Macquart relation and potentially help calibrate biases.”

I note that the final sentence of the main text, “Either possibility has direct impact on calibrating the FRB DM as a probe of cosmic baryons and on understanding FRB

origin(s)“, does touch on this point, but at no stage in the text was the link made to *how* this calibration would be done.

###Response

We have simplified the sentence to “Our results signify a wider range of host-galaxy DMs need to be considered for constraints on cosmic baryons in the IGM.”. The last paragraph was then refocused to possible relation between activity and PRS. “More such detections in the near future will also further clarify the relation between PRS and FRB activities. Active repeaters with PRS may either be a distinct population or FRB sources at earlier evolutionary stages.”

In the second last sentence of the conclusion, however, I disagree with the statement that more detections will distinguish between the two scenarios presented in the near future. It is certainly plausible that a correlation may be found between burst repetition rate and the presence of a PRS and/or a large DM_host, and that this may be related to source age, as we have already established. But I don't see how this can then be connected to apparently non-repeating sources. *Most *repeaters* already don't have a detectable PRS, and less active repeaters already have unremarkable host DM contributions.* So any such correlation between repetition rate and PRS properties, if it exists, would presumably become undetectable (since the PRS will not be detectable, and any excess contribution to DM will become negligible) well before you get to the apparently non-repeating FRBs that form the bulk of the FRB population. So I believe that this statement needs adjustment or additional support.

###Response

Agreed. We have altered the final sentences to point out that the incidence of repeaters and PRSs and possible correlations between them is simply a topic of continuing interest.

Final overarching comment on the manuscript: I am not sure why so much space is dedicated to the exclusion (based on scattering) that the host is a chance alignment in the main text. This can be adequately covered in the Methods and simply referred to, as it is not of general interest.

###Response

We have modified the paragraph in the main text that is devoted to the exclusion of a chance alignment based on scattering. The paragraph is now shorter and states the main reasoning of the argument, with the details expanded in the Methods. We have also modified the wording to emphasize that the scattering analysis affirms the low chance coincidence probability.

On the other hand, the extremely important result that the RM may be very high (like 121102) is referred to nowhere in the main text, despite this being (if confirmed) an extremely important additional link between the two sources in terms of their local environment. If the results of the analysis are not referred to in the main text, why is this in the Methods?

Response

The claim that the RM may be very high is only one interpretation of the FAST result. Higher frequency bands and more observation will be needed, per experience with FRB 121102 (no RM detection by FAST in L-band). We still leave it in the Methods in case readers may be interested in the RM results so far.

Enumerated comments in order of appearance in the text:

Ref 1:

1. In the abstract, a number of quantities are used without context or insufficiently defined. Fast Radio Bursts themselves are introduced with no context. The term “repeater” is introduced without context, as it “persistent radio source” (noting my earlier point about defining PRS precisely). “The estimated host galaxy contribution DM_{host}” is similarly imprecise: at this stage, DM (Dispersion Measure) has not been defined, so it would not be clear to most readers what this “contribution” is. Grammatical errors abound: Fast Radio Bursts (FRB) should have “FRBs” within the parentheses, line 44 should read “...to be associated with **a** persistent radio source”, no space before “Here” on line 45, line 51/52 should read “...with **a** confirmed association between **an** FRB and **a** compact PRS”, etc.

###Response

Thanks!

The summary paragraph has been restructured and some contexts were added. Some definitions have to wait for the main texts due to length limitations.

“Fast radio bursts (FRBs) are the most energetic radio transients in the Universe, the central engines of which remain unknown and could be diverse. The dispersion sweeps of FRBs provide a unique probe of the ionized baryon content of the intergalactic medium as well as FRBs’ natal environments.”

2. Line 68:

using an uncertainty in arcseconds for right ascension right next to the position (in seconds) is rather confusing. I suggest that either giving the uncertainty in seconds, or quoting the full position (RA+Dec) followed by an uncertainty ellipse in arcseconds, would be less prone to misinterpretation. Moreover, it is stated that the uncertainty is dominated by systematic considerations, but this is never shown for the FRB in the Methods (it is shown for the PRS). Finally, I disagree with the uncertainty presented for the FRB – how can the systematic uncertainty on a position which is obtained from a handful of milliseconds of exposure time have a smaller systematic uncertainty than the PRS, which has hours of exposure time? In a best case, they would be identical, but as I discuss in the methods comments below, I think the systematic contribution to the FRB position uncertainty has been given insufficient attention.

###Response

We have now included the errors quoted separately from the uncertainty to help avoid confusion. Regarding the uncertainty, we have also changed a few things but those are noted in your later comment that discusses these issues in more detail.

3. Line 71: too many significant digits in the Declination position, and too many significant digits in both RA and Dec. uncertainty.

###Response

Thanks, we have edited those.

4. Lines 78-80: While the association between the galaxy and the FRB is quite secure, a chance probability calculation is a cruder measure than a Bayesian inference (e.g., Aggarwal+2021) that also takes into account other nearby galaxies and their properties. The authors may wish to consider evaluating the probability of association using a Bayesian method such as the one indicated.

###Response

As mentioned, we believe that the association between the galaxy and the FRB is secure, and the chance coincidence of other optical blob is much higher at >20%. Further detailed association calculation would be presented in a future work.

5. Line 87/88 – both R and R' band are referred to. Which was used? I note that R' and R are again both used later in the Methods section. Line 89: How is that H-alpha luminosity derived? It is not derived anywhere in the Methods.

###Response

The phrase “extended R-band structure” includes a typo. It should be R'-band. We have corrected that typo in the text. Thanks!

The H-alpha luminosity is derived based on the luminosity distance for $z = 0.241$, and the extinction corrected H-alpha flux. The detailed approach of extinction correction on the H-alpha flux will be included in a subsequent paper in preparation.

6. Line 101: There is no reference to the Methods section where the DM_IGM number is calculated. I also am unable to reproduce the reasoning that yields the estimated value and range for DM_IGM, but we shall return to that in due course.

###Response

The main text and Methods section now elaborate on how the calculations were done, including errors in the MW estimates (NE2001 error and range of possible halo contributions to DM). The error ranges we use are generous (i.e. conservative) and indicate, also including cosmic variance in the IGM contribution, that there is no way to avoid the conclusion that the host-galaxy DM is large.

7. Line 107: “The measured DM and scattering properties exclude” – need to say what is being referred to hereafter “properties” (i.e., “of the FRB”)

###Response

Done.

8. In this same paragraph, the predicted scattering time in the event of a chance background source is only two orders of magnitude greater than observed. In the Milky Way, scattering times show a very large degree of variability. I would suggest that the (lack of) scattering on its own is not conclusive evidence to rule out a background source,

but combined with the low Pchance of the host galaxy association, it further strengthens the already very strong case.

###Response

The case is stronger than stated in this comment. Yes, the Milky Way produces a wide range of scattering times, but that is over a wide range of dispersion measures. For a specific value of DM, e.g. 500 pc/cc, we see a total range that might be as large as two orders of magnitude if outliers are included but the probable range is only about 1.5 orders of magnitude.

On geometrical grounds, the scattering from an intervening galaxy would be more than 10^4 times that of a host galaxy if the scattering medium is the same. Since the host galaxy DM appears to be > 500 pc/cc (in its frame, not the observer's), we would expect for a Galactic pulsar a mean scattering time of 9.6 ms at 1 GHz and a probable range from 1.6 to 54 ms. These numbers are from a fit to $\tau(\text{DM})$ for Galactic pulsars that appears in several places in the literature (e.g. <https://arxiv.org/abs/2108.01172> equation 8 and references therein, e.g. Krishnakumar et al. 2015, Bhat et al. 2004, NE2001 2003). For an intervening galaxy that produces the same DM of 500 pc/cc, we include the geometric factor of 10^4 and another factor of 3 that corrects the Galactic prediction for spherical waves from a pulsar to plane waves for an extragalactic source, and we divide by a $(1+z)^3$ factor to get the scattering time in the observer's frame.

This gives:

$\tau(\text{intervening}) > 10^4 * 9.6 \text{ ms} * (\text{geometric factor} = 3) / (1+0.241)^3 = 150 \text{ sec}$ at 1 GHz compared to 40 ms for the measured scattering time (scaled to 1 GHz). This is more than three orders of magnitude larger than observed. Galactic variance can reduce this by a factor of $(1.6 / 9.6) = 0.2$ or increase it by a factor of five.

An alternative might be to say that an intervening halo produces low-level scattering that, with the 10^4 geometric factor, multiples up to the observed scattering. That would mean that most of the DM not from the IGM and MW would be from the host galaxy. That would produce some scattering but perhaps not as much as observed due to the redshift factor and the greater IGM contribution if the host is at $z > 1$, say. The problem with this configuration is that there is no evidence for *any* scattering from galaxy halos, certainly not from the MW nor from M81's halo.

All told, the scattering constraint strongly favors scattering from a host galaxy at $z=0.241$ with a large DM contribution.

For completeness, one might argue that the association of the $z=0.241$ galaxy is completely unaffiliated with the FRB source and has no influence on the line of sight. Then all bets are off, but it would be a (unlikely?) coincidence that the H-alpha measurements yield a DM estimate that is compatible with the inferred DM_{host} if the host is in fact at $z=0.241$. A spurious association seems to be a more convoluted interpretation than the one we put forward in the paper and where the scattering is in fact constraining on the presence or not of an intervening galaxy.

9. Line 126: Is it not worth also commenting on the similarities in terms of an apparent star formation peak at the location of the FRB and PRS for both 121102 (Bassa+2017) and 190520?

###Response

Thanks, It is worth commenting on the similarities, we also have a follow-up optical paper to describe that.

10. Line 139/140: DM_host can also be observationally biased, as it contributes to an observational quantity (the total DM) to which different FRB searches have different sensitivities.

###Response

We agree that large values of DM_host are less likely to be detected, since FRB searches are usually sensitive to a limited range of (total) DM. We have modified this sentence to reference this possibility.

11. Line 142: To be fair, this sentence should note that not all FRBs have comparably deep limits on the presence of a PRS.

###Response

Thanks! The statement was added

“Some active repeaters do have comparably strict limits on PRS counterparts, which suggests complexities in the connection between burst activity and PRS and/or diversity among repeaters.”

12. Figure 2: I think that either a zoom in (for one of the bands) and/or some annotations of other nearby sources would be valuable. This goes to point 4 above – are any of the other relatively nearby sources larger galaxies that may have a reasonable likelihood of harboring an FRB in their outskirts?

###Response

We thank the referee for the suggestion. Here we provide the zoom-in version below in response. The members in our collaboration have considered the zoom-in version of Figure 2, but we still prefer the current plot configuration to display the relations between radio and optical emissions in the field. The zoom-in plot will be discussed in more detail in the follow-up paper soon.

The persistent radio emission and repeating FRB bursts are all coincided with the optical emission shown in the Figure 2 within the position uncertainties and the resolutions of the optical/NIR images, the probability of false association is only $< 1\%$. The next nearby object in J-band is 6.5" to the north of the FRB location. The same method put the false association to be $> 20\%$, which translates to the object being likely just in the neighborhood coincidentally.

Fig. Zoom in optical plots.

13. Figure 3: why is there no uncertainty on the MW contribution to DM from NE2001? The authors could consider the difference between NE2001 and YMW16 as a crude estimate of the uncertainty. No reference is given for the MW halo contribution and uncertainty, and likewise no reference is given for the relationship used for the expected median extragalactic DM contribution or its uncertainty.

###Response

The main text and Methods have been modified to make clear how the calculations were done and uncertainties discussed.

14. Line 283: Why would some bursts show scattering and others not? Surely the simplest assumption is that the scattering is time independent, which would enable this parameter to be fixed across all bursts and constrained with good precision, which would then lead to a better precision on the (deconvolved) gaussian burst widths for all bursts? Non-gaussian intrinsic structure will surely complicate the analysis, but it will also be complicating the present analysis (where the choice of scattering vs no scattering is apparently made by eye) already, and the assumption of constant scattering is at least physically justifiable.

###Response

The band-limited nature of the bursts means that some will be more scattered than others, depending on the center frequency and frequency extent of their measured flux density. This could have been dealt with by adopting a fitting function for all bursts with a single value of the scattering time and then taking into account the frequency range for each burst. We have reason to investigate, however, that the scattering time actually is different for different bursts. It is entirely possible that FRB environments can produce variable scattering, similar to what is seen for the Crab pulsar giant pulses, and a patchy medium can even produce different amounts of scattering for bursts that are closely spaced in time. The measured scattering time can also in fact be influenced by refraction or lensing that works in tandem with diffractive scattering, and this refraction/lensing can also be time-variable. Sorting out these possibilities is beyond the scope of the present paper but we are actively investigating it.

15. Line 284: what is “the sub-pulse”? presumably “a sub pulse”, but this needs to be much more carefully defined. What level of significance is used to define a sub-burst that should be fitted (currently “the noise baseline” is stated)? At what width?

###Response

Thanks! We change the term to “a sub pulse”. Since there is no standard pulse profile (nor average) for repeaters yet, the use of pulsar terminology is empirical. Currently, when the 'bridge' between the two closely-spaced-in-time peaks drops more than 5 sigma below the higher peak, we consider them to be two bursts. Otherwise, they are considered structures within one event (sub pulses).

16. Line 298: why is Heimdall (searching) described after the burst modelling? This is backwards, surely it would make more sense to introduce burst detection prior to the burst modelling.

###Response

That is correct, The order has been switched.

17. Line 329: How is FETCH applied to imaging data, which has multiple spatial pixels, each of which contains time-frequency data? Is the time-frequency data from a single pixel extracted and fed into FETCH? Further clarification is needed (the supplied references do not explain.)

###Response

Thank you for pointing this out. Yes, the time-frequency data from a single pixel is extracted and then converted into the required inputs (dedispersed frequency time and DM-time) for FETCH. We have modified the text to make it clear.

18. Line 334: More details of the refined imaging of candidates is appropriate. What is the pixel size used (and how does this differ to the real time search)? What is the step size of the DM grid? What RFI flagging parameters are changed? All these details should be presented. The supplied reference (Section 2.4 of Law+2020) does not go into these details.

###Response

The details of the refinement process were different for different candidates. We have added text to this section providing more details.

19. Line 348: The description of the impact of not performing deconvolution are somewhat imprecisely described in the first paragraph of Section 2. For a point source, deconvolution should not normally affect the peak of the map substantially (dirty vs clean). What is does is remove the PSF shape from the surrounding pixels and replace it with a smoother shape. After all, once a source has been deconvolved, the final image is obtained by convolving the model with a gaussian approximation of the PSF before added it to the residual, but this does not “spread the signal” out – at least, not in a significantly different way to way the PSF does. What deconvolution does do is enable a higher S/N when performing a gaussian fit in the image plane, since the weighted average of all the pixels that contain some signal are used, rather than only looking at the peak pixel. This is particularly helpful when the PSF has only been sparsely sampled (few pixels per beam) and the source lies near a pixel boundary; in this case, the S/N can be a bit higher than

simply the peak pixel value divided by the off-source rms. All that was a long way of saying: the final sentence of this paragraph should be revised.

###Response

- We have modified the last statement of this paragraph.

20. Line 351: How is the time selection of the de-dispersed visibilities achieved? Is any weighting applied based on the fitted width of the burst, or is a top-hat selection function applied?

###Response

- We used a top-hat selection function.

21. Line 364-372: it is naive to take the statistical image-plane fit uncertainty as the sole source of uncertainty for an individual burst, as there will surely be calibration errors leading to systematic position uncertainties. This likelihood should be acknowledged and estimated (the recent papers on FRB20201124A mostly include reasonable discussions of systematic uncertainties on position estimates). It is then even more naïve to take a simple weighted mean of all of the burst positions (with statistical-only uncertainties) estimated to arrive at a final FRB position and uncertainty. An estimate of systematic uncertainty in the VLA position must be presented at the same time as that final statistical uncertainty.

###Response

Thank you for pointing out this omission (and the implicit disagreement in our numerical reporting). We have now added in words here to describe how the systematic uncertainty for the FRB bursts is the same as the systematic uncertainty for the PRS (because, indeed, they use the same data and are subject to the same systematics). To summarize briefly also here so you don't have to find it in the draft, essentially we do frequent phase-referencing with a small enough interval that any short-timescale variations will be fitted out by our phase calibration. Thus, the short-timescale systematics should reflect the same systematics as the deep image. Additional note: while we didn't write this extra analysis into the draft, we also confirmed that the phase wander was consistent on short timescales by performing short-timescale imaging (imaging segments of 5-30 seconds, such that at least 1-3 sources could be detected at each frequency) and inspecting the position variations of those sources over time (positions were stable and consistent with statistical uncertainty; offsets of the sources were within the range of PRS-quoted systematics). While this could not be as rigorous as the deep image because each field only had 2-3 detectable sources on those timescales, it at least reassured us of consistency for the FRBs.

Your comment also brings up another point you also mentioned elsewhere, which is that we didn't take the full (statistical+systematic) uncertainty into account in the weighted mean. We agree with this assessment and have changed the position calculation and reported error:

- a weighted mean is now calculated for each frequency separately, with the weights scaling with the inverse statistical error of each measurement. The error on each of

those frequencies is then the propagated statistical error, added in quadrature with the systematic error at that frequency (as determined by the PRS analysis).

- The three frequency measurements are then averaged, with the appropriate error propagation (for the mean of three values with error) quoted.

We have accordingly changed the text to reflect this change. Because our FRB position was performed as a three-frequency mean, accordingly it is appropriate (particularly in a comparison between the PRS and the FRB emissions) to perform the same operation---a weighted mean of the positions at the three observing frequencies---for the PRS. The errors are now consistent and the PRS position-fitting description has been changed accordingly to reflect this change. Text throughout, have been updated to reflect these changes.

22. Line 391-393: Why was the PRS fit only with a point source model? If the source size is consistent with being unresolved then this is reasonable, but surely a gaussian fit should also be attempted to place an upper limit on size (at each frequency)?

###Response

We actually took gaussian fits using the CASA imfit task for the PRS at multiple bands (L, S and C bands). The returned results showing that the source is a point source at L and C bands, with the size may as large as (1.4", 0.89") and (0.36", 0.1"), and a component with size of (0.51", 0.14") at S band. We've revised the initial improper expressions.

23. Line 410: Which direction are the systematic offsets? VLA-PanSTARRS or vice versa? Has the correction been applied to the final PRS position that was reported, or not?

###Response

The offsets are obtained by subtraction of PanSTARRS coordinates from VLA coordinates.

We have applied systematic corrections to the PRS positions, thus the position errors are statistical errors added in quadrature with systematic errors. In the Extended Data Figure 1 we applied the systematic corrections of all bands to the PRS positions to give an idea of the consistency of the positions of the bursts and the PRS.

24. Line 433: Why is 0.1" considered a conservative estimate of systematic uncertainty, when the uncertainty in the VLA-PanSTARRS offset is >0.12" at all frequencies? (And indeed, 0.12" is used as the systematic uncertainty in the PRS position in the main text?)

###Response

0.1" is from a former version of VLA data calibration which differs from the VLA burst calibration and we hadn't corrected this value. The conservative estimate should be 0.15". Making use of this value, we calculated the chance coincidence probability as 8×10^{-6} . We have revised the corresponding numbers.

25. Line 438: It is odd that this section is titled "Optical Redshift Determination" when it also includes a lot of photometry.

###Response

We have changed the section title to "Host Galaxy Photometry and Redshift Determination".

26. Line 442: The spectral range of the J band observations is not given.

###Response

Subaru MOIRCS J-band filter cover 1.153 - 1.354 micron. We have added this information in the text.

27. Line 452: “The slit of DBSP was set to cover the FRB optical counterpart” – but the preceding text says that only PanSTARRS was used, and subsequent text says that the host galaxy is not seen in PanSTARRS. What optical counterpart is thus being referred to here?

###Response

We are referring to the optical emission detected by Palomar DBSP. To avoid confusion, we have revised the sentences associated with this statement to:

“The slit of DBSP was set to cover the persistent radio source (PRS) emission at RA = 16:02:04.27; Dec. = -11:17:17.5 detected by VLA in L-band on 22nd July 2020. The PRS location is found to coincide with the location of the burst emission from FRB 190520 within 0.18”.”

28. End of the same sentence, line 454-455: “which later is found to coincide with the location of the pulsation emission from FRB 190520 in 0.18””: this sentence needs revising, as the grammar is incorrect and it is not exactly clear what the authors mean.

###Response

As mentioned in the response to the previous question (27), we have revised the sentence to:

“The slit of DBSP was set to cover the persistent radio source (PRS) emission at RA = 16:02:04.27; Dec. = -11:17:17.5 detected by VLA in L-band on 22nd July 2020. The PRS location is found to coincide with the location of the burst emission from FRB 190520 within 0.18”.”

29. Equation 1 and surrounds: Neither lower-case lambda nor upper case gamma are defined. All symbols used should be completely defined.

###Response

Done.

30. Line 486: The writing in this section is quite poor and makes it hard to understand exactly what has been done. It needs a comprehensive re-write. Line 492 refers to a figure – what figure? What is the contiguous inactivity fraction? (I can guess, but it is not a standard term, and hence should be defined.) What step sizes were used in the period and period derivative, and what is the maximum phase error that could result? Latex errors (italicized units) abound in this section.

###Response

Thanks, we have rewritten this paragraph and described the method more accurately.

31. Line 506: why is there no uncertainty for the MW contributions?

###Response

The uncertainty wasn't stated but it was part of the calculation (which was a Bayesian calculation using uncertainties for the NE2001 and halo contributions to DM along with cosmic variance for the IGM calculation. All of these ingredients are now mentioned explicitly.

32. Line 507: Reference for the chosen value and uncertainty of Halo DM?

###Response

The range for halo DM from the MW is now mentioned (25 to 80 pc/cc) and references are now included.

33. Line 509: The text and references do not actually justify the range quoted for the IGM DM. The quoted reference considers f_{IGM} in the range 0.6 ± 0.1 , not 0.8, and in any case provides no hints as to how cosmic variance was translated into this DM range. This justification should be strengthened. Also, throughout this section (and indeed, throughout the entire paper) confidence intervals should be specified (or if are typically 68%, then this should be stated somewhere).

###Response

The range quoted for the IGM DM is now justified and confidence intervals are stated to be 68% intervals.

34. Equation 3: Half of the symbols used in this equation are not defined.

###Response

Thanks! The symbols are now defined.

35. Line 514: the uncertainties presented for DM_{host} are clearly inaccurate, having presumably neglected the uncertainties on the MW halo (and made no attempt to estimate an uncertainty on the MW contribution). The number of significant digits is also incorrect. At no point in this section is the derivation of the final value of $912 + 69 - 108$ that is quoted in the main text shown. Whatever the number is in the main text should be supported by the analysis here, and it is not. How were the multiple sources of uncertainty combined?

###Response

No, the uncertainties were not neglected. The uncertainties are now itemized to clarify.

36. Line 517: Where does the extinction-corrected H α flux come from? Which spectrum? What was the extinction correction? The caption of extended data figure 4 says that the Palomar spectrum that is plotted was not corrected for slit loss, if it comes from this spectrum, was that correction applied, if so what was the result?

###Response

The extinction-corrected H-alpha flux is from the Keck spectrum. We were using the Balmer decrement line ratio $H\text{-alpha}/H\text{-beta} = 2.8$ in case to estimate the extinction against H-alpha emission. The detailed analysis is included in the associated host galaxy analysis paper that will be submitted soon. The Palomar spectrum is not corrected for the slit loss since we mainly used it to determine the redshift but the H-alpha line is

consistent with the Keck/LRIS result. The Keck LRIS spectrum was not corrected for the slit loss either because the compact host galaxy with a seeing condition of 1.1" is considerably smaller than the slit width of 1.5".

37. Line 518: How was the size of 0.5x0.5" for the host galaxy estimated? At optical wavelengths, the situation is confused – there is enhanced emission at the site of the FRB, but the centroid of the galaxy is quite close by. It is not immediately clear to me from the description of the DBSP observations how much light the slit should be catching from the enhanced emission at the FRB location vs the bulk of the galaxy. This should be clarified.

###Response

The apparent size of the host galaxy mentioned here is an extremely rough estimate based purely on the apparent size of the H-alpha emitting region in the optical images. For the purposes of this analysis (i.e., estimating the H-alpha emission measure and corresponding DM) the uncertainty in the host galaxy dimensions is relatively unimportant. This is because the size of the H-alpha emitting region is probably smaller than the estimate used, which would only serve to increase the H-alpha surface density, emission measure, and DM that are then calculated. This would only strengthen our conclusions that the DM contribution of the host galaxy is large, and that the amount of scattering we would expect from the galaxy if it were in the foreground would be significantly larger than the scattering we observe. We have added a brief note to this effect in this section.

38. Equations 4 and 5: again, many symbols are not defined: epsilon, zeta, T_4, ...

###Response

Done.

39. Line 531: the logical conclusion from the lengthy analysis in this paragraph, that there is a local contribution to DM rather than the host contribution coming solely from the diffuse host galaxy, is not stated explicitly. And no attempt is made to say how well constrained **any** of these parameters are. This lack of rigor is relevant to the scattering analysis later.

###Response

We have modified this paragraph to provide brief motivation for the choice of fiducial values in calculating the H-alpha based DM, and we have indicated the range of possible values both here and in the main text. We also explicitly state the main conclusion of the analysis, which is that the H-alpha EM supports a significant DM contribution from the host galaxy, but it is unclear whether the H-alpha emitting gas can account for the entire DM contribution or whether there is an additional DM contribution from the FRB's local environment.

40. Line 538 (and elsewhere): there is no justification to quote the DM to 4 significant digits.

###Response

Thanks, we update the DM values to 2 significant digits through the whole paper and its uncertainty is 20% which could be referred to comment 31.

41. Equation 6: m is not defined.

###Response

We have defined m under Equation 6. (m is one of the free parameters).

42. Line 544: what does “fold the frequency” mean?

###Response

The word ‘fold’ might cause misunderstanding. We change “fold the frequency” to “Integrate data along the frequency axis”.

43. Line 542-550: As already noted in point 14, surely the simplest assumption possible would be for no time dependence to scattering, and it would make more sense to fit a single scattering time to the entire ensemble of bursts? That would probably give a better estimate of the scattering time, and then also a better estimate of the distribution of widths. Otherwise, the widths of the bursts for which no scattering time was estimated (and which were then fitted by an unscattered gaussian) are very likely to be biased high. Since the main purpose of this analysis is to estimate a scattering time in order to rule out the observed galaxy being a foreground source, obtaining a mean value this way, by fixing it across all bursts, must surely be preferred.

###Response

Please refer to the response to point 14. We do not fix the scattering time across all bursts because we see clear evidence of scattering in some bursts and not in others. Time-variable scattering can be explained by a patchy medium near the source.

44. Equation 7, **again** half the symbols are not defined (considering both this equation and the unlabeled one on line 558).

###Response

Done.

45. Line 568: I agree with the analysis but not the firmness of the conclusions reached. Many other parameters than those mentioned at the end of the paragraph contribute to the estimated scattering time, and no attempt was made to estimate their potential ranges. To be definitive here, the authors should rule out the possibility of a DM from this galaxy (the putative host, which would actually be an intervening galaxy in this scenario) being sufficiently low given the uncertainties, and they do not do so at present. I want to stress that I agree with the likely conclusion, but it needs to be far better justified – or else the possibility needs to be left open (and some estimate made of the confidence level).

###Response

We have modified this portion of the methods to demonstrate a range of possible scattering times based on a range of possible DM contributions (using the H-alpha analysis described earlier in the Methods). It turns out that even for a low DM contribution from an intervening galaxy, the implied scattering will still be extremely large because of the geometric leverage effect. See also our response to Comment #8 for more details.

46. Line 569: nowhere is it stated (and referenced) that almost all FRBs, especially repeaters, are highly linearly polarized – because otherwise, why couldn’t the observed

polarization be due to an intrinsic low linear polarization, rather than depolarization? This should be stated at the outset. For that matter, the observed polarization is not well described. What are the upper limits on linear polarization and circular polarization? This is particularly important if these results are to be referred to in the main text.

###Response

Thanks! We have added another possibility using “The observed polarization could be due to an intrinsic low linear polarization” in the second paragraph of the rotation measure. The upper limits on linear polarization and circular polarization should be 100%.

47. Extended Data Figure 1: Why does this appear out of sequence (before other extended data figures that are referred to earlier in the Methods?). It took me quite some time to realize that the values printed in red are the uncertainty in the *peak* of the plotted distribution, as opposed to some kind of confidence interval. This should be made clear. Also, the fitted distribution for both tau and width seem to have non-zero constant components, which seems like a very strange thing to allow in a gaussian fit to a distribution that should tend to zero at negative values (and positive infinity). What is the value of fitting a gaussian at all, as opposed to simply quoting e.g. the median value? In short, this needs a complete overhaul, but could be improved while also addressing my comments 14 and 43.

###Response

Thanks for your comments. We get rid of the Extended Data Figure 1 from the Method section, and simply quote the mean values for both the scattering timescale and the pulse width.

48. Extended Data Table 4: nowhere is it made clear where the systematic uncertainties in the final columns have come from. By cross-referencing, it is apparent that they are simply the overall systematic uncertainty estimated for the PRS from the stacked deep images at each frequency. But this is a lower bound to the systematic uncertainty for a given burst, since it assumes no time dependence at all – the radio positions used to estimate these offsets and corresponding uncertainties have been obtained from an image averaged over hours of integration (spread across days in the case of the L band bursts). If there are any residual calibration errors at the time of a given burst, they will lead to a systematic position offset that goes unappreciated here. If the authors provided more details of the solution interval and S/N threshold of their gain calibration step, this might provide some reassurance that the potential maximum size of any time dependent systematics is not too large.

A better approach would be to select a time range centered on the burst and use positions of the continuum sources from this subset of the data to estimate the systematic offset in each case. If that is deemed too much work, then alternative methods to take this potential underestimate into account should be used – for instance, by looking at the chi squared of the final weighted mean. Doing this, and then taking the correlation between systematic position uncertainty for bursts at the same frequency, would be a more correct way to derive a final FRB position and uncertainty.

###Response

Thanks, Please see our response to your comment #21, in which we believe we have addressed this point. Due to the long table could not be put in the main text, we mv Table 1 to supplementary Table 1, and the ED table 4 now is ED Table 3.

49. Extended Data Table 5: I'm curious how an apparently 20 sigma burst (burst C5) can be identified to have a width of ~0.1 ms with an uncertainty of 70 microsecond, given the 10ms sampling of realfast? All of the other bursts have a width uncertainty of 1-2ms, as I would expect. Perhaps this is a typo? Other numbers in these tables are curious: e.g. burst L1 has fluence 4(6) Jy-ms. Why is the fractional uncertainty >100%? Why is the localization of burst S1 more than twice as precise than any of the other S band bursts, despite burst S2 and S3 having comparable S/N? I see that it is much wider-band than the other two, but that doesn't really matter given that the image S/N is the final result of the width+fluence+bandwidth of the burst combined. Looking at the beam sizes and the S/N values quoted, it seems more like the statistical uncertainty is too large for the other bursts rather than too small for burst S1. This might indicate that the fitted source size was >> the synthesized beam size in those cases, which would be indicative of a concerning systematic error (as the FRB itself must be a point source, and hence should have a size approximately equal to the synthesized beam size).

###Response

We thank the referee for pointing this out. We have edited the values in the table and added more description to this section to explain the burst fitting process and the caveats to this analysis.

For CASA imaging, we only used the frequencies that contain burst signals to form the image. We then use that image to determine the burst position. Therefore, the beam size for each image will change based on the signal peak frequency of the respective FRB. This can lead to different size of fitting errors even for similar S/N bursts.

50. Reference 41: why is this the arxiv reference rather than the published ApJ article?

###Response

Because the article hasn't been published yet and is still under review.

Response for 2nd referee

Referee #2 (Remarks to the Author):

In this paper, the authors present the discovery of a repeating fast radio burst (FRB) source, whose host galaxy they identify. This source, FRB 190520, is similar to the first-known repeating FRB source, FRB 121102, in that it is hosted in a dwarf galaxy and has an associated compact, persistent radio source (PRS).

The most novel aspect of the paper is that the authors show – assuming that the host galaxy association is correct - that the local electron density (DM) of the FRB 190520 must be quite large compared to other known repeating FRBs.

They argue that this may mean that it is younger compared to other sources. That may be, but I think it is also possible that the source is in a galactic-centre-like environment, near an accreting massive black hole (one of the hypotheses for FRB 121102, and as we see for the Galactic centre magnetar PSR J1745-2900 in our own Milky Way).

The authors also claim that the large local DM suggests that caution is needed in using FRB DMs to estimate redshift. Given that only a few percent of FRBs are known to be repeaters, and given that several repeaters have been shown to have much lower local DM contributions, I think this claim isn't well substantiated. In other words, does this one exceptional source really suggest that DM is a poor proxy for distance for FRBs in general?

###Response

Thank you very much, Jason, for the careful reading and many good suggestions! We have moderated the statement. The emphasis was now shifted to repeaters and PRS. We still mentioned 'caution' as it is still an outstanding question whether all/most FRBs may repeat.

Overall, I found the paper to be quite interesting (great that there is now an FRB 121102 twin!) and potentially suitable for Nature, but I would like the authors to address the previous comments, as well as those below.

Sincerely,

Jason Hessels
University of Amsterdam & ASTRON

---Other general comments:

1. - Is the source actually exceptionally active compared to other well-studied repeaters? The claim of a link between “high activity” and the presence of a PRS doesn’t fit with observations of FRB 20201124A.

###Response

You are right. Also it is hard to normalize the burst rate. We thus removed ‘highly’ or similar claims throughout the texts and only refer to 121102 and 190520 as active repeaters. The PRS connection is now discussed in the context of complex environments and possible trait for identifying large local DM.

2. - The paper should discuss how some repeaters *don’t* have a PRS, down to very low luminosity limits (e.g. FRB 20180916B, Marcote et al. 2020 and FRB 20200120E, Kirsten et al. 2021).

###Response

Discussion added in the 2nd to last paragraph. The occurrence of persistent radio sources like that for FRB 20121102A is not well understood. Two nearby burst sources (FRB 20180916B and 20200120E) do not have associated PRSs down to low luminosity limits. This may indicate that a luminous PRS requires a special environment around the source or it might be an evolutionary effect.

3. - What are the actual constraints on the properties of the PRS (size and offset from FRB source and host galaxy)? How does this quantitatively compare to FRB 121102 (Chatterjee et al. 2017, Tendulkar et al. 2017, Bassa et al. 2017, Marcote et al. 2017)?

###Response

As for the PRS size: The best constraint is from the VLA C-band observation. Using 0.36” as the conservative size and adopting the redshift as 0.241, we calculated the conservative upper limit of the projected size of the source as ~ 1680 pc. This value is two orders of magnitude larger than the constraint of FRB 121102 size (~ 8 pc) since for PRS of FRB121102 the best constraint is from VLBA, which gives much better resolution. Since the VLBA analysis of FRB 190520 is ongoing, we intend to include a comparison of PRS sizes in follow-up paper(s). A preliminary VLBA size of PRS for FRB 190520 is 28.5 pc, more extended than PRS for FRB 121102. As for the offset between the PRS and the FRB source. Using the average burst and PRS position at the S band, we calculated the offset to be about 0.126”, which translates to a projected size of 590 pc, slightly higher than the size of the PRS for FRB 121102, which is 590 pc (0.1”). We will discuss the offset between the PRS and host galaxy in a follow-up paper(s).

4. -The paper should describe the burst properties (e.g. wait times distribution, energy distribution, etc.) in order to compare with other repeaters. For example, are the bursts of the same average energy compared to FRB 121102?- The apparent lack of DM variations should be discussed, since one might expect variations if the source is exceptionally young

###Response

Thanks, We added the FRB 190520B Energy distribution in the paper. The number of calibrated pulses (79) in this work is only 4.5% of what is available for 121102. The 90% detection completeness threshold here is 50% higher than that of the FRB 121102 due to the larger distance. We are continuing monitoring the source and hopefully to accumulate a much more substantial pulse set.

- There is not much discussion of the host galaxy properties. This is only the 2nd known dwarf host for an FRB, but repeaters have also been found in a wide range of galaxies. How does that fit with the author's interpretation?

###Response

The host galaxy of FRB 190520 is the 2nd known dwarf galaxy host of a repeating FRB and bears resemblance to that of FRB 121102 in several aspects, including an above average star formation rate, the collocation between the PRS and the burst, and the off-galaxy-center location of the burst.

- The chance coincidence with the host galaxy is not discussed in proper detail (there is only one sentence in the paper on this, which simply states that the chance association probability is low, without describing any of the assumptions or giving exact numbers).

###Response

We consider that the properties of the host galaxy are not the focus of this discovery paper. In addition, considering the page limit of the paper, we reserve the discussions of the host galaxy in a subsequent paper which is under preparation.

Given that many of the results depend on a robust host galaxy association, this deserves a proper treatment.- Some of the analyses - e.g. short and long-term periodicity, scintillation bandwidth, etc. - could have a related figure in the Methods

###Response

Thanks, we added these analysis and figures in the Methods.

---Detailed comments (some are minor typos!):

1. Title:- Line 1: "A highly active repeating fast radio burst in a complex local environment"

- I would suggest that the title focuses on the most novel aspect of the paper, which is the high local electron density (DM). The source activity is not unprecedented, and "complex local environment" is vague.

###Response

Thanks for pointing this out! We change the title to:

"A repeating FRB in a dense environment with a compact persistent radio source"

2. Summary paragraph:- Line 41:

- "central engine" is jargon and may not be clear to a wider readership

###Response

After rounds of discussion among Cols, we find that 'central engine' seems to be one of the less specialized descriptions. So we hope that 'central engine' is acceptable? We are open to suggestions, of course.

3. - Given current FRB observations, I would tend to talk about multiple possible source types for their origin.

###Response

Thanks, we add “the diverse origin” in the first sentence.

4. - Line 42:

- “the dispersion sweep of FRBs provides”: awkward mixing of singular and plural in this sentence. There are other examples of this in the Summary paragraph and elsewhere in the paper.

###Response

Done.

We change

“...the dispersion sweep of FRBs provides...”

to

“...the dispersion sweeps of FRBs provide...”

5. - Line 43:

- The dispersive sweep is also probing the ionised material in the host galaxy, as well as the Milky Way.

###Response

Done.

We change

“...provide a unique probe of its environment and the ionized baryon content of the intergalactic medium...”

to

“...provide a unique probe of its environment and the ionized baryon content of the interacted medium...”

6. Line 43:

- “Active repeaters has been shown”: *some* repeaters have shown this, while other repeaters have not.

###Response

Thanks , Done.

7. - Line 47:

- “host galaxy of high star formation”: high *specific* star formation, I assume.

###Response

We have corrected the description to “high specific star formation”.

8. -Line 50:

- “suggesting caution in inferring redshifts for FRBs without accurate host galaxy identifications”: but only a few percent of FRBs have been shown to be repeaters. So the paper does not present evidence that this is true for a significant fraction of FRBs.

###Response

We think 'caution' is a modest statement. In the main text, we added explicit statements about 121102 and 190520 being outliers that may be identified.

9. Line 53:

- "may point to a distinctive origin or an earlier evolutionary stage for highly active repeating FRBs": there hasn't been any quantification yet about whether this source is actually more active than other well-studied repeaters. That's necessary to justify this statement.

###Response

We remove the word 'highly' in this sentence and throughout the text. The normalization of burst rate has been extensively investigated. However, at the moment, no valuable conclusion can be articulated. For example, the ASKAP bursts, due to the small antenna size, are so much brighter than other samples. If one tries to normalize the burst rate of ASKAP to the FAST detection threshold, there is simply no constraint. Such discrepancies in observing conditions (including cadence) mean that essentially one should only compare the apparent burst rates obtained with the same instrument in the same mode.

Main text:

10. Line 59:

- "detected 75 bursts in 18.5 hrs": FRB 121102 has sometimes been seen to produce as many bursts in ~1 hour of observations. So I would not say that FRB 190520 is exceptionally active.

###Response

Agreed. See the response above.

11. - Line 59:

- "a mean pulse dispersion measure (DM) of $1202 \pm 10 \text{ pc cm}^{-3}$ ": why is the uncertainty on the DM so large?

###Response

Thanks for pointing out this. We are using the method described from Hessels et al. 2019.

We take DM search using the DM range from 1190 to 1210 pc cm^{-3} and the DM step is 0.1 pc cm^{-3} . Then we take the Gaussian fit for the DM range from the whole DM range and take the FWHM. That caused a really large range of DM variation. We revised this problem by using the code from (Seymour et al. 2019, https://github.com/danielemichilli/DM_phase).

Due to the imperfect results of some of the bursts (especially for the faint pulses), we picked the best-fit DM burst (usually for the brightest one) of each day, then use its DM and DM range for that observation day.

Fig. Left is the result of our DM search code in the previous version. Right is the revised result of DM search from the (Seymour et al. 2019).

12. Line 72:

- “Using averaged flux density at each”  “Using the average flux density of each”

###Response

Done.

13. Line 73:

- “we find a PRS flux density spectrum can”  “we find that the PRS flux density spectrum can”

###Response

Done.

14. Line 74:

- “index -0.41”  “index -0.41 ” (negative sign)

###Response

Thanks ! Done.

15. Line 78:

- “Given the measured offset of the FRB from the galaxy”: what is the measured offset found to be?

###Response

The offset is 1.3”. We have included this number in the text.

16. Line 79:

- “we estimate a chance coincidence probability”: I assume that there is a more detailed demonstration presented in Methods. That should be cited here.

###Response

Thanks for pointing it out. We have added the details of chance coincidence probability to the methods section and cited that here in the main text.

17. Line 79:

- “a chance coincidence probability of less than 1%”: why not quote the actual value?

###Response

Thanks, the coincidence probability is 0.8%. We have corrected the values in the text.

18. Line 79:

- “supporting J160204.31–111718.5’s being the host galaxy of FRB 190520”

“supporting the claim that J160204.31–111718.5 is the host galaxy of FRB 190520”

###Response

Thanks ! Done.

19. Line 84:

- “that revealed the”  “, which revealed the”

Done.

20. Line 84:

- “to be $z = 0.241$ ”: include uncertainties

###Response

We have added the uncertainty 0.001 into the text.

21. - Line 92:

- “a relatively high star-formation rate for its stellar mass”: the Summary paragraph simply says “high”. What does “relatively” mean, more quantitatively? Is the rate somewhat above average, in the top 10%, etc., compared to other dwarf galaxies?

###Response

The host galaxy of FRB190520 with stellar mass of $6 \times 10^8 M_{\text{sun}}$ (or $\sim 10^{8.8} M_{\text{sun}}$) and star formation rate of 0.41 M_{sun}/yr (or $10^{-0.39} M_{\text{sun}}/\text{yr}$) is significantly above the the star formation main sequence of SDSS galaxies (see Figure below from Calabrò et al. 2017). However, it is difficult to give a quantitative description of the deviation since there are no SDSS galaxies with similar stellar mass at the redshift $0.2 < z < 0.3$ where the FRB190520 host is at. We note that the systematic star formation rate of dwarf galaxies at the stellar mass of 10^8 - $10^9 M_{\text{sun}}$ at the redshift $0.2 < z < 0.3$ might suffer from the low completeness issue.

To clarify, we rephrase the statement to “a relatively high star-formation rate for its stellar mass compared with local SDSS galaxies”.

The SFR- M_{star} diagram from Figure 11 of Calabrò et al. (2017, A&A, 601, A95). The orange data points at $M_{\text{star}} \sim 10^8 - 10^9$ are from zCOSMOS but at a higher ($z > 0.5$) redshift than that of the host of FRB190520.

22. - Line 92:

- “At the luminosity distance implied by the redshift”: would be better to also state what that distance is.

###Response

We have added the number of luminosity distance of 1218 Mpc in the text.

23. - Line 95:

- “of a FRB source”  “of an FRB source”

Done

24. - Line 98:

- “For nominal DM contributions from the Milky Way (100 pc cm⁻³ for MW disk and MW halo)”: is this what electron density models would predict in this direction? Would be good to state that this source is well off the Galactic plane.

###Response

Thanks, The Galactic latitude is around 30 degrees. We have revised the text about the DM contribution from the MW disk and MW halo in the new PDF version Line 555-559.

25. - Line 99:

- “and host galaxy (50 pc cm⁻³)”: why is the host galaxy contribution assumed to be smaller than that from the Milky Way?

###Response

The purpose of this text item was to in fact demonstrate that assumption of a small host-galaxy DM yields a very large, inconsistent redshift estimate. One reason for doing this is that papers in the literature do adopt the assumed value of 50 pc/cc for host galaxies, which we think is unrealistic. The comment’s comparison with the Milky Way value is not particularly relevant, especially since we used a value of 60 pc/cc +/- 20% error. In our

update, we more conservatively use $\pm 40\%$ for the MW contribution and propagate that uncertainty into the uncertainty for the host-galaxy DM that best describes all the data.

26. - Line 99:

- “and also assuming baryon fractions of 0.6 to 1 for the ionized IGM”: citation needed.

###Response

Done.

27. Line 117:

- “observed scattering time of 10.8 ms”: uncertainty should be included, as well as a reference to Methods for more detail on how this was determined.

###Response

Thanks, we add the uncertainty and update the scatter timescale to 9.8 ± 2 ms.

28. - Line 122:

-“FRB 190520 shows that the distribution of DM_{host} values can have a long tail, which adds considerable variance to estimates for the IGM.”: this is only true if a significant number of FRBs are like this source. Given that $>\sim 95\%$ of FRBs appear to be non-repeaters, it seems to me that one can still accurately estimate the IGM contribution in the vast majority of cases - unless non-repeaters are, in the future, shown to often inhabit dense local environments.

###Response

We have slightly modified the sentence to emphasize that a subset of FRBs like FRB 190520, that is those with large DM_{host}, *may* add considerable variance to estimates for the IGM. It is unclear at this point how large (or small) that variance is, but a larger sample of localized FRBs will significantly improve our knowledge of the DM_{host} distribution, and FRB 190520 demonstrates that our knowledge of that DM_{host} distribution *may* be more incomplete than previously thought.

Also, at the time of writing, only 15 FRBs have host-galaxy redshift measurements and one of those (190520B) has a large host DM and others have less dramatic but still not insignificant host-galaxy DMs.

29. - Line 128:

- “Another repeating source, FRB 201124A, was also associated with persistent radio emission. However, through...”: I don’t think that it’s important to discuss this in the Main part of the paper, since the upshot is that FRB 20201124A (note correct name) doesn’t have a PRS like that of FRB 121102 or FRB 190520. Much more

important, is to comment on the fact that other repeaters have **no** PRS, down to very constraining limits. In particular, FRB 20180916B’s potential PRS is constrained to be $\sim 300\times$ less luminous than FRB 121102 (Marcote et al. 2020), and the odd FRB 20200120E in a globular cluster provides even tighter constraints on PRS luminosity because of its exceptional proximity (Kirsten et al. 2021). Actually, it’s also important to discuss that FRB 20201124A doesn’t have a compact persistent radio counterpart (the preamble explanation isn’t necessary in the Main part of the paper though) because that

FRB has been (at times!) extremely active. That doesn't fit with the idea that more active sources are associated with PRSs.

###Response

We agree that the burst rate alone is not sufficient to predict the presence of a PRS. We have added some qualifications to the description of this connection. New FRB discoveries are also cited appropriately.

30. - Line 132:

- "the PRS luminosity would imply a star-formation rate of $\sim 10 M_{\odot} \text{ yr}^{-1}$ ": how would that compare to what has been observed from dwarf galaxies? Is it known to be possible to have such a high rate in a dwarf?

###Response

Dwarf galaxies with a $M_{\text{star}} < 10^9 M_{\text{sun}}$ have a star formation rate that is much less than that implied by the PRS radio luminosity. For host galaxies like that of FRB 190520, the largest known SFR is roughly $2 M_{\text{sun}}/\text{yr}$ (e.g., <https://ui.adsabs.harvard.edu/abs/2004MNRAS.351.1151B/abstract>).

31. - Line 133:

- "Given the extreme PRS luminosity, its unresolved structure in VLA observations": it would be good to state what the constraints on the physical size (and offset) actually are, and to compare these to FRB 121102 (Chatterjee et al. 2017; Marcote et al. 2017).

###Response

The best constraint to the source's deconvolved size from the gaussian fits is $0.36'' \times 0.1''$ from the C band observation. Using $0.36''$ as the conservative size and adopting the redshift as 0.241, we calculate the conservative upper limit of the projected size of the source as ~ 1680 pc. This value is two orders of magnitude larger than constraints of FRB 121102 size (~ 8 pc) since for PRS of FRB121102 the best constraint is from VLBA, which gives much better resolution. Since the VLBA analysis of FRB 190520 is ongoing, we intend to include a comparison of PRS sizes in follow-up paper(s). A preliminary VLBA size of PRS for FRB 190520 is 28.5 pc, ~ 3 times larger than PRS for FRB 121102.

32. - Line 134:

- "and its offset from the center of the optical emission of the host galaxy": I think it's important to note, however, that FRB 121102 and its persistent radio source are very close to a dominant knot of star formation (Bassa et al. 2017) in its dwarf host galaxy (though in HST observations ever so slightly offset from it...).

###Response

This story is clear for FRB 121102. Figure 2 shows how this may also be true for FRB 190520. The optical counterpart of FRB 190520 is extended and has a color gradient that may indicate enhanced star formation near the FRB location. More detailed analysis will appear in a future publication, so we prefer the current, more restrained, description.

33. - Line 136:

- “as found for FRB 1211022.“: here I would also cite Marcote et al. (2017), since that provides the most precise quantification of the physical size of FRB 121102's PRS, as well as it's maximum possible offset from the FRB source itself.

Done

34. - Line 137:

- “Burst repetition and spectral structure have been used to argue”: also Faraday rotation measure (e.g. Michilli et al. 2018) and time-frequency structure (e.g. Hessels et al. 2019) have been used to distinguish repeaters from apparent non-repeaters.

###Response

Thanks, we added the reference and changed it to:

“Various methods have been used to argue that repeating and non-repeating FRBs comprise different subclasses...”

35. - Line 139:

- “The observed burst properties are subject to observational biases”: this seems to imply that the current study isn't also affected by observational biases, which is of course not true. Rephrase.

###Response

Done. Instead of mentioning biases, we now simply state that “PRS and DM_host reflect different aspects”.

36. - Line 139:

- “but PRS emission and DM_host reflect different aspects of the FRB environment.“: difference between repeaters and apparent non-repeaters? Difference compared to the aforementioned burst properties? I'm unclear on what point this short paragraph is trying to make.

###Response

Good point. We revise the statement to “While the observed burst rate and spectral structures can be temporal due to various mechanisms, PRS emission and DM_host reflect more persistent aspects of the FRB environment and thus may be more reliable tracers of any putative subclasses.”

37. - Line 143:

- “two FRBs associated with PRSs are among the most active”: is this a robust statement? FRB 121102 has still only shown one burst in CHIME/FRB observations, whereas FRB 20180916B has been detected many dozens of times by CHIME/FRB and shows no PRS. FRB 20201124A has also been extremely active (at times!) at 1.4 GHz, but shows no PRS.

###Response

The ‘most’ was removed.

38. - Line 144:

- “and have large DM_host values”: the DM_host of FRB 121102 hasn't been mentioned yet (see Tendulkar et al. 2017 and Bassa et al. 2017). The FRB 121102 constraint of 55

$\text{cm}^{-3} < \text{DM}_{\text{host}} < 225 \text{ pc cm}^{-3}$ isn't necessarily "large", like is being claimed for FRB 190520.

###Response

We agree that the FRB 121102 host-galaxy DM range is not "large" in comparison with that for 190520 and we have rephrased text accordingly.

39. - Line 153:

- "The discovery of FRB 190520 and its high similarity to FRB 121102 demonstrate that some FRBs have very large local DM and PRS counterparts.": I'm surprised that the main part of the paper hasn't said anything explicit about the extremely high and variable RM of FRB 121102 and how FRB 190520 compares.

###Response

Thanks! It is really a good idea to compare the RM with FRB 121102 and FRB 190520. Unfortunately, we did not detect any reliable RM. Therefore, we do not discuss RM in the main text.

40. Figure 1:- What time and frequency resolutions are being used for plotting?- What DM is being used for plotting?- Indicate that some frequency channels have been removed to excise radio frequency interference. Also, it is best practice (I think) to mark these with ticks at the side of the plot.- Is the color scaling of the dynamic spectra linear? Any clipping at the low/high end of the value distribution? Why the blue color instead of greyscale?- Indicate the typical observation durations here.

###Response

Thanks for the comments. The caption is revised more clearly now. The time and frequency resolutions are normalized to 0.786 ms and 3.91 MHz, respectively. The DM is taking the value from the ED tabel 1. The RFI channels are labeled by the red patches. The color scaling is changed to linear now. All the information is added to the caption.

41. Figure 2:- Indicate what is causing the bright artefacts in the optical images.

- Could be useful to have a zoom-in at the source position in the optical images.

###Response

The bright artifacts in the figure 2 are caused by the bright star in the west of the FRB190520B.

42. Figure 3:- "The expected DM contribution of the intergalactic medium (orange line) is": citation to Macquart et al. 2020 needed.- Mark FRB 121102 in this diagram.

###Response

Thanks, Done.

Author contributions:

43. - Line 259:

- "Energy"  "energy"

Done.

44. - Line 261:

- "Parks observations"  "Parkes observations"

Sorry for the mistake, Done.

Methods:1 Observations:

45. - Line 271:

- "In this first discovery observation, 3 bursts were detected in 10 seconds, and another burst was detected 20 seconds later.": the paper would benefit from describing some basic statistics of the bursts, like wait time distribution and energy distribution. This is to inform the comparison to FRB 121102 and other repeaters.

###Response

Thanks, The energy distribution and fluence-width distribution have been added. Please see the response of general comment 4.

46. - Line 274:

- "follow-up observations were performed with FAST on April 25th and May 22nd": what year?

###Response

Thanks, They are in 2020. We have changed:

" follow-up observations were performed with FAST on April 25th and May 22nd"
to

"follow-up observations were performed with FAST on April 25th and May 22nd in 2020"

47. Line 276:

- "using the ~ 100 mas localization from VLA": point to the relevant section that describes this.

###Response

Thanks, The referred method part has been added.

48. - Line 279:

- "monitor observation"  "monitoring observation"

Done

49. - Line 281:

- "has been transformed to the arrival time at the solar system barycentre (SSB) at 1.5GHz": using what DM(s)?

Due to the long table could not be put in the main text, we mv Table 1 to supplementary Table 1, and the sequence is changed.

These are using DMs from supplementary Table 1. The text has been revised.

50. - Line 284:

- "The sub-pulse is recognized if the profile peak does not fall behind the noise baseline.": I'm not sure what this means. Needs to be explained more clearly.

###Response

Thanks! We change the term to "a sub pulse". Since there is no standard pulse profile (nor average) for repeaters yet, the use of a pulsar terminology is empirical. Currently, when the 'bridge' between the two closely-spaced-in-time peaks drops more than 5 sigma below

the higher peak, we consider them to be two bursts. Otherwise, they are considered structures within one event (sub pulses).

51. - Line 287:

- "Radio Frequency Interference": don't capitalise.

Done

52. - Line 300:

- "stokes I"  "Stokes I"

Done

53. - Line 300:

- "and the": missing space before "and"

Done

54. - Line 300:

- "the pulse width is adapted by a boxcar in the search": reword, this is an awkward and unclear description.

###Response

We revised the "the pulse width is adapted by a boxcar in the search" to

"we matched the pulse width by a boxcar search"

55. - Line 303:

- "DM range": missing space after "range"

Done

56. - Line 304:

- "from central beam"  "from the central beam"

Done

57. - Line 305:

- "stokes parameters"  "Stokes parameters"

Done

58. - Line 318:

- "-11"  "\$-11\$"

Done

59. - Line 322:

- "detail in"  "detail in Refs."

Done

60. - Line 332:

- "goes through"  "go through" (because "candidates" is plural)

Done

61. - Line 336:

- "etc"  "etc."

Done

62. - Line 336:

- "section 2.4 of"  "Section 2.4 of Ref."

Done

63. - Line 338:

- “see section 2 of”  “see Section 2 of Ref.”

Done

2 Localization of bursts:

64. - Line 357:

- “J1558-1409”  “J1558 $\text{\$}$ - $\text{\$}$ 1409”

Done

65. - Line 367:

- “to signal to noise ratio”  “to S/N” (for consistency with rest of paper)

Done

66. - Line 371:

- “We estimate statistical error to be”: the scatter in positions is ~3-4 larger than this. Would be good to comment on that.

###Response

Based on comments from the other referee, we have now changed how we discuss and quote localization errors on the FRB, and the numbers are now larger, and consistent throughout the draft.

67. - Line 371:

- “-11”  “ $\text{\$}$ -11 $\text{\$}$ ”

Done

68. - Line 371:

- “0.023”[“]: would be better to use “ $0.023^{\{\prime\prime\}}$ ” (here and elsewhere in the paper where arc-minute/second is meant)

Thanks, All the symbols have been corrected.

69. - Line 374:

- “discussion of”  “discussion of Ref.”

- “been described in”  “been described in Ref.”

Done

70. - Line 374:

- “We model the pulse profile”: is the fitted DM from maximizing the peak S/N? Would be good to comment on what these DMs mean compared to the analysis method used for the FAST bursts.

The FAST burst DMs show little-to-know scatter between epochs when the structure-maximizing assumption/technique is used.

###Response

-- Due to the poor time resolution of VLA observations (10ms) as compared to the pulse width, all of the VLA detected bursts lie only in 1 to 2 time samples. Therefore, any temporal structure in these bursts would have been resolved, making it impossible to apply the structure-maximizing process on these bursts. We also verified it visually. We therefore use a single component model, and fitted both the profile and spectra using a Gaussian. We have added text in this section to discuss this.

71. - Line 376:

- “Following”  “Following Ref.”

Done

3 Persistent Radio Source:

72. - Line 383:

- "The VLA visibilities with 3 or 5 s sampling time were saved...": I would clarify to the reader that the data around bursts are saved at high time resolution (~10ms), and that the entire observing span (tens of minutes / hours) are in parallel saved at this lower time resolution.

###Response

- We have modified the text to explain this.

73. - Line 386:

- "J1558-1409"  "J1558-1409"

Done

74. - Line 388:

- "its Stokes I"  "its Stokes I data"

Done

75. - Line 394:

- "The VLA campaign obtained two-epochs": in this sub-section, I think it would make more sense to first describe the observations and thereafter the analysis, as opposed to the opposite order that's currently used.

###Response

- We agree with the reviewer, and have modified the text accordingly.

76. - Line 414:

- "We report the PRS coordinates based on the measurements at 3 GHz": and what is that position?

###Response

- We have added the PRS position measured at 3GHz in this statement.

77. - Line 419:

- "Power-Law(PL)"  "Power-Law (PL)"

Done

78. - Line 420:

- "-0.41"  "\$-0.41\$"

Done

79. - Line 421:

- "image, We"  "image, we"

Done

80. - Line 422:

- "sources , including"  "sources, including"

Done

81. - Line 423:

- “with a flux density higher than 260 μJy ”: make it clear to the reader why this particular flux density is relevant.

###Response

The reason to choose this value as the threshold is because each of the other 7 sources has a flux density at L band higher than the PRS flux, which is 260 μJy .

4 Optical Redshift Determination:

82. - Line 442:

- “of FRB 190520 field”  “of the FRB 190520 field”

Done

83. - Line 443:

- “August 05th”  “August 5th”

Done

84. - Line 454:

- “which later is found to coincide with the location of the pulsation emission from FRB 190520 in 0.18”.“: this is confusingly worded.

###Response

We have reworded the sentence:

“The PRS location later is found to coincide with the location of the burst emission from FRB 190520 within 0.18”.

85. - Line 455:

- “pulsation emission”  “burst emission”

Done

86. Line 460:

- “in to the slit”  “into the slit”

Done

87. - Line 462:

- “reduction on other”  “reduction of other”

- “instrumentation effects”  “instrumental effects”

Done

88. - Line 467:

- “The corresponding redshift derived based on these two spectral lines is $z = 0.241$.”: with what uncertainty?

###Response

We have added the uncertainty of 0.001 in the text.

89. - Line 473:

- “indicating the extended R'-band structure has the same redshift of $z = 0.241$ ”: what is the actual measurement and uncertainty in this additional analysis?

###Response

We do not see any detectable redshift shift across the slit larger than the redshift uncertainty of $dz = 0.001$.

5 FAST Burst Sample Analysis:

90. Line 484:

- “mJy · ms”  “mJy ms”

Done

91. - Line 485:

- “for excluding waiting time”  “when excluding waiting times”

Done

92. - Line 486:

“A period search was conducted from the total 75 bursts from FAST.”: this is confusing because Table 1 reports 79 bursts.

###Response

Sorry for the misleading. The previous 4 bursts are detected at the drift scan survey. The latter 75 bursts are from tracking mode. We already included the total 79 bursts for the period search. Also, due to the limit of long table, we put the table 1 to supplementary table 1.

93. - Line 491:

- ‘by traversing the period (P) among’  “by searching for period (P) from”

Done

94. - Line 492:

- “Almost all values in this figure are less than 0.4.”: what “figure” is being referred to? Also, why is 0.4 the threshold? This is confusing.

###Response

Thanks, we have rewritten this paragraph and described the method more accurately. 0.4 is the upper limit value of the actual contiguous inactive phase segment obtained by folding the ToAs. In this case, when the bursts occupy 60% of the phase window in one period, we think it is hard to say that this is a true period pattern.

95. - Line 494:

- “range of period (P) among”  “range of period (P) from”

Done

96. - Line 495:

- “2 – 365 d”: “d” shouldn’t be italicise here (or later in the same sentence)

Done

97. - Line 495:

- “and period derivative (Pdot) among”  “and period derivative (Pdot) from”

Done

98. - Line 496:

- “the mjds”  “the MJDs”

Done

99. - Line 498:

- “observing session mjds”  “observing session MJDs”

Done

100. - Line 510:

- "discussion in"  "discussion in Ref."

Done

101. - Line 513:

- "The averaged DM_obs from all bursts is 1202 ± 10 pc cm⁻³": the paper should be more clear about whether the 10 pc cm⁻³ uncertainty here is due to measurement uncertainty or scatter in the measurements.

###Response

Thanks, The error has been corrected using the method from (Seymour et al. 2019), also see the response to comment 11 in the main text.

102. - Line 514:

- " 921.1 ± 10 "  " 921 ± 10 " (overquoting significant digits)

Done

103. - Line 515:

- "for NE2001"  " for NE2001" (missing space before "for")

Done

104. - Line 515:

- "for NE2001 and YMW16 model separately"  "for the NE2001 and YMW16 models, respectively"

Done

105. - Line 525:

- "inferred value of DMh"  "inferred value of DM_host" (for consistency with the rest of the paper; use the same name for the same quantity throughout)

Done

106. - Line 536:

- "Using Equation": be consistent in use of "Eq." vs. "Equation"

Done

107. - Line 542:

- "Among the detected 79 pulses of FRB 190520, 28 pulses show frequency dependent temporal width that is consistent with scattering.": is it also potentially related to an unresolved time-frequency drift (sad-trombone effect)? What is the evidence that it must be solely due to scattering?

###Response

Thanks for the question, please see the response to comments 115.

108. - Line 544:

- "we first fold the frequency": unclear what this means.

###Response

Thanks for pointing out this, it is a misleading description. We change it to: "we first integrate data along the frequency axis"

109. - Line 547:

- "The upper panels of Figure 1"  "The upper panels of Figure ED1"

###Response

Thanks! Due to the label name is "Extended data| Figure 1",

We change the description to:

"The upper panels of extended data (ED) Figure 1" .

All the labels of referred figures have been revised to this format.

110. - Line 581:

- "Such a large RM is even larger than that of FRB 121102": the text here should be more clear that it's also possible that the source is intrinsically depolarized at low frequencies, or that generalised Faraday rotation is prohibiting detection of an RM. I think that's just as plausible a hypothesis as intra-channel Faraday rotation and a large RM.

###Response

Thanks! We have made it more clear by adding other possibilities like "The observed polarization could be due to an intrinsic low linear polarization".

111. - Line 582:

- "Large RM also indicates that the FRB 121102 could be very young": not necessarily. It has also been hypothesised that the source is in a galactic-centre-like environment, in the vicinity of an accreting massive black hole (which in turn is the PRS). The Galactic centre magnetar in our galaxy has a high RM, but we don't have reason to believe that that is because it is very young compared to other magnetars.

###Response

Thanks for pointing it out. To make the text more clear, we have removed this discussion.

112. - Line 583:

- "similarly, FRB 190520 could also be very young": quantify.

###Response

Following the previous point, to make the text more clear, we have removed this discussion.

113. - Line 585:

- The references in Main go up to and including #31, so the first reference in Methods should be #32.

Done

114. - Line 626:

- Ref. 50 also appears as Ref. 4. Double check that there aren't more redundancies between the two reference lists in the Main and Methods sections.

Thanks for pointing it out, they are redundant. Ref. 50 has been removed.

Extended Data Figure

115. Figure ED1:

- Given that the inferred scattering timescales are comparable to the burst widths, how can one know that this isn't actually scattering, but rather due to an unresolved time-frequency drift (sad trombone effect)? Can the authors provide some figures of dynamic spectra and fits that support this interpretation and disfavour (or rule out) time-frequency drift?

###Response

A figure showing examples of bursts both with and without evidence of pulse broadening has been added to the Extended Data (see ED Figure 10), and a brief reference to the figure has also been added to the Methods section on scattering. The main way we distinguish between pulse broadening and time-frequency drift is by examining the frequency and time dependence of the burst widths, and by considering evidence of unresolved burst components in spectra such as the one shown in the bottom left panel of ED Figure 2.

116. Figure ED2:- No comments.

117. Figure ED3:

- "corresponding radio flux are shown"  "corresponding radio flux densities are shown"

Done

- "the flux error"  "the flux density error"

Done

- "as -0.41"  "as -0.41σ "

Done

- I did not see a discussion of the apparent variability of the flux density of the PRS. What is the amplitude of variability (~20%?), how does that compare with FRB 121102, and is this consistent with being due to scintillation in the Milky Way ISM?

###Response

The amplitude of variability from L or S band observation is about 20% and C band is 10% among all epochs. In the FRB 121102 paper (Chatterjee et al. 2017), the authors show that the flux density of the FRB 121102 PRS varies by around ten percent on day timescales but no results for the amplitude of variability on an epoch basis. They instead gave the significance of the variability of the PRS at the C band, which is larger than 5. For our study of FRB 190520, the significance is 0.29 at L band, 2.98 at S band, 0.64 at C band. This suggests the variability on the epoch-epoch timescale is insignificant as compared to FRB 121102 studied in Chatterjee et al. 2017.

Based on the epoch-to-epoch flux measurements of the PRS, the variations are not likely to be caused by refractive scintillation. This can be accounted for if the source is extended than PRS for FRB 121102 (~ 8 pc). Since our preliminary result from VLBA is that the size of the PRS for FRB 190520 is ~ 30 pc, this supports the lack of refractive scintillation for it.

118. Figure ED4:

- Are there any constraints on the metallicity of the host galaxy?

###Response

We will discuss the metallicity of the host galaxy in the consecutive paper.

119. Figure ED5:- No comments.

Extended Data Table

120. Table ED1:

- Given how close the DMs are to each other, from day to day, it appears like the DM uncertainties are overestimated. A figure of DM and apparent scattering measure vs. time would be valuable.

###Response

Thanks for this comment. We added the scattering measure and DM vs. time in the Method. Due to the limit of long table, we put the previous table 1 to supplementary table 1.

- Include information on the reference system used for the burst times.

###Response

Thanks, The used reference system is added.

- Include note on how the DMs were determined (and why it's the same for each day).

###Response

The DMs are determined from the method described in Hessels et al. 2019. However, some bursts could not be resolved at higher time resolution and are hard to take multi-peak profile analysis. We assume the intrinsic DM value does not change obviously during the same observation. Thus we take the burst with the highest S/N for DM fit, and use its determined DM for all the bursts from that day.

- Emphasise that the “Energy” is assuming isotropic emission, which is unlikely to be true.

###Response

Done. Add to the caption: “Energy here refers to equivalent isotropic energy.”

121. Table ED2:

- Are these “on source” observation durations, i.e. not including phase calibrator (etc.) scans? Clarify in a table note.

###Response

Thank you for pointing it out. We have modified the table notes. Due to the limit of long table, we put the previous table 1 to supplementary table 1, the table sequence has been changed in the main tex.

-

122. Table ED3:

- “in ra. and dec.”  “in RA and Dec” (to match table labels)

Done

- Values in table have hyphens instead of negative signs (\$-\$).

Done

123. Table ED4:

- “The alphabet in the first”  “The label in the first”

Done

- “image signal to noise ratio”  “image S/N”

Done

- The DMs reported here are wildly different. Explain why, and what these correspond to. Were these the discovery trial DMs?

###Response

- We thank the reviewer for pointing this out. These DMs were estimated using the offline refinement analysis on the bursts. They represent S/N maximising DM value for each burst. They could be different because we are not able to resolve the temporal structure of the bursts, and hence these DMs do not represent the true intrinsic DM of the FRB, but are influenced by systematics.

124. Table ED5:

- The DMs quoted here were determined using different assumptions compared to the FAST burst analysis. Why?

###Response

Also, the DMs appear to vary, but there is not discussion of this.

- As noted in our previous response, the time resolution of VLA observations (10ms) was comparable to or more than the width of the bursts. Due to this, it is not possible to estimate structure maximising DM for VLA bursts, as any temporal structure would be resolved. We therefore report S/N maximising DMs for VLA bursts. This can also explain the apparent variation in DM values. We have added text to the paper to discuss this.

- Some of these bursts are ~10x brighter than the brightest bursts seen by FAST. Isn't that surprising?

###Response

For FAST, the fluences were determined using the burst signal present in the observing band. For VLA, by modeling the burst spectra using a Gaussian shape, we were able to determine the fluences of the bursts that were not completely within the observing band, and therefore these values can be higher than the burst fluences observed with FAST. We have also modified the values and errors in ED Table 4.

Reviewer Reports on the First Revision:

The authors have responded in detail to the points raised in the first review (by both referees) and I think the manuscript is improved as a result. In many cases, the author responses and associated changes were sufficient to close out the point raised. However, in quite a few cases, while an answer was provided in the author's response, no change was made to the manuscript to provide that information in the manuscript. Apologies if that wasn't clear in the initial report, but generally if a question is raised it implies a gap in the manuscript, and an answer to the referee alone doesn't close that gap. I have highlighted responses where I think additions to the manuscript are still warranted below. There are also still a couple of more major points that I do not think were adequately addressed in the first review: again, they are included below. Finally, there are still quite a few grammatical errors, missing words or spaces, etc. I have highlighted some but not all.

I start the second-round review with a couple of new comments on the revisions themselves, followed by a comment on one response to the other referee, followed by any unresolved points from the original review, where I keep the original numbering – resolved points from my first review I have removed, so there are gaps, and the original comments refer to the line numbers in the original submission. I've put in ***** lines as delimiters between each comment, since it is a bit hard to see in the text box. I've also attached a word doc that has colouring (black= original, blue=author response, red=my latest text) which is probably easier to read.

NEW COMMENTS

Line 158: “properties of FRB 190520B could be found in Table .” (missing table number).

Line 161: “While the observed burst repetition and morphology can be temporal...” – I think time-dependent is a more correct word to use than temporal here.

Line 184: “Active repeaters with PRS may...” -> with *a* PRS

Figure 1: The inset labels are much too small, and should be enlarged for readability.

Figure 3: It only occurred to me when re-reading the manuscript that it doesn't appear like the uncertainty region plotted with shading on the plot has been drawn from a lognormal distribution, as is described in the Methods. Also, the MW halo DM is listed in the caption as $50 \pm 25 \text{ pc cm}^{-3}$, while elsewhere in the manuscript it is listed as $25 - 80 \text{ pc cm}^{-3}$.

SELECTED RESPONSES TO THE OTHER REFEREE REPORT

Other Referee: Some of these bursts are ~10x brighter than the brightest bursts seen by FAST. Isn't that surprising?

Author Response: For FAST, the fluences were determined using the burst signal present in the observing band. For VLA, by modeling the burst spectra using a Gaussian shape, we were able to determine the fluences of the bursts that were not completely within the observing band, and therefore these values can be higher than the burst fluences observed with FAST. We have also modified the values and errors in ED Table 4.

My comment: I don't understand how the bandwidth of a burst's spectrum should make any difference to the fluence. Fluence is a measure of spectral flux density multiplied by time, so it is measured per unit bandwidth. If the spectral flux density is being measured only within the burst's spectral envelope, then it doesn't matter whether a half, a quarter, three quarters, or the entire burst spectrum is visible – the result will be the same (at least, it would be for the simplest case of the emission being a top hat function in frequency). The author's response implies to me that the approach taken for the FAST bursts was different and simply averaged the entire FAST observing band prior to estimating the fluence (which will of course reduce the average spectral flux density, if the emission is band-limited, by diluting the burst with regions of spectrum with no emission). If that is the case, why was this done? Why not just select the region of spectrum in which the burst is "on", which is what is effectively done with the VLA bursts? It means that the FAST bursts and VLA bursts are being represented in a totally different manner – one has the intrinsic burst fluence being down-weighted by some amount dependent on how much of the FAST bandpass it was present in, and the other does not, which would lead to the odd result highlighted by Jason that the VLA seemingly sees much brighter bursts. I think this should be rectified, because otherwise it gives the misleading appearance of an average spectral index since the VLA observations are at higher frequency. (And not enough information is provided in the paper about the FAST bursts – namely, their bandwidths - to enable a reader to "undo" the effect.) If the approach taken to estimating the FAST fluence is not changed, then a note should be added to say that the FAST fluences are biased low by it.

REACTIONS TO COMMENTS ON FIRST REPORT

Overall comment on conclusions, prior to the enumerated comments:

Original Comment:

In terms of conclusions, the primary conclusion of the paper, that FRB repeat activity level may be correlated with local source environment, is well supported. As I noted in the first paragraph of the review, I think that a logical extension of this connection is that outlier sources that might otherwise bias the Macquart relation between DM and redshift may be

identifiable (and potentially even calibrate-able.). It would seem logical to comment on this directly in the manuscript.

Author Response:

Thanks for pointing this out. We add the comment to the conclusion:

“Further study of such correlations may help identify outliers to the Macquart relation and potentially help calibrate biases.”

My new response:

I'm going to nitpick here and say that this wording does not necessarily convey the full depth of what is possible if there is a relation between some other FRB observable (or combination of observables) and the host DM contribution. When I said “outlier sources” in my initial report, I meant sources whose host DM contributions are outliers (relative to the typical host DM in the broader FRB population). I realise that was a bit contextual and could have been interpreted as outliers to the Macquart relation, but I would argue that the Macquart relation relates extragalactic DM (after stripping out the Milky Way and estimated host galaxy) against redshift, hence it doesn't make sense to talk about a large host galaxy contribution causing an outlier to the Macquart relation – that will only happen if the host galaxy contribution is mis-estimated. Anyway, I would suggest a phrasing such as “..identify outliers in the host galaxy DM contribution to observed FRBs, and potentially help calibrate effects that would otherwise bias the Macquart relation” is clearer. Basically, the outliers are in the host DM contribution, the removal of which will calibrate away a bias to the Macquart relation.

Overall comment on the RM, prior to the enumerated comments:

Original Comment

On the other hand, the extremely important result that the RM may be very high (like 121102) is referred to nowhere in the main text, despite this being (if confirmed) an extremely important additional link between the two sources in terms of their local environment. If the results of the analysis are not referred to in the main text, why is this in the Methods?

Author Response

The claim that the RM may be very high is only one interpretation of the FAST result. Higher frequency bands and more observation will be needed, per experience with FRB 121102 (no RM detection by FAST in L-band). We still leave it in the Methods in case readers may be interested in the RM results so far.

My new response

I guess it is up to the journal whether they consider it appropriate to have material in the methods that is not referred to in the main article at all and not necessary to support any of the conclusions. My opinion is that either it should be referred to in the main text or dropped from the Methods, but the editors can rule on that.

UNRESOLVED ORIGINAL ENUMERATED COMMENTS IN ORDER OF APPEARANCE

1. In the abstract, a number of quantities are used without context or insufficiently defined. Fast Radio Bursts themselves are introduced with no context. The term “repeater” is introduced without context, as it “persistent radio source” (noting my earlier point about defining PRS precisely). “The estimated host galaxy contribution DM_{host} ” is similarly imprecise: at this stage, DM (Dispersion Measure) has not been defined, so it would not be clear to most readers what this “contribution” is. Grammatical errors abound: Fast Radio Bursts (FRB) should have “FRBs” within the parentheses, line 44 should read “...to be associated with **a** persistent radio source”, no space before “Here” on line 45, line 51/52 should read “...with **a** confirmed association between **an** FRB and **a** compact PRS”, etc.

Author Response

Thanks!

The summary paragraph has been restructured and some contexts were added. Some definitions have to wait for the main texts due to length limitations.

“Fast radio bursts (FRBs) are the most energetic radio transients in the Universe, the central engines of which remain unknown and could be diverse. The dispersion sweeps of FRBs provide a unique probe of the ionized baryon content of the intergalactic medium as well as FRBs’ natal environments.”

My new response

Another nitpick, but FRBs are not the most energetic radio transients. Longer-lasting events such as TDEs give off much more energy in the radio, just over a longer period of time. I’m not sure how the authors want to present this (most luminous fast radio transients?), but it should be clarified.

Later in the paragraph, “The estimated host galaxy dispersion measure (DM) $DM_{\text{host}} \approx 902+88-128 \text{ pc cm}^{-3}$ is nearly an order of magnitude higher than the -50 average of FRB host galaxies, and much larger than those of the intergalactic medium” – the part of the sentence after the comma doesn’t make grammatical sense. Perhaps “and is much larger (for this source) than the contribution from the diffuse intergalactic medium.”, but I’m not exactly sure what this part of the sentence is really trying to say.

Finally, I still think the word “repeater” is not something that can be used without definition in an abstract of a non-astronomy journal. “repeating FRB” would be clearer.

2. Line 68:

using an uncertainty in arcseconds for right ascension right next to the position (in seconds) is rather confusing. I suggest that either giving the uncertainty in seconds, or quoting the full position (RA+Dec) followed by an uncertainty ellipse in arcseconds, would be less prone to misinterpretation. Moreover, it is stated that the uncertainty is dominated by systematic considerations, but this is never shown for the FRB in the Methods (it is shown for the PRS). Finally, I disagree with the uncertainty presented for the FRB – how can the systematic uncertainty on a position which is obtained from a handful of milliseconds of exposure time

have a smaller systematic uncertainty than the PRS, which has hours of exposure time? In a best case, they would be identical, but as I discuss in the methods comments below, I think the systematic contribution to the FRB position uncertainty has been given insufficient attention.

Author Response

We have now included the errors quoted separately from the uncertainty to help avoid confusion. Regarding the uncertainty, we have also changed a few things but those are noted in your later comment that discusses these issues in more detail.

My new comment

“persistent, radio continuum...”: drop the comma

5. Line 87/88 – both R and R' band are referred to. Which was used? I note that R' and R are again both used later in the Methods section. Line 89: How is that H α luminosity derived? It is not derived anywhere in the Methods.

Author Response

The phrase “extended R-band structure” includes a typo. It should be R'-band. We have corrected that typo in the text. Thanks!

The H α luminosity is derived based on the luminosity distance for $z = 0.241$, and the extinction corrected H α flux. The detailed approach of extinction correction on the H α flux will be included in a subsequent paper in preparation.

My new response

But where is the extinction-corrected H α flux presented? Extended Data Figure 3 shows the flux not corrected for slit loss or extinction correction, and then suddenly the final extinction-corrected H α flux appears on line 650 when deriving the DM contribution, but the luminosity itself is never derived in the Methods. Surely this should be developed clearly in the Methods (first the raw flux should be stated, then extinction and slit loss corrections applied, then luminosity derived from that corrected flux, and then later the DM also derived from the corrected flux)? (and the main text should refer to the methods?)

6. Line 101: There is no reference to the Methods section where the DM_IGM number is calculated. I also am unable to reproduce the reasoning that yields the estimated value and range for DM_IGM, but we shall return to that in due course.

Author Response

The main text and Methods section now elaborate on how the calculations were done, including errors in the MW estimates (NE2001 error and range of possible halo contributions to DM). The error ranges we use are generous (i.e. conservative) and indicate, also including cosmic variance in the IGM contribution, that there is no way to avoid the conclusion that the host-galaxy DM is large.

My new comment:

I agree with the conclusions, and the changes have brought some improvement, but I think there is still some room for improvement in the presentation. Specifically:

Line 110: I think that “Increasing the ionised fractions from” should be “Varying the ionised fraction between”, since the initially used value was 0.85, not 0.6. Also, in the next sentence, the value calculated for the range doesn’t look quite right – it looks like only the mean value of DM_IGM was varied with f_IGM, rather than varying sigma_DM_IGM also, which would increase the range just a bit more. Why is the DM_IGM then presented in the form X to Y, rather than mean +pos – neg as it was in the previous sentence?

Going to those calculations in the Methods: On line 639, the result for sigma_DM_IGM implies that it is linearly dependent on f_IGM, while the preceding equation implies that sigma_DM_IGM should depend on the square root of that quantity. So I think these numbers, while they won’t change materially, need to be checked and harmonised. Finally, at line 643/644, it would make more sense to swap the order of the sentences and calculate DM_IGM first and then DM_host (since DM_host is calculated by subtracting DM_IGM and DM_MW+halo from DM_observed.) And it would make sense to perform the calculations that are reported in the main text (i.e., calculating DM_IGM when varying f_IGM also, thus coming up with the final number that is reported in the main text.)

8. In this same paragraph, the predicted scattering time in the event of a chance background source is only two orders of magnitude greater than observed. In the Milky Way, scattering times show a very large degree of variability. I would suggest that the (lack of) scattering on its own is not conclusive evidence to rule out a background source, but combined with the low Pchance of the host galaxy association, it further strengthens the already very strong case.

Author Response

The case is stronger than stated in this comment. Yes, the Milky Way produces a wide range of scattering times, but that is over a wide range of dispersion measures. For a specific value of DM, e.g. 500 pc/cc, we see a total range that might be as large as two orders of magnitude if outliers are included but the probable range is only about 1.5 orders of magnitude.

On geometrical grounds, the scattering from an intervening galaxy would be more than 10^4 times that of a host galaxy if the scattering medium is the same. Since the host galaxy DM appears to be > 500 pc/cc (in its frame, not the observer’s), we would expect for a Galactic pulsar a mean scattering time of 9.6 ms at 1 GHz and a probable range from 1.6 to 54 ms. These numbers are from a fit to $\tau(\text{DM})$ for Galactic pulsars that appears in several places in the literature (e.g. <https://arxiv.org/abs/2108.01172> equation 8 and references therein, e.g. Krishnakumar et al. 2015, Bhat et al. 2004, NE2001 2003). For an intervening galaxy that produces the same DM of 500 pc/cc, we include the geometric factor of 10^4 and another factor of 3 that corrects the Galactic prediction for spherical waves from

a pulsar to plane waves for an extragalactic source, and we divide by a $(1+z)^3$ factor to get the scattering time in the observer's frame.

This gives:

$\tau(\text{intervening}) > 10^4 * 9.6 \text{ ms} * (\text{geometric factor} = 3) / (1+0.241)^3 = 150 \text{ sec}$ at 1 GHz compared to 40 ms for the measured scattering time (scaled to 1 GHz). This is more than three orders of magnitude larger than observed. Galactic variance can reduce this by a factor of $(1.6 / 9.6) = 0.2$ or increase it by a factor of five.

An alternative might be to say that an intervening halo produces low-level scattering that, with the 10^4 geometric factor, multiples up to the observed scattering. That would mean that most of the DM not from the IGM and MW would be from the host galaxy. That would produce some scattering but perhaps not as much as observed due to the redshift factor and the greater IGM contribution if the host is at $z > 1$, say. The problem with this configuration is that there is no evidence for *any* scattering from galaxy halos, certainly not from the MW nor from M81's halo.

All told, the scattering constraint strongly favors scattering from a host galaxy at $z=0.241$ with a large DM contribution.

For completeness, one might argue that the association of the $z=0.241$ galaxy is completely unaffiliated with the FRB source and has no influence on the line of sight. Then all bets are off, but it would be a (unlikely?) coincidence that the H-alpha measurements yield a DM estimate that is compatible with the inferred DM host if the host is in fact at $z=0.241$. A spurious association seems to be a more convoluted interpretation than the one we put forward in the paper and where the scattering is in fact constraining on the presence or not of an intervening galaxy.

My new comment:

I think the changed text dealing with scattering is an improvement. I note however that in the methods, I think there is an error of three orders of magnitude in equation 9. It seems to mix and match equations 1 and 2 from the Koch Ocker paper – the constant should be in nanoseconds if all the lengths are in the same units (their equation 1). The constant only goes to microsec if the distances to the galaxies are in Gpc and the scale length is in Mpc. If using the formalism from the second equation in the Koch Ocker paper, then the units of the various distances and lengths should be clearly stated in the Methods text here (lines 724 – 726), which is not currently the case. Also, the value assumed for \tilde{F} (along with where that is obtained, and what its uncertainty might be) should be stated, to enable the reader to reproduce the numbers that are obtained (it looks like that was removed in the text change). The uncertainty in \tilde{F} is nearly as important as the unknown length scale L ; driving \tilde{F} down and L up to their minimum and maximum reasonable values respectively can actually get to quite a low value of scattering time. I agree that it isn't very likely, but the reader should be presented with all the facts to judge for themselves.

10. Line 139/140: DM_host can also be observationally biased, as it contributes to an observational quantity (the total DM) to which different FRB searches have different sensitivities.

Author Response

We agree that large values of DM_host are less likely to be detected, since FRB searches are usually sensitive to a limited range of (total) DM. We have modified this sentence to reference this possibility.

My new response:

It's even worse than that, though, because it is systems where the total DM is largest (so – distant FRBs with a large host DM) that are most affected – not all FRBs with large host DMs will be affected equally. Nearby FRBs with large host DMs might be mostly detectable. This is just to say that the situation is complicated, and hence the sentence that immediately follows this revised one might be a little glib – a large sample on its own won't be enough, the selection effects need to either be negligible (systems with good sensitivity out to high DMs) or clearly tracked.

12. Figure 2: I think that either a zoom in (for one of the bands) and/or some annotations of other nearby sources would be valuable. This goes to point 4 above – are any of the other relatively nearby sources larger galaxies that may have a reasonable likelihood of harboring an FRB in their outskirts?

Author Response

We thank the referee for the suggestion. Here we provide the zoom-in version below in response. The members in our collaboration have considered the zoom-in version of Figure 2, but we still prefer the current plot configuration to display the relations between radio and optical emissions in the field. The zoom-in plot will be discussed in more detail in the follow-up paper soon.

The persistent radio emission and repeating FRB bursts are all coincided with the optical emission shown in the Figure 2 within the position uncertainties and the resolutions of the optical/NIR images, the probability of false association is only $< 1\%$. The next nearby object in J-band is 6.5" to the north of the FRB location. The same method put the false association to be $> 20\%$, which translates to the object being likely just in the neighborhood coincidentally.

Fig. Zoom in optical plots.

My new response:

That is useful information – why not put that (the chance coincidence of the next nearest object) in the Methods too? A chance coincidence probability on its own is not always sufficiently informative. If the Pchance of the next nearest galaxy was 1.5% rather than >20%, would you still feel confident claiming an association with the preferred galaxy just because it had a Pchance of 0.8%? I wouldn't. While I still think that a Bayesian approach that incorporates all the galaxies nearby is the best one, in clear-cut cases like this mentioning the Pchance for the next nearest galaxy is sufficient to reassure the reader. So I would strongly suggest including that information in the Methods.

14. Line 283: Why would some bursts show scattering and others not? Surely the simplest assumption is that the scattering is time independent, which would enable this parameter to be fixed across all bursts and constrained with good precision, which would then lead to a better precision on the (deconvolved) gaussian burst widths for all bursts? Non-gaussian intrinsic structure will surely complicate the analysis, but it will also be complicating the present analysis (where the choice of scattering vs no scattering is apparently made by eye) already, and the assumption of constant scattering is at least physically justifiable.

Author Response

The band-limited nature of the bursts means that some will be more scattered than others, depending on the center frequency and frequency extent of their measured flux density. This could have been dealt with by adopting a fitting function for all bursts with a single value of the scattering time and then taking into account the frequency range for each burst. We have reason to investigate, however, that the scattering time actually is different for different bursts. It is entirely possible that FRB environments can produce variable scattering, similar to what is seen for the Crab pulsar giant pulses, and a patchy medium can even produce different amounts of scattering for bursts that are closely spaced in time. The measured scattering time can also in fact be influenced by refraction or lensing that works in tandem with diffractive scattering, and this refraction/lensing can also be time-

variable. Sorting out these possibilities is beyond the scope of the present paper but we are actively investigating it.

My new response:

I will concede that time-variable scattering is certainly possible due to the changing line of sight through a turbulent medium. I stand by my assertion that time independence is a better default assumption unless shown otherwise, though.

Now that the “fold the frequency” confusion is cleared up, I understand that no frequency dependence was assumed for the scattering when fitting individual bursts (the new text on line 704 makes this clear.). However, I can’t see how that approach can be justified, since we know that scattering will be very strongly frequency dependent (ν^{-4}). The only defense I can imagine is that most of the bursts have a reasonably narrow fractional bandwidth, but even for a burst from say 1000-1200 MHz, the scattering is going to be 2x higher at the bottom end of the burst than at the top.

Table 1 and ED Figure 11 both refer to scattering times at 1.25 GHz, so I assume (although it is not stated, and should be) that the scattering times measured for individual bursts have scaled to this reference frequency based on the central burst frequency in each case and an assumed power law. But maybe they haven’t been scaled at all? In which case I can’t see how or why the times for different bursts with different central frequencies can be compared to each other in any useful way. I would have expected ν^{-4} to be assumed, but this should be stated clearly, both in the Methods text, and in the extended data figure 11 caption.

Anyway, looking at extended data Figures 10 and 11, I am not convinced by the claim that selecting sources by eye in order to choose whether to fit an unscattered gaussian or a scattered gaussian template is more robust than assuming a constant scattering time (appropriately scaled with frequency, of course). In Figure 11, all the measured scattering times are remarkably consistent with the mean value, given the (necessary) gross oversimplification of the intrinsic pulse shape to be a gaussian in every case. If there was a counter example of a narrow burst whose measured width is too low, clearly inconsistent with this scattering, that would be a good counter argument supporting the variable scattering, but nothing in Figure 11 supports that presently. P31 has clear intrinsic structure (so I don’t think there is evidence that that burst is over-scattered, relative to the mean), P47 might be on the narrow side but it is at relatively high frequency – what is the goodness of fit of an unscattered component compared to a component scattered with the mean scattering time? – and P50 is intrinsically wide enough that I wouldn’t expect to see the effects of the mean scattering time, but again the goodness of fit of an unscattered vs scattered single component could be compared.

To summarise: Each burst could be fit one of three ways: 1) with a gaussian only, 2) with a gaussian convolved with a (freely fitted, frequency dependent) exponential, 3) with a gaussian convolved with a frequency dependent exponential whose scale was fixed by

fitting the ensemble of all bursts. Option 2) has one more free parameter than options 1) and 3). At the moment, either option 1) or option 2) is chosen in an ad-hoc manner, burst by burst. I can't see how that can be justified compared to choosing a physically motivated default model (option 3) and evaluating all three models and only choosing option 1 or 2 if they return a goodness of fit that is significantly better than the default (in which case there is now clear evidence for time variability of the scattering – at least if the better fit is definitely not just resulting from intrinsic burst structure.)

Ultimately, what is this used for? For this paper, only to estimate the mean scattering time. That would be better done with a single fixed scattering kernel for all bursts – choosing only the bursts with apparently visible scattering would, if anything, bias that mean result high. The time dependence of scattering, if present, is not actually investigated in the main paper. Given that, the probably slightly biased view of the scattering that would result from the ad hoc modelling choices has a pretty negligible impact on the conclusions, but others may take the data in this paper and use it for other purposes. So while my suggestion is to do the modelling consistently, if it is not changed it at least should be clearly explained (the frequency dependence vs not of the scattering kernel) and the potential impacts of the ad hoc choices listed. And it is absolutely imperative to clearly describe how the results were scaled to a reference frequency of 1.25 GHz. If, as seems possible, no scaling was done, then this really must be justified (although I don't really see how it could be.)

15. Line 284: what is “the sub-pulse”? presumably “a sub pulse”, but this needs to be much more carefully defined. What level of significance is used to define a sub-burst that should be fitted (currently “the noise baseline” is stated)? At what width?

Author Response

Thanks! We change the term to “a sub pulse”. Since there is no standard pulse profile (nor average) for repeaters yet, the use of pulsar terminology is empirical. Currently, when the 'bridge' between the two closely-spaced-in-time peaks drops more than 5 sigma below the higher peak, we consider them to be two bursts. Otherwise, they are considered structures within one event (sub pulses).

My new response:

Okay, good to know – but please put that information in the Methods text, so all the readers know that, not just me! Also, my original point was meaning that grammatically, the sentence should begin “A sub pulse”, not “The sub pulse”.

20. Line 351: How is the time selection of the de-dispersed visibilities achieved? Is any weighting applied based on the fitted width of the burst, or is a top-hat selection function applied?

Author Response

- We used a top-hat selection function.

My new response:
Please add that information to the text!

21. Line 364-372: it is naive to take the statistical image-plane fit uncertainty as the sole source of uncertainty for an individual burst, as there will surely be calibration errors leading to systematic position uncertainties. This likelihood should be acknowledged and estimated (the recent papers on FRB20201124A mostly include reasonable discussions of systematic uncertainties on position estimates). It is then even more naïve to take a simple weighted mean of all of the burst positions (with statistical-only uncertainties) estimated to arrive at a final FRB position and uncertainty. An estimate of systematic uncertainty in the VLA position must be presented at the same time as that final statistical uncertainty.

Author Response

Thank you for pointing out this omission (and the implicit disagreement in our numerical reporting). We have now added in words here to describe how the systematic uncertainty for the FRB bursts is the same as the systematic uncertainty for the PRS (because, indeed, they use the same data and are subject to the same systematics). To summarize briefly also here so you don't have to find it in the draft, essentially we do frequent phase-referencing with a small enough interval that any short-timescale variations will be fitted out by our phase calibration. Thus, the short-timescale systematics should reflect the same systematics as the deep image. Additional note: while we didn't write this extra analysis into the draft, we also confirmed that the phase wander was consistent on short timescales by performing short-timescale imaging (imaging segments of 5-30 seconds, such that at least 1-3 sources could be detected at each frequency) and inspecting the position variations of those sources over time (positions were stable and consistent with statistical uncertainty; offsets of the sources were within the range of PRS-quoted systematics). While this could not be as rigorous as the deep image because each field only had 2-3 detectable sources on those timescales, it at least reassured us of consistency for the FRBs.

Your comment also brings up another point you also mentioned elsewhere, which is that we didn't take the full (statistical+systematic) uncertainty into account in the weighted mean. We agree with this assessment and have changed the position calculation and reported error:

- a weighted mean is now calculated for each frequency separately, with the weights scaling with the inverse statistical error of each measurement. The error on each of those frequencies is then the propagated statistical error, added in quadrature with the systematic error at that frequency (as determined by the PRS analysis).
- The three frequency measurements are then averaged, with the appropriate error propagation (for the mean of three values with error) quoted.

We have accordingly changed the text to reflect this change. Because our FRB position was performed as a three-frequency mean, accordingly it is appropriate (particularly in a

comparison between the PRS and the FRB emissions) to perform the same operation---a weighted mean of the positions at the three observing frequencies---for the PRS. The errors are now consistent and the PRS position-fitting description has been changed accordingly to reflect this change. Text throughout, have been updated to reflect these changes.

My new response

I think these changes are an improvement, however, there are a few things that remain to be addressed. First, using a cycle time of 10-15 minutes at B array does not guarantee that short timescale phase errors will be fitted out. The timescale of the phase noise depends on atmospheric conditions, etc, and in any case there is also still the spatial extrapolation from the calibrator to the target direction. The recommended cycle times for the VLA ensure that the data should be phase-connectable in normal observing conditions, but not that the residual astrometric errors will be negligible. If the authors can show (with a couple of brighter sources) that there are no astrometric residuals inconsistent with thermal noise on those 5-30s timescales, then that allays the concerns – but that information (the consistency of the short timescale imaging) should be included in the Methods text.

Second, the FRB positions themselves offer a natural way to check this. Are all the individual positions consistent with the weighted mean (what is the reduced chi-squared of the individual positions compared to the weighted mean?). By eye from ED Figure 1, it looks like the reduced chi squared would be >1 , but not by very much. (The L band positions look internally consistent, and consistent with the single C band position, but marginally inconsistent with the mean of the S band positions). Including that information (the reduced chi squared of the final weighted mean) would similarly improve the confidence of the final result; there is no good reason not to include it.

22. Line 391-393: Why was the PRS fit only with a point source model? If the source size is consistent with being unresolved then this is reasonable, but surely a gaussian fit should also be attempted to place an upper limit on size (at each frequency)?

Author Response

We actually took gaussian fits using the CASA imfit task for the PRS at multiple bands (L, S and C bands). The returned results showing that the source is a point source at L and C bands, with the size may as large as (1.4", 0.89") and (0.36", 0.1"), and a component with size of (0.51", 0.14") at S band. We've revised the initial improper expressions.

My new response:

Ok, this is good – but these upper limits to the PRS size should also be included in the Methods text somewhere (and, ideally, in the main text too! The main text says “unresolved” at some point, but doesn't give a maximum size). At the moment, the text merely says that imfit was used, and while the results show the positions, they don't show the size limits.

23. Line 410: Which direction are the systematic offsets? VLA-PanSTARRS or vice versa? Has the correction been applied to the final PRS position that was reported, or not?

Author Response

The offsets are obtained by subtraction of PanSTARRS coordinates from VLA coordinates. We have applied systematic corrections to the PRS positions, thus the position errors are statistical errors added in quadrature with systematic errors. In the Extended Data Figure 1 we applied the systematic corrections of all bands to the PRS positions to give an idea of the consistency of the positions of the bursts and the PRS.

My new response:

OK – so this information (the direction of the systematic offsets) should be included in the Methods section. Also, and apologies for not noticing this the first time, but why was a cross-match of $0.5''$ used? That basically imposes a prior that the true offset is (quite a bit) less than $0.5''$. I don't have any reason to believe that the astrometry would be worse than the measured value, but it would be good to confirm that the results don't change substantially if a looser cross-match (e.g., $1''$) was used.

33. Line 509: The text and references do not actually justify the range quoted for the IGM DM. The quoted reference considers f_{IGM} in the range 0.6 ± 0.1 , not 0.8 , and in any case provides no hints as to how cosmic variance was translated into this DM range. This justification should be strengthened. Also, throughout this section (and indeed, throughout the entire paper) confidence intervals should be specified (or if are typically 68%, then this should be stated somewhere).

Author Response

The range quoted for the IGM DM is now justified and confidence intervals are stated to be 68% intervals.

My new response:

This is definitely improved, although a primary reference for the f_{IGM} should be given rather than a secondary reference (Zhang et al 2018 simply cite Fukugita et al. 1998 to support their value of 0.83).

34. Equation 3: Half of the symbols used in this equation are not defined.

Author Response

Thanks! The symbols are now defined.

My new response:

I presume Ω_{Λ} in the text is meant to be Ω_{λ} in equations 4 and 5 (or vice versa).

35. Line 514: the uncertainties presented for DM_host are clearly inaccurate, having presumably neglected the uncertainties on the MW halo (and made no attempt to estimate an uncertainty on the MW contribution). The number of significant digits is also incorrect. At no point in this section is the derivation of the final value of $912 + 69 - 108$ that is quoted in the main text shown. Whatever the number is in the main text should be supported by the analysis here, and it is not. How were the multiple sources of uncertainty combined?

Author Response

No, the uncertainties were not neglected. The uncertainties are now itemized to clarify.

My new response:

The uncertainties are now different to the original value though? In the initial submission DM_host was $912+69-108$, now it is $902+88-128$, so something was updated?

As I mentioned in a previous response, it seems like the expression for σ_{DM_IGM} under what is now equation 5 is incorrect, although I suspect that it is only the expression that is written incorrectly and the value was used was correct. But the more salient point is that this expression gives a single value (the standard deviation) for the DM_IGM, while the final uncertainty is DM_IGM is asymmetric (as it should be) – it would aid the understanding of this passage of text to reorder the text such that the calculation of DM_IGM and its (asymmetric) uncertainties follows immediately after the description of the standard deviation of the lognormal distribution, and only then go on to calculating the host DM. So basically bringing the sentence on line 633/644 up to line 639.

36. Line 517: Where does the extinction-corrected H α flux come from? Which spectrum? What was the extinction correction? The caption of extended data figure 4 says that the Palomar spectrum that is plotted was not corrected for slit loss, if it comes from this spectrum, was that correction applied, if so what was the result?

Author Response

The extinction-corrected H-alpha flux is from the Keck spectrum. We were using the Balmer decrement line ratio $H\text{-alpha}/H\text{-beta} = 2.8$ in case to estimate the extinction against H-alpha emission. The detailed analysis is included in the associated host galaxy analysis paper that will be submitted soon. The Palomar spectrum is not corrected for the slit loss since we mainly used it to determine the redshift but the H-alpha line is consistent with the Keck/LRIS result. The Keck LRIS spectrum was not corrected for the slit loss either because the compact host galaxy with a seeing condition of $1.1''$ is considerably smaller than the slit width of $1.5''$.

My new response:

Ok. Please clarify what was done in Section 4! Section 4 should say which spectrum is being used to for the final H-alpha estimates used subsequently, what corrections were applied, etc. I can appreciate that more details may be forthcoming in a subsequent paper, but these are pretty fundamental.

43. Line 542-550: As already noted in point 14, surely the simplest assumption possible would be for no time dependence to scattering, and it would make more sense to fit a single scattering time to the entire ensemble of bursts? That would probably give a better estimate of the scattering time, and then also a better estimate of the distribution of widths. Otherwise, the widths of the bursts for which no scattering time was estimated (and which were then fitted by an unscattered gaussian) are very likely to be biased high. Since the main purpose of this analysis is to estimate a scattering time in order to rule out the observed galaxy being a foreground source, obtaining a mean value this way, by fixing it across all bursts, must surely be preferred.

Author Response

Please refer to the response to point 14. We do not fix the scattering time across all bursts because we see clear evidence of scattering in some bursts and not in others. Time-variable scattering can be explained by a patchy medium near the source.

My new response:

See the response to point 14.

49. Extended Data Table 5: I'm curious how an apparently 20 sigma burst (burst C5) can be identified to have a width of ~0.1 ms with an uncertainty of 70 microsecond, given the 10ms sampling of realfast? All of the other bursts have a width uncertainty of 1-2ms, as I would expect. Perhaps this is a typo? Other numbers in these tables are curious: e.g. burst L1 has fluence 4(6) Jy-ms. Why is the fractional uncertainty >100%? Why is the localization of burst S1 more than twice as precise than any of the other S band bursts, despite burst S2 and S3 having comparable S/N? I see that it is much wider-band than the other two, but that doesn't really matter given that the image S/N is the final result of the width+fluence+bandwidth of the burst combined. Looking at the beam sizes and the S/N values quoted, it seems more like the statistical uncertainty is too large for the other bursts rather than too small for burst S1. This might indicate that the fitted source size was >> the synthesized beam size in those cases, which would be indicative of a concerning systematic error (as the FRB itself must be a point source, and hence should have a size approximately equal to the synthesized beam size).

Author Response

We thank the referee for pointing this out. We have edited the values in the table and added more description to this section to explain the burst fitting process and the caveats to this analysis.

For CASA imaging, we only used the frequencies that contain burst signals to form the image. We then use that image to determine the burst position. Therefore, the beam size for each image will change based on the signal peak frequency of the respective FRB. This can lead to different size of fitting errors even for similar S/N bursts.

My new response:

There are still a number of inconsistencies in these two tables. The most glaring is the case of burst C1, which remains unaddressed. I absolutely do not believe that it is possible to measure the width of this burst, observed with 20 sigma significance using 10 ms sampling, to a value of 0.1ms with an uncertainty of 70 microsec. Consider this: the DM uncertainty for that burst is listed as $1300 \pm 200 - 200$ pc/cm³: over the reported bandwidth of 230 MHz centred on 5.27 GHz, DM 1500 would lead to 17ms of DM sweep vs 12 ms for DM 1100 – if that uncertainty in DM is right, there is ****no way**** to get a sub-ms width precision. Not that I think that it is possible with a 20 sigma burst and 10ms sampling in any case. There must be either an error in the fitting or an error in the transcription. I'm doubtful on the widths of the other bursts too: applying the same approach to S4 yields an uncertainty in DM sweep time of 7-8ms, making the inferred width of $0.7 \pm 0.6 - 0.5$ look infeasible, although it is less glaring than C1.

Apart from that, there is still no explanation for why S1 has an astrometric precision >2x higher than S2. S2 has a higher central frequency and a comparable S/N. So there is no way that S1 should have a position which is 2-3x more precise than S2 – their uncertainties should be comparable. I'll ask again explicitly: why is the statistical precision of S2 so low? It is only about 1/10th of the synthesised beam, when it should be 3x better.

50. Reference 41: why is this the arxiv reference rather than the published ApJ article?

Author Response

Because the article hasn't been published yet and is still under review.

My new response:

Reference 41: "Law, C. J. et al. A Multi-telescope Campaign on FRB 121102: Implications for the FRB Population. ArXiv e-prints (2017)" was published in ApJ over 4 years ago.

Referee #2 (Remarks to the Author):

General comments:

The revised manuscript is definitely improved, and the authors have implemented or addressed most of my comments on the original submission. I read the revised version and provided detailed comments (see below) *before* reading the author's responses to my original report. I see that some of my comments are still similar to issues that I flagged originally. In these cases, I think that the paper is still unclear and requires further editing to address these points fully (in some cases, that may mean including information that was only provided in the referee response). This includes, for instance, being more clear on whether the *apparently large* DM variations are genuine or reflect an underestimation of the uncertainties (I suspect that it's the latter).

My main substantive criticisms of the paper's claims remain:

1) Since repeaters appear to be a small fraction (~5%) of observed FRBs, the following claim is too general in my opinion: "suggesting caution in inferring redshifts for FRBs without accurate host galaxy identifications". I think this statement should be clearly qualified with "at least for repeating sources".

2) Since there are also "active repeating FRBs" with low local DM and *no* persistent radio counterpart, the following claim "The dense, complex host galaxy environment and the associated persistent radio source may point to a distinctive origin or an earlier evolutionary stage for active repeating FRBs." is more accurately stated as:
"Repeating FRBs have been found in a range of environments, some much less extreme compared to FRB 121102 and FRB 190520. Whether this diversity reflects different evolutionary stages of a single source type, or multiple source types, requires further investigation."

I appreciate that the authors have softened their claims compared to the original version, but it is important that the paper is clear on what is well-supported by their new results - as opposed to making claims that "may" be true, but are actually quite debatable, and in some cases contradicted by other observational results in the literature.

I think that with this sharpening of the main claims, and by addressing the detailed comments below, that the paper contains results whose novelty and significance will make the paper appropriate for publication in Nature.

I look forward to seeing a revised version.

Sincerely,

Jason Hessels
University of Amsterdam & ASTRON

Title:

Line 1:

- Suggest not using an acronym in the title.
- From the title, it's now not clear what is novel compared to FRB 121102.

Summary:

Line 42: "the most energetic radio transients in the Universe,": can you really say that they're more energetic than gamma-ray burst afterglows? Maybe higher peak luminosity?

Line 45: "as FRBs' natal environments"  "as the natal environments of FRBs"

Line 48: "and identified with"  "and associated with"

Line 49: "a redshift $z = 0.241$ ": quote uncertainties.

Line 53: "source after FRB 121102 with"  "source, after FRB 121102, with"

Line 54: "a compact PRS.": I don't think it's worth defining this acronym in the summary. In fact, you don't even use it in the following sentence.

Line 54: "The dense, complex host galaxy environment": is it really the "host galaxy" environment that is dense? It must surely be local to the FRB source. Also, what does "complex" mean in this context?

Line 54: "The dense, complex host galaxy environment and the associated persistent radio source may point to a distinctive origin or an earlier evolutionary stage for active repeating FRBs.": why? FRB 20201124A is also "active", and closer, but doesn't have a persistent radio source counterpart or a very high local DM contribution.

Main:

Line 57: "FRB 190520B (corresponding TNS name FRB 20190520B, same hereafter)": why not just use a single name for the source from the start? Note that, even after saying you'll use "FRB 20190520B", you then use "FRB 190520B" again. Has the name "FRB 190520B" ever appeared before? Seems like you should either use "FRB 190520" or "FRB 20190520B".

Line 60: "Four bursts were detected during the initial scan, suggesting a repeater.": I would say that multiple bursts "demonstrates", rather than "suggests", a repeating source.

Line 63: "hr": unit shouldn't be italicised.

Line 64: "Similar to other repeaters, this FRB shows...": reference needed. Hessels et al. 2019 would be a good one for this, especially since it's already in your list (Ref. 25), but I'm of course biased.

Line 70: "We measured a burst source position of": is the quote position in the ICRF? Be explicit about this.

Line 84: "shows the most likely stellar component": I don't understand what "most likely stellar component" means.

Line 89: " $z = 0.241$ based": quote uncertainties.

Line 112: "the extremely low chance coincidence probability": I would drop the "extremely". I don't think that $\sim 1\%$ is "extremely" low.

Line 132: "consistent with the low chance coincidence of FRB 190520B with J160204.31-11718.5": this is redundant with the beginning of the paragraph, so could be dropped.

Line 139: "may add considerable variance to estimates for the IGM contribution": if a significant fraction of FRB sources are like FRB 190520, so it should be noted here that repeaters are a small fraction of the known FRBs ($\sim 5\%$, cite CHIME/FRB first catalogue).

Line 140: "It is also more likely that FRBs with large DMhost will not be detected by search systems sensitive to a limited DM range.": true, but the DM of FRB 190520 is well within the range being searched by most experiments, and it is far from being the highest FRB DM to be observed to date.

Line 141: "A large sample of FRBs"  "A large sample of precisely localised FRBs with measured host galaxy redshifts" (would also be better if you quantified "large"; I'd say that hundreds should be enough to get a robust idea of what is "typical")

Line 142: "allow the host galaxies, their circumgalactic media, and near-source environments to be probed statistically along with the IGM.": I would cite the recent Mannings et al. paper here.

Line 145: "the first repeating"  "the first known repeating"

Line 146: "and the first to be identified with a compact, luminous PRS": it would be useful to have a table that compares the properties of both sources and their hosts, including: i. burst rates, ii. burst energies, iii. local DM, iv. local RM, v. host galaxy stellar mass, vi. local SFR, vii. persistent source luminosity and spectrum, etc.

Line 153: "larger than highest"  "larger than the highest"

Line 155: "unresolved structure in VLA observations": it's important to quote the maximum possible size of this source (and compare with the case of FRB 121102).

Line 157: "a true PRS": the switching between "radio continuum counterpart" and "PRS" / "persistent radio source" terminology is confusing. "true" is even more confusing: the source is a persistent radio source regardless of whether it's associated with FRB 190520.

The discussion and terminology should just use a single term and focus on whether the emission is

- i. spatially co-located with FRB 190520 and
- ii. sufficiently compact that it is unlikely to be due to star formation.

Line 157: "The whole properties"  "The properties"

Line 158: "could be found"  "can be found"

Line 158: "in Table .": missing table number.

Line 161: "While the observed burst repetition and morphology can be temporal due to various mechanisms": this sentence is confusing. I think you mean that the burst and source properties likely vary with time.

Line 162: "PRS emission and DMhost reflect more persistent aspects of the FRB environment and thus may be more reliable tracers of any putative subclasses.": this seems to contradict your earlier argument that active repeaters might be found in such environments.

Line 167: "implying that such features could require dense, magnetized plasma within parsecs of the source of the bursts": importantly, FRB 121102 is *known* to be in dense, highly magnetised local environment, as shown by its large and highly variable RM (Michilli et al. 2018). That this is necessary for the activity to be high is not well supported, however, by the contradicting case of FRB 20201124A (and FRB 20180916B).

Line 173: "Some active repeaters do have comparably strict limits": comparable to what? Also, the limits are *much* deeper (in the case of FRB 20180916B and FRB 20200120E) than the observed luminosities of the radio sources coincident with FRB 121102 and FRB 190520).

Line 174: "which suggests complexities in the connection between burst activity": I would say that it argues that the current study doesn't convincingly argue for *any* connection between burst activity and the presence of a persistent radio counterpart.

Line 180: "and other properties.": what other properties? This is vague.

Line 183: "Active repeaters with PRS may either be a distinct population or FRB sources at earlier evolutionary stages.": these are not the only options. They may simply be similar sources, of comparable age, that happen to be in a denser local environment.

Figure 1:

- The bursts would be easier to visualise, in some cases, if the time-frequency integration factors were larger.

- "These six bursts are chosen from each observation epoch by its characteristic dynamic spectra.": it's unclear what "by its characteristic dynamic spectra" means.

- "The 'P0' means the first burst of all the 79 bursts."  "The burst labels 'PXX' are in order of arrival time." (then also point to the table of burst properties)

- "The color are linear scaling"  "The color map is linearly scaled"

- "The bad frequency channels are set to zero and labeled in red patches on the left.": would be better to plot no data here at all, since the values aren't 0, there is no data.

- y-axis label on the bottom panel: "Detection"  "Number of bursts"

Figure 2:

- "the best FRB position at": indicate that these correspond to RA and Dec, and at epoch J2000.

- "The infrared Jband image by Subaru/MOIRCS shows emission only at the location of the peak of the optical light profile of the host galaxy.": that's not so easy to see, given how zoomed-out these figures are. A zoom-in on the host galaxy would be useful.

- What is the optical/IR source that is right next to the host galaxy?
Is that known to be a foreground star?

- Indicate whether these images are oriented with East to the left, or not. It would be better to include coordinate axes and labels (RA and Dec).

- Indicate what the artefacts in the optical/IR images come from.

Figure 3:

- "FRB 190520" is used on the plot, whereas "FRB 190502B" is used in the caption.

Table 1:

- "Measured Parameters"  "Measured Parameters of Bursts"

- A table comparing properties with FRB 121102 and FRB 20201124A would also be useful.

- Provide the uncertainties for RA and Dec.

- Number of detections = 81. But the first two paragraphs of the paper quote 75 bursts seen by FAST and 9 with VLA (= 84).

- "Measured width (ms)": is this an average measured width for all bursts (at all observing frequencies)?

- "Scattering timescale (ms) at 1.25GHz": also an average (or median)?
- "Scintillation bandwidth (MHz) at 1.4GHz": also an average (or median)?
- Also for the persistent radio source position, the uncertainties should be quoted.
- I would suggest also quoting an upper limit on the size of the persistent radio source.
- "redshift"  "Redshift" (since all the other rows are capitalised)
- MSun: indicate that this is the *stellar* mass of the galaxy.
- Footnote b: "electron density mode"  "electron density model"

Author contributions:

Line 295: "the associated PRS": I'd avoid using this acronym in the acknowledgements.

Line 296: "Optical/NIR"  "optical/NIR"

Line 298: "contributed to measured the burst scattering, modelling combined with analysis of propagation": this isn't a grammatically correct sentence. Needs rewording.

Line 301: "RFI": I'd avoid using this acronym in the acknowledgements.

Methods:

1 Observations:

Line 325: "structure(see"  "structure (see" (missing space)

Line 325: "see result plot from ref.34": which plot? I'm confused.

Line 326: "in that date"  "on that date"

Line 328: "and only kept Stokes I"  "and only Stokes I was recorded"

Line 329: "the trial DM range is from": start new sentence.

Line 329: "we matched the pulse width by a boxcar search.": what range of widths was searched?

Line 337: "The sub pulse is recognized if the profile peak does not fall behind the noise baseline.": this needs to be rewritten. It's awkwardly worded and unclear.

Line 339: "The bandwidth of each burst is roughly estimated by its spectrum": it's obvious that the

bandwidth of the burst is estimated from the spectrum, but *how*?

Line 345: "DDT project: 20A-557"  "DDT project 20A-557"

Line 373: "formed by realtime system"  "formed by the realtime system"

Line 377: "when the data is de-dispersed at a DM closer to the DM of the candidate": maybe better to say "closer to the true DM"

2 Localization of bursts:

Line 392: "data with Common"  "data using the Common"

Line 394: "each observation with the burst"  "each observation with a detected burst"

Line 395: "data using CASA task"  "data using the CASA task"

Line 395: "Observation of 3C 286 (before the FRB observation) was used"  "Observations of 3C 286 (before the FRB observations) were used"

Line 400: "Therefore, the burst positions on short timescales will have systematic errors of the same magnitude as the deep imaging.": I think that this is probably true, but I would point out that the uv-coverage is much worse for individual bursts than it is for the integrated image (see e.g. Nimmo et al. FRB 20201124A paper, where the situation is even worse).

Line 404: "For each burst, we search different spectral window ranges to generate the image with highest S/N": quantify how many trials are being done here.

Line 425: "the best estimate for burst"  "the best estimate for the burst"

Line 437: "Due to this any temporal structure in the bursts (multiple components, scattering, sub-burst drift) would be resolved out and we would not be able to model it.": don't you mean the opposite? The burst structure is *unresolved*, as opposed to being "resolved out".

Line 445: "Burst times are defined in barycentric MJD time": do you mean TCB or TDB? Be clear.

3 Persistent Radio Source:

Line 496: "This can be accounted for if the source is more extended than the PRS for FRB 121102.": would be useful to quantify what lower-limit on physical scale this implies (>~ 1pc, right?).

Line 505: "of a few tens mJy"  "of a few tens of mJy"

4 Galaxy Photometry and Redshift Determination:

Line 530: "was obtained with Double Spectrograph (DBSP)"  "was obtained with the Double Spectrograph (DBSP)"

Line 535: "The PRS location later is found to"  "The PRS location was later found to"

Line 538: "as reported in PanSTARRS DR1 catalog"  "as reported in the PanSTARRS DR1 catalog"

Line 540: "allowing both the coordinate of VLA persistent radio source and the M-star to fall into the slit"  "ensuring that both the VLA persistent radio source and the M-star fell within the slit"

Line 541: "under photometric sky condition"  "under photometric sky conditions"

Line 553: "indicating the extended"  "indicating that the extended"

Line 556: "coincidence probability of the galaxy (J160204.31-111718.5) to that of the burst"  "coincidence probability of the galaxy (J160204.31-111718.5) and burst source"

Line 559: "the half light radius of galaxy"  "the half light radius of the galaxy"

Line 560: "the localization error region": you mean the *burst* localisation region? Be clear.

Line 561: "the localization error on the FRB location"  "the localization error on the FRB position"

Line 562: "the size of the host"  "and the size of the host"

5 FAST Burst Sample Analysis:

Line 575: "We find the burst rate of FRB 190520B is": it's important to associate this with a fluence threshold, since the observed activity rate depends on the sensitivity of the telescope and brightness of the bursts.

Line 582: "A high value means that the burst is concentrated in a small phase window, which indicates a possible periodicity pattern.": I would caution that this assumes that there is a single phase window. Many pulsars/magnetars/giant pulse sources show multiple phase windows of emission.

Line 585: "session, we then folded"  "session. We then folded"

Line 592: "which indicates the observation selection effects rather than true periodicity patterns"  "which indicates that this is dominated by observational selection effects as opposed to a true periodic behaviour related to the source"

Line 598: "the off-pulsed noise level"  "the off-pulse noise level"

Line 599: "zenith angle depended telescope gain"  "zenith angle dependent telescope gain"

Line 599: "The Kelvin unit was then converted to Jy"  "We then converted from units of Kelvin to Jy"

Line 604: "Assuming it is of flat broadband spectrum, We then"  "Assuming a flat, broadband spectrum, we then"

Line 608: "for each pulses at"  "for each pulse at"

Line 614: "The total of 79 bursts detected in 16 months show a faint trend in DM.": unclear what "faint trend" means.

Line 615: "The DMs of each day use the best-fit DM value of that day.": how was this determined.

Line 616: "More detection are needed for a detailed analysis.": for a detailed analysis of what?

Line 635: "gravitation constant G"  "gravitational constant G"

Line 642: "the uncertainty is due to the measurement": not sure what is meant by this. I don't think it adds anything to the sentence.

Line 693: "and showed some of"  "and show some of"

Line 693: "Averaged scintillation bandwidth over"  "By averaging scintillation bandwidth over"

Line 723: "The geometric factor G is": I would avoid using "G", since that was previously used for the gravitational constant.

Line 741: "Rotation measure (RM) is searched at L-band with FAST data.": why not also in the VLA data, which provide higher frequencies and large bandwidth?

Line 746: "The observed polarization could be due to an intrinsic low linear polarization.": or propagation effects in the local environment that go beyond the "normal" Faraday rotation effect.

Line 755: "we place an lower limit"  "we place a lower limit"

Line 756: "Such a large RM is even larger than that of FRB 121102, which suggests that FRB 190520B also resides in an extreme magneto-ionic environment": maybe, but it also bears mentioning that FRB 121102 is also depolarized at L-band, even when taking intra-channel rotation into account. It also bears mentioning that FRB 121102 has a highly variable RM, so be clear on whether **all** the FAST bursts were searched for an RM.

Extended Data:

ED Fig. 1:

- Given that the 1.5 GHz and 3 GHz bursts are systematically offset from each other, I'm not sure that this earlier averaging of the positions between frequencies (Line 423: "Finally, we take a weighted average of the three burst positions at the three frequency bands, weighing each position by the inverse of the total error obtained at the respective frequency band.") makes sense. How can the overall uncertainty on the position be 0.1 arcsecond when the offset between the groups of bursts at 1.5 GHz and 3 GHz seen in this figure is ~ 0.2 arcsecond?

- Why were the visibilities of multiple bursts on the same day (and/or between epochs) not combined (cf. Nimmo et al. FRB 20201124A paper)?

This would give a "coherent" summation of the burst data, as opposed to "incoherently" averaging the positions of multiple bursts.

ED Fig. 2:

- Would be better to label the x-axis "Frequency (GHz)" rather than " ν (GHz)".

ED Fig. 3:

- "Two emission lines are"  "These two emission lines are"

ED Fig. 4:

- Indicate what threshold fluence these rates correspond to.

ED Fig. 5:

- "P1 stands for \dot{p} ": why define a variable that just means the same thing as another already defined variable?

- I'm not sure that this figure is very useful. The number of period trials in the left panel must be very high, and I'm not sure if any promising candidates would be visible within the forest of other lines.

ED Fig. 8:

- Given the large scatter in apparent DMs for epochs with MJD > 58900, it is not well motivated to fit a linear trend back to a single data point at MJD ~ 58620 . Also, the large scatter in the DM values, compared to the uncertainties on each "best-fit value per epoch" is huge. Either the uncertainties are grossly underestimated, or there are huge DM variations on short timescales. This needs to be reconciled or interpreted astrophysically.

ED Fig. 9:

- "for burst P5, P46, P42, P51 that for examples"  "for bursts P5, P46, P42, P51, as examples"

- "the blue line is the best fit result": using what kind of fitting function?

- Has the zero lag been removed? Indicate this.

ED Fig. 10:

- "All 1D burst profiles have been normalized to a noise threshold of one.": I'm not sure what's meant by this, and the y-axis in the burst profile sub-panels is confusing. Why not plot S/N or flux density?

- "frequency drift,"  "time-frequency drift,"

ED Tab. 3:

- Don't need to quote 4 digits for S/N.

ED Tab. 4:

- "MJD is referenced to the solar system barycenter": TCB or TDB?

- "As mentioned in text, width values here should be considered as upper limits.": some of these widths are $\sim < 1$ ms, even though the sampling time is 10ms. I don't see how this can be a robust measurement. It's just a reflection of the fact that you're also fitting for scattering and that leads to an apparently very narrow "intrinsic width", right? I think you're over-fitting the data with a too complicated model and that the quoted widths are misleading. But in the description of the fitting (Line 429) there's no mention of fitting for scattering... so how can the fitting widths be much lower than the time resolution of the data?!

- You need to caution that the uncertainties on the DM values are underestimated; or, the paper should interpret these variations, if you consider them to be genuine.

Supplementary:

- "The P1 to P4 were detected"  "P1 to P4 were detected"

- "Arrival time of burst at the solar system barycenter.": TCB or TDB?

- "All the bursts detected on the same day are assigned the best fit DM value of the burst from that day.": of which burst?

- "We only report scattering timescales for bursts with an obvious scattering tail.": how does one judge if there is "obviously" a scattering tail, given that the scattering time is for the most part, comparable to or less than the pulse width (and given time-frequency drifts, and uncertain DM).

- The DMs appear to vary quite drastically between epochs. Either this should be interpreted from an astrophysical point of view, or the uncertainties should capture systematic effects.

Author Rebuttals to First Revision:

Second round response

AUTHORS' RESPONSE TO THE REFEREES:

The authors thank the referees for their substantive comments, which have been addressed in the revised manuscript. In addition to re-formatting the manuscript to comply with Nature guidelines (see below), we have edited language throughout the manuscript to make it as concise and direct as possible, as well as grammatically correct. The authors have also assessed all of the analyses described in the manuscript, along with all figures and tables. As a result, several significant revisions have been made to the Methods (see below). In some cases, these revisions have made specific comments from the referees inapplicable, and whenever this is the case we explain our response to the referee's comment accordingly. All other comments from the referees are taken into account in the revised manuscript.

Formatting and stylistic revisions:

- Summary paragraph: The first two sentences of the summary paragraph have been revised to provide relevant context for the manuscript as concisely as possible. The total word count of the summary paragraph is now 174 words.
- References: The main text now has 29 references. References 30-50 appear after the Methods and are numbered correctly.
- Main text statements: All statements required by Nature are now listed after the Methods reference section.
- Data availability statements: Data availability and code availability statements have been added to the main text statements.
- Figure legends: The figure legends are now listed separately after the main text references. The word counts are 107, 185, 147 for Figures 1-3, respectively. Extended Data figure legends are listed at the end of the Extended Data section.
- Display items: All display items have been assessed for their utility to the reader and for compliance with Nature guidelines. As a result of this assessment, the bottom panel of Figure 1 and ED Figure 5 from the previous submission have been removed.
- Figure formatting: Figure formatting has been revised for compliance with Nature guidelines. Panels have been labeled alphabetically and legends revised, where necessary.

The authors have decided to remove the following three analyses from the Methods, along with two related Extended Data figures:

- DM variability
- Scintillation bandwidth
- Polarization and rotation measure

These three analyses have been removed because they were not discussed in the main text of previous submissions, and are not strictly necessary to support the main results and conclusions of the manuscript. All three analyses are the subject of ongoing work, and require a treatment beyond the scope of this paper.

The authors also note that in response to referee comments on the calculation of the FRB and PRS positions, the authors have revised the astrometry to use a 1" cross-match, which produces slightly larger radio position errors. A detailed explanation of this revision can be found in the response to referee #1 below. The authors have also corrected an error in the mean total DM reported in the manuscript. The correct value of 1204.7 ± 4.0 pc/cc is obtained by averaging the best-fit DM reported for each observing epoch, which is based on the highest S/N burst detected in each epoch. The previous submission incorrectly weighted this best-fit DM by the number of bursts detected. Both the total DM and DM_host have been updated throughout the manuscript to reflect this correction.

Finally, the authors note that the last three paragraphs in the main text (discussion of the results, starting on line 146) have been revised significantly for clarity and flow. While the key ideas in these paragraphs remain the same, their presentation has been modified substantially.

RESPONSE TO REFEREE #1:

NEW COMMENTS

Line 158: “properties of FRB 190520B could be found in Table .” (missing table number).

– **Response:** Done.

Line 161: “While the observed burst repetition and morphology can be temporal...” – I think time-dependent is a more correct word to use than temporal here.

– **Response:** The description now is: “While the observed burst repetition and morphology can be time-dependent due to various mechanisms,...” We note that this sentence now appears on Line 168, as the discussion has been re-structured for clarity and improved flow.

Line 184: “Active repeaters with PRS may...” -> with *a* PRS

– **Response:** Done.

Figure 1: The inset labels are much too small, and should be enlarged for readability.

– **Response:** Done.

Figure 3: It only occurred to me when re-reading the manuscript that it doesn't appear like the uncertainty region plotted with shading on the plot has been drawn from a lognormal distribution, as is described in the Methods. Also, the MW halo DM is listed in the caption as $50 \pm 25 \text{ pc cm}^{-3}$, while elsewhere in the manuscript it is listed as $25 - 80 \text{ pc cm}^{-3}$.

– **Response:** We were previously using a simplification to estimate the uncertainty region. We have now revised the figure to show the median and inner 68% range of DM_{IGM} derived using Equation 5 in the Methods, and the caption has been revised accordingly. The halo range has also been changed to $25-80 \text{ pc cm}^{-3}$ for consistency.

SELECTED RESPONSES TO THE OTHER REFEREE REPORT

Other Referee: Some of these bursts are ~10x brighter than the brightest bursts seen by FAST. Isn't that surprising?

Author Response: For FAST, the fluences were determined using the burst signal present in the observing band. For VLA, by modeling the burst spectra using a Gaussian shape, we were able to determine the fluences of the bursts that were not completely within the observing band, and therefore these values can be higher than the burst fluences observed with FAST. We have also modified the values and errors in ED Table 4.

My comment: I don't understand how the bandwidth of a burst's spectrum should make any difference to the fluence. Fluence is a measure of spectral flux density multiplied by time, so it is measured per unit bandwidth. If the spectral flux density is being measured only within the burst's spectral envelope, then it doesn't matter whether a half, a quarter, three quarters, or the entire burst spectrum is visible – the result will be the same (at least, it would be for the simplest case of the emission being a top hat function in frequency). The author's response implies to me that the approach taken for the FAST bursts was different and simply averaged the entire FAST observing band prior to estimating the fluence (which will of course reduce the average spectral flux density, if the emission is band-limited, by diluting the burst with regions of spectrum with no emission). If that is the case, why was this done? Why not just select the region of spectrum in which the burst is “on”, which is what is effectively done with the VLA bursts? It means that the FAST bursts and VLA bursts are being represented in a totally different manner – one has the intrinsic burst fluence being down-weighted by some amount dependent on how much of the FAST bandpass it was present in, and the other does not, which would lead to the odd result highlighted by Jason that the VLA seemingly sees much brighter bursts. I think this should be rectified, because otherwise it gives the misleading appearance of an average spectral index since the VLA observations are at higher frequency. (And not enough information is provided in the paper about the FAST bursts – namely, their bandwidths - to enable a reader to “undo” the effect.) If the approach taken to estimating the FAST fluence is not changed, then a note should be added to say that the FAST fluences are biased low by it.

– **Response:** The fluence for the FAST bursts was, in fact, determined using the portions of the band where burst emission was detected (or “on”). This is now stated clearly in the Methods (see line 302). The fluence determination for the VLA has been revised to match the methodology that is used for the FAST bursts, as stated on line 391, and the fluences reported in ED Table 4 have been updated accordingly.

REACTIONS TO COMMENTS ON FIRST REPORT

Overall comment on conclusions, prior to the enumerated comments:

Original Comment:

In terms of conclusions, the primary conclusion of the paper, that FRB repeat activity level may be correlated with local source environment, is well supported. As I noted in the first paragraph of the review, I think that a logical extension of this connection is that outlier sources that might otherwise bias the Macquart relation between DM and redshift may be identifiable (and potentially even calibrate-able.). It would seem logical to comment on this directly in the manuscript.

Author Response:

Thanks for pointing this out. We add the comment to the conclusion:

“Further study of such correlations may help identify outliers to the Macquart relation and potentially help calibrate biases.”

My new response:

I’m going to nitpick here and say that this wording does not necessarily convey the full depth of what is possible if there is a relation between some other FRB observable (or combination of observables) and the host DM contribution. When I said “outlier sources” in my initial report, I meant sources whose host DM contributions are outliers (relative to the typical host DM in the broader FRB population). I realise that was a bit contextual and could have been interpreted as outliers to the Macquart relation, but I would argue that the Macquart relation relates extragalactic DM (after stripping out the Milky Way and estimated host galaxy) against redshift, hence it doesn’t make sense to talk about a large host galaxy contribution causing an outlier to the Macquart relation – that will only happen if the host galaxy contribution is mis-estimated. Anyway, I would suggest a phrasing such as “..identify outliers in the host galaxy DM contribution to observed FRBs, and potentially help calibrate effects that would otherwise bias the Macquart relation” is clearer. Basically, the outliers are in the host DM contribution, the removal of which will calibrate away a bias to the Macquart relation.

– **Response:** Due to restructuring of the discussion, the original sentence “Further study of such correlations...” no longer appears in its original form, but we have considered the referee’s comments in our revisions to the paragraph on lines 128-133 and the last sentence of the main text on lines 169-171.

Overall comment on the RM, prior to the enumerated comments:

Original Comment

On the other hand, the extremely important result that the RM may be very high (like 121102) is referred to nowhere in the main text, despite this being (if confirmed) an extremely important additional link between the two sources in terms of their local environment. If the results of the analysis are not referred to in the main text, why is this in the Methods?

Author Response

The claim that the RM may be very high is only one interpretation of the FAST result. Higher frequency bands and more observation will be needed, per experience with FRB 121102 (no RM detection by FAST in L-band). We still leave it in the Methods in case readers may be interested in the RM results so far.

My new response

I guess it is up to the journal whether they consider it appropriate to have material in the methods that is not referred to in the main article at all and not necessary to support any of the conclusions. My opinion is that either it should be referred to in the main text or dropped from the Methods, but the editors can rule on that.

– **Response:** The authors have decided to remove the section on RM from the Methods, as it is not discussed in the main text and is not critical to the main conclusions of the manuscript. An in-depth RM analysis is currently in preparation and will be reported on in detail in a future paper.

UNRESOLVED ORIGINAL ENUMERATED COMMENTS IN ORDER OF APPEARANCE

1. In the abstract, a number of quantities are used without context or insufficiently defined. Fast Radio Bursts themselves are introduced with no context. The term “repeater” is introduced without context, as is “persistent radio source” (noting my earlier point about defining PRS precisely). “The estimated host galaxy contribution DM_{host} ” is similarly imprecise: at this stage, DM (Dispersion Measure) has not been defined, so it would not be clear to most readers what this “contribution” is. Grammatical

errors abound: Fast Radio Bursts (FRB) should have “FRBs” within the parentheses, line 44 should read “...to be associated with **a** persistent radio source”, no space before “Here” on line 45, line 51/52 should read “...with **a** confirmed association between **an** FRB and **a** compact PRS”, etc.

Author Response

Thanks!

The summary paragraph has been restructured and some contexts were added. Some definitions have to wait for the main texts due to length limitations.

“Fast radio bursts (FRBs) are the most energetic radio transients in the Universe, the central engines of which remain unknown and could be diverse. The dispersion sweeps of FRBs provide a unique probe of the ionized baryon content of the intergalactic medium as well as FRBs’ natal environments.”

My new response

Another nitpick, but FRBs are not the most energetic radio transients. Longer-lasting events such as TDEs give off much more energy in the radio, just over a longer period of time. I’m not sure how the authors want to present this (most luminous fast radio transients?), but it should be clarified.

– **Response:** The first two sentences of the summary paragraph have been revised and no longer use this terminology.

Later in the paragraph, “The estimated host galaxy dispersion measure (DM) $DM_{\text{host}} \approx 902+88-128 \text{ pc cm}^{-3}$ is nearly an order of magnitude higher than the -50 average of FRB host galaxies, and much larger than those of the intergalactic medium” – the part of the sentence after the comma doesn’t make grammatical sense. Perhaps “and is much larger (for this source) than the contribution from the diffuse intergalactic medium.”, but I’m not exactly sure what this part of the sentence is really trying to say.

– **Response:** This sentence has been revised for grammatical clarity.

Finally, I still think the word “repeater” is not something that can be used without definition in an abstract of a non-astronomy journal. “repeating FRB” would be clearer.

– **Response:** “Repeater” has been modified to “repeating FRB” throughout the manuscript.

2. Line 68:

using an uncertainty in arcseconds for right ascension right next to the position (in seconds) is rather confusing. I suggest that either giving the uncertainty in seconds, or quoting the full position (RA+Dec) followed by an uncertainty ellipse in arcseconds, would be less prone to misinterpretation. Moreover, it is stated that the uncertainty is dominated by systematic considerations, but this is never shown for the FRB in the Methods (it is shown for the PRS). Finally, I disagree with the uncertainty presented for the FRB – how can the systematic uncertainty on a position which is obtained from a handful of milliseconds of exposure time have a smaller systematic uncertainty than the PRS, which has hours of exposure time? In a best case, they would be identical, but as I discuss in the methods comments below, I think the systematic contribution to the FRB position uncertainty has been given insufficient attention.

Author Response

We have now included the errors quoted separately from the uncertainty to help avoid confusion. Regarding the uncertainty, we have also changed a few things but those are noted in your later comment that discusses these issues in more detail.

My new comment

“persistent, radio continuum...”: drop the comma

– **Response:** Done.

5. Line 87/88 – both R and R’ band are referred to. Which was used? I note that R’ and R are again both used later in the Methods section. Line 89: How is that H-alpha luminosity derived? It is not derived anywhere in the Methods.

Author Response

The phrase “extended R-band structure” includes a typo. It should be R’-band. We have corrected that typo in the text. Thanks! The H-alpha luminosity is derived based on the luminosity distance for $z = 0.241$, and the extinction corrected H-alpha flux. The detailed approach of extinction correction on the H-alpha flux will be included in a subsequent paper in preparation.

My new response

But where is the extinction-corrected H-alpha flux presented? Extended Data Figure 3 shows the flux not corrected for slit loss or extinction correction, and then suddenly the final extinction-corrected H-alpha flux appears on line 650 when deriving the DM contribution, but the luminosity itself is never derived in the Methods. Surely this should be developed clearly in the Methods (first the raw flux should be stated, then extinction and slit loss corrections applied, then luminosity derived from that corrected flux, and then later the DM also derived from the corrected flux)? (and the main text should refer to the methods?)

– **Response:** Essential information about the derivation of the H-alpha luminosity has been added to Section 4 of the Methods. We have also corrected an error in the H-alpha flux quoted in the Methods (line 600). The original value incorrectly used the angular diameter distance rather than the luminosity distance to determine the flux from the measured luminosity. We have also corrected all subsequent estimates of the DM implied from the H-alpha EM, and any subsequent estimates that rely on the H-alpha flux. The corrected DM estimates can be found on lines 615-633, and are about a factor of two smaller than our original values. This correction does not significantly change our overall interpretation.

6. Line 101: There is no reference to the Methods section where the DM_IGM number is calculated. I also am unable to reproduce the reasoning that yields the estimated value and range for DM_IGM, but we shall return to that in due course.

Author Response

The main text and Methods section now elaborate on how the calculations were done, including errors in the MW estimates (NE2001 error and range of possible halo contributions to DM). The error ranges we use are generous (i.e. conservative) and indicate, also including cosmic variance in the IGM contribution, that there is no way to avoid the conclusion that the host-galaxy DM is large.

My new comment:

I agree with the conclusions, and the changes have brought some improvement, but I think there is still some room for improvement in the presentation. Specifically:

Line 110: I think that “Increasing the ionised fractions from” should be “Varying the ionised fraction between”, since the initially used value was 0.85, not 0.6. Also, in the next sentence, the value calculated for the range doesn’t look quite right – it looks like only the mean value of DM_IGM was varied with f_IGM, rather than varying sigma_DM_IGM also, which would increase the range just a bit more. Why is the DM_IGM then presented in the form X to Y, rather than mean +pos – neg as it was in the previous sentence?

Going to those calculations in the Methods: On line 639, the result for sigma_DM_IGM implies that it is linearly dependent on f_IGM, while the preceding equation implies that sigma_DM_IGM should depend on the square root of that quantity. So I think these numbers, while they won’t change materially, need to be checked and harmonised. Finally, at line 643/644, it would make more sense to swap the order of the sentences and calculate DM_IGM first and then DM_host (since DM_host is calculated by subtracting DM_IGM and DM_MW+halo from DM_observed.) And it would make sense to perform the calculations that are reported in the main text (i.e., calculating DM_IGM when varying f_IGM also, thus coming up with the final number that is reported in the main text.)

– **Response:** We have changed ‘increasing’ to ‘varying’ even though the sentence in the previous version was grammatically and factually correct. The analysis (now and previously) varies both sigma_DM_igm and mean(DM_igm) because the latter is used to calculate the former, as discussed in Methods section ‘DM Inventory Analysis.’ DM values are now presented with the same format x_y^z . The typo in Methods (previous version) that implied sigma_DM_IGM \propto f_igm is now corrected. That typo was in the manuscript not in the analysis code, so it had no effect on the results. The IGM DM value is now stated before the DM_host value. We also have added the results for DM_igm when f_IGM is varied to Methods.

8. In this same paragraph, the predicted scattering time in the event of a chance background source is only two orders of magnitude greater than observed. In the Milky Way, scattering times show a very large degree of variability. I would suggest that the (lack of) scattering on its own is not conclusive evidence to rule out a background source, but combined with the low Pchance of the host galaxy association, it further strengthens the already very strong case.

Author Response

The case is stronger than stated in this comment. Yes, the Milky Way produces a wide range of scattering times, but that is over a wide range of dispersion measures. For a specific value of DM, e.g. 500 pc/cc, we see a total range that might be as large as two orders of magnitude if outliers are included but the probable range is only about 1.5 orders of magnitude. is the same. Since the host galaxy DM appears to be > 500 pc/cc (in its frame, not the observer's), we would On geometrical grounds, the scattering from an intervening galaxy would be more than 10^4 times that of a host galaxy if the scattering medium expect for a Galactic pulsar a mean scattering time of 9.6 ms at 1 GHz and a probable range from 1.6 to 54 ms. These numbers are from a fit to $\tau(\text{DM})$ for Galactic pulsars that appears in several places in the literature (e.g. <https://arxiv.org/abs/2108.01172> equation 8 and references therein, e.g. Krishnakumar et al. 2015, Bhat et al. 2004, NE2001 2003). For an intervening galaxy that produces the same DM of 500 pc/cc, we include the geometric factor of 10^4 and another factor of 3 that corrects the Galactic prediction for spherical waves from a pulsar to plane waves for an extragalactic source, and we divide by a $(1+z)^3$ factor to get the scattering time in the observer's frame.

This gives:

$\tau(\text{intervening}) > 10^4 * 9.6 \text{ ms} * (\text{geometric factor} = 3) / (1+0.241)^3 = 150 \text{ sec}$ at 1 GHz compared to 40 ms for the measured scattering time (scaled to 1 GHz). This is more than three orders of magnitude larger than observed. Galactic variance can reduce this by a factor of $(1.6 / 9.6) = 0.2$ or increase it by a factor of five.

An alternative might be to say that an intervening halo produces low-level scattering that, with the 10^4 geometric factor, multiples up to the observed scattering. That would mean that most of the DM not from the IGM and MW would be from the host galaxy. That would produce some scattering but perhaps not as much as observed due to the redshift factor and the greater IGM contribution if the host is at $z > 1$, say. The problem with this configuration is that there is no evidence for *any* scattering from galaxy halos, certainly not from the MW nor from M81's halo.

All told, the scattering constraint strongly favors scattering from a host galaxy at $z=0.241$ with a large DM contribution.

For completeness, one might argue that the association of the $z=0.241$ galaxy is completely unaffiliated with the FRB source and has no influence on the line of sight. Then all bets are off, but it would be a (unlikely?) coincidence that the H-alpha measurements yield a DM estimate that is compatible with the inferred DM_{host} if the host is in fact at $z=0.241$. A spurious association seems to be a more convoluted interpretation than the one we put forward in the paper and where the scattering is in fact constraining on the presence or not of an intervening galaxy.

My new comment:

I think the changed text dealing with scattering is an improvement. I note however that in the methods, I think there is an error of three orders of magnitude in equation 9. It seems to mix and match equations 1 and 2 from the Koch Ocker paper – the constant should be in nanoseconds if all the lengths are in the same units (their equation 1). The constant only goes to microsec if the distances to the galaxies are in Gpc and the scale length is in Mpc. If using the formalism from the second equation in the Koch Ocker paper, then the units of the various distances and lengths should be clearly stated in the Methods text here (lines 724 – 726), which is not currently the case. Also, the value assumed for \tilde{F} (along with where that is obtained, and what its uncertainty might be) should be stated, to enable the reader to reproduce the numbers that are obtained (it looks like that was removed in the text change). The uncertainty in \tilde{F} is nearly as important as the unknown length scale L ; driving \tilde{F} down and L up to their minimum and maximum reasonable values respectively can actually get to quite a low value of scattering time. I agree that it isn't very likely, but the reader should be presented with all the facts to judge for themselves.

– **Response:** The numerical pre-factor in Equation 9 of the Methods is indeed for distances in Gpc and the path length in Mpc -- we have modified this text accordingly. We have added text in the Methods explaining the range of \tilde{F} in the Milky Way, which is the only galaxy for which we have well-calibrated knowledge of the fluctuation parameter based on the pulsar population. The expression for G_{scatt} also had a typo which has been corrected; it now includes a factor of 2 which was omitted in the Ocker et al. paper (see Cordes et al. 2021). We have also corrected the scattering time estimated from the H-alpha implied DM, which now uses the correct H-alpha flux. We have modified our conclusions to acknowledge the latitude in this scattering estimate. While our fiducial estimates still suggest that the observed scattering is smaller than the scattering expected from an intervening galaxy, our ability to distinguish between these two scenarios based on the scattering alone involves a number of parameters that are not well constrained, and we have endeavored to make this clear to the reader, both by modifying the main text and the Methods.

10. Line 139/140: DM_{host} can also be observationally biased, as it contributes to an observational quantity (the total DM) to which different FRB searches have different sensitivities.

Author Response

We agree that large values of DM_{host} are less likely to be detected, since FRB searches are usually sensitive to a limited range of (total) DM. We have modified this sentence to reference this possibility.

My new response:

It's even worse than that, though, because it is systems where the total DM is largest (so – distant FRBs with a large host DM) that are most affected – not all FRBs with large host DMs will be affected equally. Nearby FRBs with large host DMs might be mostly detectable. This is just to say that the situation is complicated, and hence the sentence that immediately follows this revised one might be a little glib – a large sample on its own won't be enough, the selection effects need to either be negligible (systems with good sensitivity out to high DMs) or clearly tracked.

– **Response:** The language on lines 130-132 has been clarified to address this point. We also note in the text that large host DMs may imply large scattering, which can reduce search sensitivity. This effect will actually be worse for lower redshift sources.

12. Figure 2: I think that either a zoom in (for one of the bands) and/or some annotations of other nearby sources would be valuable. This goes to point 4 above – are any of the other relatively nearby sources larger galaxies that may have a reasonable likelihood of harboring an FRB in their outskirts?

Author Response

We thank the referee for the suggestion. Here we provide the zoom-in version below in response. The members in our collaboration have considered the zoom-in version of Figure 2, but we still prefer the current plot configuration to display the relations between radio and optical emissions in the field. The zoom-in plot will be discussed in more detail in the follow-up paper soon.

The persistent radio emission and repeating FRB bursts are all coincided with the optical emission shown in the Figure 2 within the position uncertainties and the resolutions of the optical/NIR images, the probability of false association is only $< 1\%$. The next nearby object in J-band is $6.5''$ to the north of the FRB location. The same method put the false association to be $> 20\%$, which translates to the object being likely just in the neighborhood coincidentally.

My new response:

That is useful information – why not put that (the chance coincidence of the next nearest object) in the Methods too? A chance coincidence probability on its own is not always sufficiently informative. If the Pchance of the next nearest galaxy was 1.5% rather than $>20\%$, would you still feel confident claiming an association with the preferred galaxy just because it had a Pchance of 0.8%? I wouldn't. While I still think that a Bayesian approach that incorporates all the galaxies nearby is the best one, in clear-cut cases like this mentioning the Pchance for the next nearest galaxy is sufficient to reassure the reader. So I would strongly suggest including that information in the Methods.

– **Response:** The next nearest galaxy is a J-band only object with marginal detection. This source is not detected by deep optical imaging (covering up to 7041A) from CFHT, making its D4000 (Balmer break at $\sim 4000\text{\AA}$) redder than 7040A, which implies a redshift $z > 0.75$. At that large redshift, the angular separation of 6.5 arcsec implies a projected distance of 50 kpc between the center of the J-band-only object and the FRB, which is unlikely. The chance coincidence probability for this galaxy and the FRB is $>20\%$, as noted in our previous response, and we have now added this information to the Methods as requested (see lines 517-518).

14. Line 283: Why would some bursts show scattering and others not? Surely the simplest assumption is that the scattering is time independent, which would enable this parameter to be fixed across all bursts and constrained with good precision, which would then lead to a better precision on the (deconvolved) gaussian burst widths for all bursts? Non-gaussian intrinsic structure will surely complicate the analysis, but it will also be complicating the present analysis (where the choice of scattering vs no scattering is apparently made by eye) already, and the assumption of constant scattering is at least physically justifiable.

Author Response

The band-limited nature of the bursts means that some will be more scattered than others, depending on the center frequency and frequency extent of their measured flux density. This could have been dealt with by adopting a fitting function for all bursts with a single value of the scattering time and then taking into account the frequency range for each burst. We have reason to investigate, however, that the scattering time actually is different for different bursts. It is entirely possible that FRB environments can produce variable scattering, similar to what is seen for the Crab pulsar giant pulses, and a patchy medium can even produce different amounts of scattering for bursts that are closely spaced in time. The measured scattering time can also in fact be influenced by refraction or lensing that works in tandem with diffractive scattering, and this refraction/lensing can also be time-variable. Sorting out these possibilities is beyond the scope of the present paper but we are actively investigating it.

My new response:

I will concede that time-variable scattering is certainly possible due to the changing line of sight through a turbulent medium. I stand by my assertion that time independence is a better default assumption unless shown otherwise, though.

Now that the “fold the frequency” confusion is cleared up, I understand that no frequency dependence was assumed for the scattering when fitting individual bursts (the new text on line 704 makes this clear.). However, I can't see how that approach can be justified, since we know that scattering will be very strongly frequency dependent (ν^{-4}). The only defense I can imagine is that most of the bursts have a reasonably narrow fractional bandwidth, but even for a burst from say 1000-1200 MHz, the scattering is going to be 2x higher at the bottom end of the burst than at the top.

Table 1 and ED Figure 11 both refer to scattering times at 1.25 GHz, so I assume (although it is not stated, and should be) that the scattering times measured for individual bursts have scaled to this reference frequency based on the central burst frequency in each case and an assumed power law. But maybe they haven't been scaled at all? In which case I can't see how or why the times for different bursts with different central frequencies can be compared to each other in any useful way. I would have expected ν^{-4} to be assumed, but this should be stated clearly, both in the Methods text, and in the extended data figure 11 caption.

Anyway, looking at extended data Figures 10 and 11, I am not convinced by the claim that selecting sources by eye in order to choose whether to fit an unscattered gaussian or a scattered gaussian template is more robust than assuming a constant scattering time (appropriately scaled with frequency, of course). In Figure 11, all the measured scattering times are remarkably consistent with the mean value, given the (necessary) gross oversimplification of the intrinsic pulse shape to be a gaussian in every case. If there was a counter example of a narrow burst whose measured width is too low, clearly inconsistent with this scattering, that would be a good counter argument supporting the variable scattering, but nothing in Figure 11 supports that presently. P31 has clear intrinsic structure (so I don't think there is evidence that that burst is over-scattered, relative to the mean), P47 might be on the narrow side but it is at relatively high frequency – what is the goodness of fit of an unscattered component compared to a component scattered with the mean scattering time? – and P50 is intrinsically wide enough that I wouldn't expect to see the effects of the mean scattering time, but again the goodness of fit of an unscattered vs scattered single component could be compared.

To summarise: Each burst could be fit one of three ways: 1) with a gaussian only, 2) with a gaussian convolved with a (freely fitted, frequency dependent) exponential, 3) with a gaussian convolved with a frequency dependent exponential whose scale was fixed by fitting the ensemble of all bursts. Option 2) has one more free parameter than options 1) and 3). At the moment, either option 1) or option 2) is chosen in an ad-hoc manner, burst by burst. I can't see how that can be justified compared to choosing a physically motivated default model (option 3) and evaluating all three models and only choosing option 1 or 2 if they return a goodness of fit that is significantly better than the default (in which case there is now clear evidence for time variability of the scattering – at least if the better fit is definitely not just resulting from intrinsic burst structure.)

Ultimately, what is this used for? For this paper, only to estimate the mean scattering time. That would be better done with a single fixed scattering kernel for all bursts – choosing only the bursts with apparently visible scattering would, if anything, bias that mean result high. The time dependence of scattering, if present, is not actually investigated in the main paper. Given that, the probably slightly biased view of the scattering that would result from the ad hoc modelling choices has a pretty negligible impact on the conclusions, but others may take the data in this paper and use it for other purposes. So while my suggestion is to do the modelling consistently, if it is not changed it at least should be clearly explained (the frequency dependence vs not of the scattering kernel) and the potential impacts of the ad hoc choices listed. And it is absolutely imperative to clearly describe how the results were scaled to a reference frequency of 1.25 GHz. If, as seems possible, no scaling was done, then this really must be justified (although I don't really see how it could be.)

– **Response:** The referee raises a number of important questions which require clarification both here and in the paper itself. First, we note that the Methods section on scattering has been significantly modified to address the referee's concerns. In particular, we address the previous ambiguity in how the subset of bursts with reported scattering times was chosen. All 79 bursts in this paper were initially fitted with a 1D burst profile model consisting of a Gaussian component convolved with a one-sided exponential. Of the resulting 79 scattering times, only 26 had fractional uncertainties <50%, and these are the scattering times we report in the Supplementary

Table and use to calculate the mean scattering time. The details of this procedure are outlined in the Methods. Below we delineate selected responses to the referee's comments, in the order by which the referee gave them:

1. The burst spectrum is highly variable between bursts, both with respect to the burst temporal width and the frequency bandwidth, and preliminary evidence suggests some of this spectral variability may be related to variable scattering. For these reasons, the scattering time is not fixed across all of the bursts. While some of the bursts with high S/N show frequency-dependent asymmetries consistent with scattering times ~ 10 ms at 1.25 GHz, two of the bursts in the sample are symmetric across the radio frequency band and have burst widths (corresponding to the FWHM of a Gaussian fit) that are less than 7 ms. One of these narrow bursts is shown in the Extended Data Figure. Several bursts are similarly symmetric but have full-width-at-half-maxima significantly broader than 10 ms. The narrow bursts suggest that the scattering time may vary by at least ~ 3 ms at 1.25 GHz, but a detailed analysis and interpretation of this apparent variation is beyond the scope of this paper. A follow-up paper on this exact topic is currently under preparation.
2. The referee is correct that integrating bursts across the entire frequency band in order to determine a scattering time from the average burst profile carries significant uncertainties. We note this in the Methods, and a more robust scattering analysis is the topic of another follow-up paper currently in preparation; this subsequent analysis verifies the frequency dependence we expect from scattering, which corroborates the scattering interpretation presented in this manuscript. Nonetheless, the current analysis is sufficient to demonstrate that the mean scattering time is likely too small for the galaxy to lie in the foreground, which is the main use of the scattering time in this paper.
3. All scattering times are scaled to 1.25 GHz assuming a ν^{-4} frequency scaling; this has been clarified in the Methods.

15. Line 284: what is "the sub-pulse"? presumably "a sub pulse", but this needs to be much more carefully defined. What level of significance is used to define a sub-burst that should be fitted (currently "the noise baseline" is stated)? At what width?

Author Response

Thanks! We change the term to "a sub pulse". Since there is no standard pulse profile (nor average) for repeaters yet, the use of pulsar terminology is empirical. Currently, when the 'bridge' between the two closely-spaced-in-time peaks drops more than 5 sigma below the higher peak, we consider them to be two bursts. Otherwise, they are considered structures within one event (sub pulses).

My new response:

Okay, good to know – but please put that information in the Methods text, so all the readers know that, not just me! Also, my original point was meaning that grammatically, the sentence should begin with "A sub pulse", not "The sub pulse".

– **Response:** Done (see lines 298-299).

20. Line 351: How is the time selection of the de-dispersed visibilities achieved? Is any weighting applied based on the fitted width of the burst, or is a top-hat selection function applied?

Author Response

- We used a top-hat selection function.

My new response:

Please add that information to the text!

– **Response:** Done.

21. Line 364-372: it is naive to take the statistical image-plane fit uncertainty as the sole source of uncertainty for an individual burst, as there will surely be calibration errors leading to systematic position uncertainties. This likelihood should be acknowledged and estimated (the recent papers on FRB20201124A mostly include reasonable discussions of systematic uncertainties on position estimates). It is then even more naive to take a simple weighted mean of all of the burst positions (with statistical-only uncertainties) estimated to arrive at a final FRB position and uncertainty. An estimate of systematic uncertainty in the VLA position must be presented at the same time as that final statistical uncertainty.

Author Response

Thank you for pointing out this omission (and the implicit disagreement in our numerical reporting). We have now added in words here to describe how the systematic uncertainty for the FRB bursts is the same as the systematic uncertainty for the PRS (because, indeed, they use the same data and are subject to the same systematics). To summarize briefly also here so you don't have to find it in the draft, essentially we do frequent phase-referencing with a small enough interval that any short-timescale variations will be fitted out by our phase calibration. Thus, the short-timescale systematics should reflect the same systematics as the deep image. Additional note: while we didn't write this extra analysis into the draft, we also confirmed that the phase wander was consistent on short timescales by performing short-timescale imaging (imaging segments of 5-30 seconds, such that at least 1-3 sources could be detected at each frequency) and inspecting the position variations of those sources over time (positions were stable and consistent with statistical uncertainty; offsets of the sources were within the range of PRS-quoted systematics). While this could not be as rigorous as the deep image because each field only had 2-3 detectable sources on those timescales, it at least reassured us of consistency for the FRBs.

Your comment also brings up another point you also mentioned elsewhere, which is that we didn't take the full (statistical+systematic) uncertainty into account in the weighted mean. We agree with this assessment and have changed the position calculation and reported error:

- a weighted mean is now calculated for each frequency separately, with the weights scaling with the inverse statistical error of each measurement. The error on each of those frequencies is then the propagated statistical error, added in quadrature with the systematic error at that frequency (as determined by the PRS analysis).
- The three frequency measurements are then averaged, with the appropriate error propagation (for the mean of three values with error) quoted.

We have accordingly changed the text to reflect this change. Because our FRB position was performed as a three-frequency mean, accordingly it is appropriate (particularly in a comparison between the PRS and the FRB emissions) to perform the same operation---a weighted mean of the positions at the three observing frequencies---for the PRS. The errors are now consistent and the PRS position-fitting description has been changed accordingly to reflect this change. Text throughout, have been updated to reflect these changes.

My new response

I think these changes are an improvement, however, there are a few things that remain to be addressed. First, using a cycle time of 10-15 minutes at B array does not guarantee that short timescale phase errors will be fitted out. The timescale of the phase noise depends on atmospheric conditions, etc, and in any case there is also still the spatial extrapolation from the calibrator to the target direction. The recommended cycle times for the VLA ensure that the data should be phase-connectable in normal observing conditions, but not that the residual astrometric errors will be negligible. If the authors can show (with a couple of brighter sources) that there are no astrometric residuals inconsistent with thermal noise on those 5-30s timescales, then that allays the concerns – but that information (the consistency of the short timescale imaging) should be included in the Methods text.

– **Response:** Indeed, as previously mentioned we did imaging of the brightest source(s) at each band on timescales from 5-30 seconds, and indeed saw a distribution consistent with only radiometer noise. We have now added comments along these lines into the text in the “determining the properties of individual bursts” part of the Methods.

Second, the FRB positions themselves offer a natural way to check this. Are all the individual positions consistent with the weighted mean (what is the reduced chi-squared of the individual positions compared to the weighted mean?). By eye from ED Figure 1, it looks like the reduced chi squared would be >1 , but not by very much. (The L band positions look internally consistent, and consistent with the single C band position, but marginally inconsistent with the mean of the S band positions). Including that information (the reduced chi squared of the final weighted mean) would similarly improve the confidence of the final result; there is no good reason not to include it.

– **Response:** By comparing the individual burst positions to that of the weighted mean, we obtained a reduced chi-square value of 0.53 and 0.77 for R.A. and Dec. respectively. We have added text to the Methods section regarding the same.

22. Line 391-393: Why was the PRS fit only with a point source model? If the source size is consistent with being unresolved then this is reasonable, but surely a gaussian fit should also be attempted to place an upper limit on size (at each frequency)?

Author Response

We actually took gaussian fits using the CASA imfit task for the PRS at multiple bands (L, S and C bands). The returned results showing that the source is a point source at L and C bands, with the size may as large as (1.4", 0.89") and (0.36", 0.1"), and a component with size of (0.51", 0.14") at S band. We've revised the initial improper expressions.

My new response:

Ok, this is good – but these upper limits to the PRS size should also be included in the Methods text somewhere (and, ideally, in the main text too! The main text says “unresolved” at some point, but doesn’t give a maximum size). At the moment, the text merely says that imfit was used, and while the results show the positions, they don’t show the size limits.

– **Response:** We have added the fit sizes of the source across the VLA bands and give a maximum possible size obtained from the VLA C-band in the main text and at its angular diameter distance in Methods.

23. Line 410: Which direction are the systematic offsets? VLA-PanSTARRS or vice versa? Has the correction been applied to the final PRS position that was reported, or not?

Author Response

The offsets are obtained by subtraction of PanSTARRS coordinates from VLA coordinates.

We have applied systematic corrections to the PRS positions, thus the position errors are statistical errors added in quadrature with systematic errors. In the Extended Data Figure 1 we applied the systematic corrections of all bands to the PRS positions to give an idea of the consistency of the positions of the bursts and the PRS.

My new response:

OK – so this information (the direction of the systematic offsets) should be included in the Methods section. Also, and apologies for not noticing this the first time, but why was a cross-match of 0.5” used? That basically imposes a prior that the true offset is (quite a bit) less than 0.5”. I don’t have any reason to believe that the astrometry would be worse than the measured value, but it would be good to confirm that the results don’t change substantially if a looser cross-match (e.g., 1”) was used.

– **Response:** We have investigated the issue you raised and conclude that a 1” cross-match radius is most appropriate. We tried cross-matching with radii of 0.5”, 1”, and 2”, and found that going from 1” to 2” added source counts consistent with chance coincidence, as summarized in the table below.

The VLA images have 375, 113, and 43 sources in the 1.5 GHz (L band), 3 GHz (S band), 5.5 GHz (C band) deep image, respectively. In the optical, there are 3619 PanSTARRS DR1 sources within 10’ of the VLA field center, implying a 1% chance of false association when using a cross-match radius of 1”.

Cross-Match Results:

Radius | L-band | S-band | C-band

0.5” | 109 | 27 | 9

[0.5” | 0.9 | 0.3 | 0.1 expected by chance]

1” | 27 | 4 | 0 additional sources matched (over 0.5”)

[1” | 3.8 | 1.1 | 0.4 additional expected by chance]

2” | 11 | 1 | 0 additional sources matched (over 1”)

[2” | 15 | 4.5 | 1.7 additional expected by chance]

Therefore, in the revised analysis we choose a cross-match radius of 1” to estimate the systematic offsets without significant bias. The estimated systematic offsets are now:

L band:

1” (136 sources matched) cross match: ra: 0.0373” +/- 0.2460”; dec: 0.0144” +/- 0.3035”

S band:

1” (31 sources matched) cross match: ra: -0.0696” +/- 0.2772”; dec: 0.0382” +/- 0.1552”

C band:

1” (9 sources matched) cross match: ra: -0.0947” +/- 0.1161”; dec: 0.1069” +/- 0.0554”

We have made the relevant changes in the text by updating the average positions of the bursts and the PRS in the text, tables, and figures. The most accurate PRS position is now obtained at C-band. We have also recalculated the chance coincidence probability for the PRS and burst association (see “Methods: Persistent Radio Source”).

33. Line 509: The text and references do not actually justify the range quoted for the IGM DM. The quoted reference considers f_{IGM} in the range 0.6 ± 0.1 , not 0.8, and in any case provides no hints as to how cosmic variance was translated into this DM range. This justification should be strengthened. Also, throughout this section (and indeed, throughout the entire paper) confidence intervals should be specified (or if are typically 68%, then this should be stated somewhere).

Author Response

The range quoted for the IGM DM is now justified and confidence intervals are stated to be 68% intervals.

My new response:

This is definitely improved, although a primary reference for the f_{IGM} should be given rather than a secondary reference (Zhang et al 2018 simply cite Fukugita et al. 1998 to support their value of 0.83).

– **Response:** The primary reference Fukugita et al. has been added to the Methods along with the Zhang reference.

34. Equation 3: Half of the symbols used in this equation are not defined.

Author Response

Thanks! The symbols are now defined.

My new response:

I presume Ω_{Lambda} in the text is meant to be Ω_{lambda} in equations 4 and 5 (or vice versa).

– **Response:** Equation 3 in the first submission is Eq 4 now, and Ω_{Lambda} is now defined and consistently used with capital lambda.

35. Line 514: the uncertainties presented for DM_{host} are clearly inaccurate, having presumably neglected the uncertainties on the MW halo (and made no attempt to estimate an uncertainty on the MW contribution). The number of significant digits is also incorrect. At no point in this section is the derivation of the final value of $912 + 69 - 108$ that is quoted in the main text shown. Whatever the number is in the main text should be supported by the analysis here, and it is not. How were the multiple sources of uncertainty combined?

Author Response

No, the uncertainties were not neglected. The uncertainties are now itemized to clarify.

My new response:

The uncertainties are now different to the original value though? In the initial submission DM_{host} was $912+69-108$, now it is $902+88-128$, so something was updated?

As I mentioned in a previous response, it seems like the expression for σ_{DM_IGM} under what is now equation 5 is incorrect, although I suspect that it is only the expression that is written incorrectly and the value was used was correct. But the more salient point is that this expression gives a single value (the standard deviation) for the DM_IGM , while the final uncertainty is DM_IGM is asymmetric (as it should be) – it would aid the understanding of this passage of text to reorder the text such that the calculation of DM_IGM and its (asymmetric) uncertainties follows immediately after the description of the standard deviation of the lognormal distribution, and only then go on to calculating the host DM. So basically bringing the sentence on line 633/644 up to line 639.

– **Response:** This section has been revised to clarify the methodology (see line 582 on). The previous results were correct but we are now using a slightly updated value of observed total DM ($1204.7 \pm 4.0 \text{ pc cm}^{-3}$). There was also a typo in the expression for σ_{DM} that is now corrected so it scales as the square root of f_{igm} . Also we have added explicit expressions that relate the mean DM_{igm} calculated from equation 5 and the RMS DM_{igm} to the parameters σ and μ for the log-normal PDF. This makes clear the distinction between the parameters of that distribution and the net mean and RMS (or variance) of DM_{igm} .

36. Line 517: Where does the extinction-corrected H α flux come from? Which spectrum? What was the extinction correction? The caption of extended data figure 4 says that the Palomar spectrum that is plotted was not corrected for slit loss, if it comes from this spectrum, was that correction applied, if so what was the result?

Author Response

The extinction-corrected H-alpha flux is from the Keck spectrum. We were using the Balmer decrement line ratio H-alpha/H-beta = 2.8 in case to estimate the extinction against H-alpha emission. The detailed analysis is included in the associated host galaxy analysis paper that will be submitted soon. The Palomar spectrum is not corrected for the slit loss since we mainly used it to determine the redshift but the H-alpha line is consistent with the Keck/LRIS result. The Keck LRIS spectrum was not corrected for the slit loss either because the compact host galaxy with a seeing condition of 1.1" is considerably smaller than the slit width of 1.5".

My new response:

Ok. Please clarify what was done in Section 4! Section 4 should say which spectrum is being used to for the final H-alpha estimates used subsequently, what corrections were applied, etc. I can appreciate that more details may be forthcoming in a subsequent paper, but these are pretty fundamental.

– **Response:** This section has been modified for clarity, and explains how the H-alpha luminosity was derived from the Keck spectrum (see lines 501-504).

43. Line 542-550: As already noted in point 14, surely the simplest assumption possible would be for no time dependence to scattering, and it would make more sense to fit a single scattering time to the entire ensemble of bursts? That would probably give a better estimate of the scattering time, and then also a better estimate of the distribution of widths. Otherwise, the widths of the bursts for which no scattering time was estimated (and which were then fitted by an unscattered gaussian) are very likely to be biased high. Since the main purpose of this analysis is to estimate a scattering time in order to rule out the observed galaxy being a foreground source, obtaining a mean value this way, by fixing it across all bursts, must surely be preferred.

Author Response

Please refer to the response to point 14. We do not fix the scattering time across all bursts because we see clear evidence of scattering in some bursts and not in others. Time-variable scattering can be explained by a patchy medium near the source.

My new response:

See the response to point 14.

– **Response:** All future correspondence on this point is referred back to point 14 above.

49. Extended Data Table 5: I'm curious how an apparently 20 sigma burst (burst C5) can be identified to have a width of ~0.1 ms with an uncertainty of 70 microsecond, given the 10ms sampling of realfast? All of the other bursts have a width uncertainty of 1-2ms, as I would expect. Perhaps this is a typo? Other numbers in these tables are curious: e.g. burst L1 has fluence 4(6) Jy-ms. Why is the fractional uncertainty >100%? Why is the localization of burst S1 more than twice as precise than any of the other S band bursts, despite burst S2 and S3 having comparable S/N? I see that it is much wider-band than the other two, but that doesn't really matter given that the image S/N is the final result of the width+fluence+bandwidth of the burst combined. Looking at the beam sizes and the S/N values quoted, it seems more like the statistical uncertainty is too large for the other bursts rather than too small for burst S1. This might indicate that the fitted source size was >> the synthesized beam size in those cases, which would be indicative of a concerning systematic error (as the FRB itself must be a point source, and hence should have a size approximately equal to the synthesized beam size).

Author Response

We thank the referee for pointing this out. We have edited the values in the table and added more description to this section to explain the burst fitting process and the caveats to this analysis.

For CASA imaging, we only used the frequencies that contain burst signals to form the image. We then use that image to determine the burst position. Therefore, the beam size for each image will change based on the signal peak frequency of the respective FRB. This can lead to different size of fitting errors even for similar S/N bursts.

My new response:

There are still a number of inconsistencies in these two tables. The most glaring is the case of burst C1, which remains unaddressed. I absolutely do not believe that it is possible to measure the width of this burst, observed with 20 sigma significance using 10 ms sampling, to a value of 0.1ms with an uncertainty of 70 microsec. Consider this: the DM uncertainty for that burst is listed as 1300+200-200 pc/cm³: over the reported bandwidth of 230 MHz centred on 5.27 GHz, DM 1500 would lead to 17ms of DM sweep vs 12 ms for DM 1100 – if that uncertainty in DM is right, there is **no way** to get a sub-ms width precision. Not that I think that it is possible with a 20 sigma burst and 10ms sampling in any case. There must be either an error in the fitting or an error in

the transcription. I'm doubtful on the widths of the other bursts too: applying the same approach to S4 yields an uncertainty in DM sweep time of 7-8ms, making the inferred width of $0.7 +0.6 -0.5$ look infeasible, although it is less glaring than C1.

– **Response:** We have modified the methodology and removed the burst fitting analysis. We now report the burst flux obtained using CASA imfit (using only the spectral windows with burst signal), upper limits on burst widths, and visually determined the burst bandwidths.

Apart from that, there is still no explanation for why S1 has an astrometric precision $>2x$ higher than S2. S2 has a higher central frequency and a comparable S/N. So there is no way that S1 should have a position which is 2-3x more precise than S2 – their uncertainties should be comparable. I'll ask again explicitly: why is the statistical precision of S2 so low? It is only about 1/10th of the synthesised beam, when it should be 3x better.

– **Response:** We have corrected the burst properties now. S2 is present around a frequency of 2.7GHz while S1 is present around a frequency of 3.3GHz. As only the spectral windows consisting of burst signals were used for imaging, the beam size generated in case of radio image for S2 would be larger than that for S1, leading to a poorer error estimate. Further, as we are modeling the image, any noise or artifacts present near the FRB might affect the final fit. Such artifacts are seen in the radio image of S2.

50. Reference 41: why is this the arxiv reference rather than the published ApJ article?

Author Response

Because the article hasn't been published yet and is still under review.

My new response:

Reference 41: "Law, C. J. et al. A Multi-telescope Campaign on FRB 121102: Implications for the FRB Population. ArXiv e-prints (2017)" was published in ApJ over 4 years ago.

– **Response:** We made a mistake with the reference number. We have corrected that to refer to: Aggarwal, K. et al. Comprehensive analysis of a dense sample of FRB 121102 bursts. arXiv e-prints arXiv:2107.05658 (2021). 2107.05658.

RESPONSE TO REFEREE #2:

GENERAL COMMENTS

The revised manuscript is definitely improved, and the authors have implemented or addressed most of my comments on the original submission. I read the revised version and provided detailed comments (see below) *before* reading the author's responses to my original report. I see that some of my comments are still similar to issues that I flagged originally. In these cases, I think that the paper is still unclear and requires further editing to address these points fully (in some cases, that may mean including information that was only provided in the referee response). This includes, for instance, being more clear on whether the *apparently large* DM variations are genuine or reflect an underestimation of the uncertainties (I suspect that it's the latter).

– **Response:** Discussion of DM variations has been shortened significantly in the manuscript because they are not relevant to the main results of the paper, and currently seem to be no more than those expected from changes in burst width and burst structure. The Extended Data Figure showing DM vs. time has been removed and a brief comment on the apparent DM variations evident in Supplementary Table 1 has been added both to the footnote of this table and in the Methods (see last response to this report below).

My main substantive criticisms of the paper's claims remain:

1) Since repeaters appear to be a small fraction (~5%) of observed FRBs, the following claim is too general in my opinion: "suggesting caution in inferring redshifts for FRBs without accurate host galaxy identifications". I think this statement should be clearly qualified with "at least for repeating sources".

– **Response:** The issue of inferring an FRB redshift from its DM is independent of whether the FRB repeats or not. It is a more conservative choice to recommend caution for all FRBs rather than just for the repeating sample.

2) Since there are also "active repeating FRBs" with low local DM and *no* persistent radio counterpart, the following claim "The dense, complex host galaxy environment and the associated persistent radio source may point to a distinctive origin or an earlier evolutionary stage for active repeating FRBs." is more accurately stated as: "Repeating FRBs have been found in a range of environments, some much less extreme compared to FRB 121102 and FRB 190520. Whether this diversity reflects different evolutionary stages of a single source type, or multiple source types, requires further investigation."

– **Response:** The last sentence of the summary paragraph has been modified to: "The dense FRB environment and the association with a compact persistent radio source may point to a distinctive origin or an earlier evolutionary stage for this FRB source." The relevant discussion in the main text has also been modified substantially (see lines 146-172.)

Title:

Line 1:

- Suggest not using an acronym in the title.
- From the title, it's now not clear what is novel compared to FRB 121102.

– **Response:** Acronym removed. The authors believe the title adequately reflects the main results of the paper (and we note in the main text that DM_host for FRB 190520 is ~3 times larger than the DM_host for FRB 121102).

Summary:

Line 42: "the most energetic radio transients in the Universe,": can you really say that they're more energetic than gamma-ray burst afterglows? Maybe higher peak luminosity?

– **Response:** This sentence has been revised and no longer uses this terminology.

Line 45: "as FRBs' natal environments"  "as the natal environments of FRBs"

– **Response:** Done

Line 48: "and identified with"  "and associated with"

– **Response:** Done

Line 49: "a redshift $z = 0.241$ ": quote uncertainties.

– **Response:** Done

Line 53: "source after FRB 121102 with"  "source, after FRB 121102, with"

– **Response:** N/A (this sentence has been significantly revised)

Line 54: "a compact PRS.": I don't think it's worth defining this acronym in the summary. In fact, you don't even use it in the following sentence.

– **Response:** The acronym is no longer used here.

Line 54: "The dense, complex host galaxy environment": is it really the "host galaxy" environment that is dense? It must surely be local to the FRB source. Also, what does "complex" mean in this context?

– **Response:** This statement was simplified to "The dense FRB environment..."

Line 54: "The dense, complex host galaxy environment and the associated persistent radio source may point to a distinctive origin or an earlier evolutionary stage for active repeating FRBs.": why? FRB 20201124A is also "active", and closer, but doesn't have a persistent radio source counterpart or a very high local DM contribution.

– **Response:** We removed 'active' from the statement and restrict the statement to the quantifiable characteristics

"The dense FRB environment and the association with a compact persistent radio source may point to a distinctive origin or an earlier evolutionary stage for this FRB source."

Main:

Line 57: "FRB 190520B (corresponding TNS name FRB 20190520B, same hereafter)": why not just use a single name for the source from the start? Note that, even after saying you'll use "FRB 20190520B", you then use "FRB 190520B" again. Has the name "FRB 190520B" ever appeared before? Seems like you should either use "FRB 190520" or "FRB 20190520B".

– **Response:** We have clarified the choice of abbreviation on line 55, and use FRB 190520B because there is already a 20190520A. To avoid confusion, '190520B' seems the best we can do.

Line 60: "Four bursts were detected during the initial scan, suggesting a repeater.": I would say that multiple bursts "demonstrates", rather than "suggests", a repeating source.

– **Response:** No longer applicable, this sentence has been edited for brevity.

Line 63: "hr": unit shouldn't be italicised.

– **Response:** Done.

Line 64: "Similar to other repeaters, this FRB shows...": reference needed. Hessels et al. 2019 would be a good one for this, especially since it's already in your list (Ref. 25), but I'm of course biased.

– **Response:** Done.

Line 70: "We measured a burst source position of": is the quote position in the ICRF? Be explicit about this.

– **Response:** Yes, The description has been revised in Line 69.

Line 84: "shows the most likely stellar component": I don't understand what "most likely stellar component" means.

– **Response:** The text has been revised from "most likely stellar component" to "most likely stellar continuum emission".

Line 89: " $z = 0.241$ based": quote uncertainties.

– **Response:** Done.

Line 112: "the extremely low chance coincidence probability": I would drop the "extremely". I don't think that ~1% is "extremely" low.

– **Response:** Done.

Line 132: "consistent with the low chance coincidence of FRB 190520B with J160204.31-11718.5.": this is redundant with the beginning of the paragraph, so could be dropped.

– **Response:** Done.

Line 139: "may add considerable variance to estimates for the IGM contribution": if a significant fraction of FRB sources are like FRB 190520, so it should be noted here that repeaters are a small fraction of the known FRBs (~5%, cite CHIME/FRB first catalogue).

– **Response:** If large DM_host is indeed related to repeating FRBs, then we agree with this statement. However, the possible linkage between DM_host and burst repetition is independent from the impact that misestimated host DMs have on estimates of DM_IGM. There is still ambiguity in where the large DM_host for FRB 190520 arises in the host galaxy (distributed or localized near the source). We have endeavored to make this ambiguity explicit in the main text discussion (see line 157).

Line 140: "It is also more likely that FRBs with large DMhost will not be detected by search systems sensitive to a limited DM range.": true, but the DM of FRB 190520 is well within the range being searched by most experiments, and it is far from being the highest FRB DM to be observed to date.

– **Response:** This text has been modified (see lines 128-133). The referee makes an accurate point, and we simply note that "FRB searches need to accommodate large values of DM_host as part of the DM budget for FRB sources."

Line 141: "A large sample of FRBs"  "A large sample of precisely localised FRBs with measured host galaxy redshifts" (would also be better if you quantified "large"; I'd say that hundreds should be enough to get a robust idea of what is "typical")

– **Response:** We revise it as: "... a larger sample of precisely localised FRBs with measured host galaxy redshifts..." and the full, revised statement is now on lines 128-130.

Line 142: "allow the host galaxies, their circumgalactic media, and near-source environments to be probed statistically along with the IGM.": I would cite the recent Mannings et al. paper here.

– **Response:** Done. And now the corresponding line is Line 130.

Line 145: "the first repeating"  "the first known repeating"

– **Response:** Done.

Line 146: "and the first to be identified with a compact, luminous PRS": it would be useful to have a table that compares the properties of both sources and their hosts, including: i. burst rates, ii. Burst energies, iii. local DM, iv. local RM, v. host galaxy stellar mass, vi. local SFR, vii. persistent source luminosity and spectrum, etc.

– **Response:** While we agree that such a table would be useful for a detailed comparison of these two sources, the authors believe this table is not strictly necessary given the limited room for tables and figures in the paper. Salient points of comparison between FRB 121102 and FRB 190520 are summarized on lines 147-157, and a more detailed comparison is beyond the scope of this paper.

Line 153: "larger than highest"  "larger than the highest"

– **Response:** Done.

Line 155: "unresolved structure in VLA observations": it's important to quote the maximum possible size of this source (and compare with the case of FRB 121102).

– **Response:** We have added the fitted sizes of the source across the VLA bands and given a maximum possible size obtained at C-band at its angular diameter distance in both Methods and the main text (line 73). The maximum size derived from the VLA observations is 0.36". We also have VLBA observations that limit the maximum possible size much better and will be discussed in the context of FRB 121102 in a follow-up paper.

Line 157: "a true PRS": the switching between "radio continuum counterpart" and "PRS" / "persistent radio source" terminology is confusing. "true" is even more confusing: the source is a persistent radio source regardless of whether it's associated with FRB 190520. The discussion and terminology should just use a single

term and focus on whether the emission is i. spatially co-located with FRB 190520 and ii. sufficiently compact that it is unlikely to be due to star formation.

– **Response:** The nomenclature in common use is confusing. The phrase “persistent radio source” is generic, but has been applied to a specific kind of object seen previously towards only one FRB. Since the source origin is unknown, we need some specific name for the phenomenon. However, the argument is structured in a way that requires we refer to the radio counterpart before we know it is a “PRS”. Therefore, we use “radio continuum counterpart” generically (before we conclude it is an FRB-related phenomenon), but “PRS” specifically. The addition of “true” seems to confuse the issue further, so we have reworded the sentence to “... a PRS like that associated with FRB 121102”.

Line 157: "The whole properties"  "The properties"

– **Response:** Done.

Line 158: "could be found"  "can be found"

– **Response:** Done.

Line 158: "in Table .": missing table number.

– **Response:** Done.

Line 161: "While the observed burst repetition and morphology can be temporal due to various mechanisms": this sentence is confusing. I think you mean that the burst and source properties likely vary with time.

– **Response:** The statement is changed to “While the observed burst repetition and morphology can be time-dependent due to various mechanisms...” Note that this discussion has been moved to the last paragraph of the main text.

Line 162: "PRS emission and DMhost reflect more persistent aspects of the FRB environment and thus may be more reliable tracers of any putative subclasses.": this seems to contradict your earlier argument that active repeaters might be found in such environments.

– **Response:** This sentence has been modified as part of the discussion that appears between lines 158-171, and in its revised form does not appear to be self-contradictory to the authors.

Line 167: "implying that such features could require dense, magnetized plasma within parsecs of the source of the bursts": importantly, FRB 121102 is *known* to be in dense, highly magnetised local environment, as shown by its large and highly variable RM (Michilli et al. 2018). That this is necessary for the activity to be high is not well supported, however, by the contradicting case of FRB 20201124A (and FRB 20180916B).

– **Response:** This discussion has been significantly modified for clarity, please see the discussion beginning on line 147.

Line 173: "Some active repeaters do have comparably strict limits": comparable to what? Also, the limits are *much* deeper (in the case of FRB 20180916B and FRB 20200120E) than the observed luminosities of the radio sources coincident with FRB 121102 and FRB 190520).

– **Response:** We have reworded this sentence to note that some repeating FRBs have much deeper PRS limits.

Line 174: "which suggests complexities in the connection between burst activity": I would say that it argues that the current study doesn't convincingly argue for *any* connection between burst activity and the presence of a persistent radio counterpart.

– **Response:** Indeed there are strict limits on PRSs for other repeating FRBs, but the discussion here is focused on the coincidence of activity, PRS, and extremely large DM. The excess DM is a key signature of FRB 190520

and we present some potential explanations for how these properties might appear together in some FRBs and not in others. Parts of this discussion are speculative, but we felt it was valuable to present it, as it highlights a major open question in the study of FRBs: whether there is one source model or more.

Line 180: "and other properties.": what other properties? This is vague.

– **Response:** N/A, the discussion where this originally appeared has been revised significantly.

Line 183: "Active repeaters with PRS may either be a distinct population or FRB sources at earlier evolutionary stages.": these are not the only options. They may simply be similar sources, of comparable age, that happen to be in a denser local environment.

– **Response:** This discussion has been modified for clarity (see lines 165-171).

Figure 1:

- The bursts would be easier to visualise, in some cases, if the time-frequency integration factors were larger.

– **Response:** The time-frequency resolution used in this figure is the same for each burst, and was chosen to optimize the visibility of the bursts' substructure.

- "These six bursts are chosen from each observation epoch by its characteristic dynamic spectra.": it's unclear what "by its characteristic dynamic spectra" means.

– **Response:** The word "characteristic" has been removed, and the sentence now states: "These six bursts are chosen from each observation epoch."

- "The 'P0' means the first burst of all the 79 bursts."  "The burst labels 'PXX' are in order of arrival time." (then also point to the table of burst properties)

– **Response:** Done.

- "The color are linear scaling"  "The color map is linearly scaled"

– **Response:** Done.

- "The bad frequency channels are set to zero and labeled in red patches on the left.": would be better to plot no data here at all, since the values aren't 0, there is no data.

– **Response:** The bad channels have been masked and the figure caption revised accordingly.

- y-axis label on the bottom panel: "Detection"  "Number of bursts"

– **Response:** This panel was removed after the authors re-assessed the number and layout of figures, per the editor's request.

Figure 2:

- "the best FRB position at": indicate that these correspond to RA and Dec, and at epoch J2000.

– **Response:** Done.

- "The infrared Jband image by Subaru/MOIRCS shows emission only at the location of the peak of the optical light profile of the host galaxy.": that's not so easy to see, given how zoomed-out these figures are. A zoom-in on the host galaxy would be useful.

– **Response:** We have added insets with zoomed in images of the host galaxy at the optical and infrared bands.

- What is the optical/IR source that is right next to the host galaxy? Is that known to be a foreground star?

– **Response:** It is a spectroscopic confirmed M-type star, as described in the third paragraph of Section 4 in the Methods.

- Indicate whether these images are oriented with East to the left, or not. It would be better to include coordinate axes and labels (RA and Dec).

– **Response:** We have added that North is up and East is to the left, as usual (and indicated by the crosshairs). Given the complexity of the panels and the limited space, we prefer not to include numerical coordinate axes.

- Indicate what the artefacts in the optical/IR images come from.

– **Response:** We have clarified that the streak is an artifact due to a bright star outside the image field of view.

Figure 3:

- "FRB 190520" is used on the plot, whereas "FRB 190520B" is used in the caption.

– **Response:** The plot label has now been changed to FRB190520B for consistency.

Table 1:

- "Measured Parameters"  "Measured Parameters of Bursts"

– **Response:** We have changed this to "Burst Parameters," as some of the parameters listed are inferred or modeled rather than measured quantities.

- A table comparing properties with FRB 121102 and FRB 20201124A would also be useful.

– **Response:** As stated in our response to a previous comment above, the authors believe this table is not strictly necessary given the limited room for tables and figures in the paper, as the relevant points of comparison are already noted in the main text discussion.

- Provide the uncertainties for RA and Dec.

– **Response:** The uncertainties have been added.

- Number of detections = 81. But the first two paragraphs of the paper quote 75 bursts seen by FAST and 9 with VLA (= 84).

– **Response:** Sorry for the mistake, the FAST detection is 75(tracking mode)+4(drift scan), and VLA has 9 detection. So the total number is 88. The table has been corrected accordingly.

- "Measured width (ms)": is this an average measured width for all bursts (at all observing frequencies)?

– **Response:** Yes, this is an average measured width for all bursts detected by FAST. A footnote has been added to the table and the text in the table has been modified for clarity.

- "Scattering timescale (ms) at 1.25GHz": also an average (or median)?

– **Response:** Yes, this is also an average value. The table has been edited to clarify this.

- "Scintillation bandwidth (MHz) at 1.4GHz": also an average (or median)?

– **Response:** Yes, this is also an average value. Note however that we have removed the scintillation bandwidth analysis from this paper, as it is not relevant to the main results and conclusions. A robust scintillation analysis is the topic of another paper currently under preparation and is beyond the scope of this paper.

- Also for the persistent radio source position, the uncertainties should be quoted.

– **Response:** Done.

- I would suggest also quoting an upper limit on the size of the persistent radio source.

– **Response:** Upper limit on the size of the PRS added.

- "redshift"  "Redshift" (since all the other rows are capitalised)

– **Response:** Done.

- MSun: indicate that this is the *stellar* mass of the galaxy.

– **Response:** Done.

- Footnote b: "electron density mode"  "electron density model"

– **Response:** Done.

Author contributions:

Line 295: "the associated PRS": I'd avoid using this acronym in the Acknowledgements.

– **Response:** Done.

Line 296: "Optical/NIR"  "optical/NIR"

– **Response:** Done.

Line 298: "contributed to measured the burst scattering, modeling combined with analysis of propagation": this isn't a grammatically correct sentence. Needs rewording.

– **Response:** This language has been corrected.

Line 301: "RFI": I'd avoid using this acronym in the Acknowledgements.

– **Response:** Done.

Methods:

1 Observations:

Line 325: "structure(see"  "structure (see" (missing space)

– **Response:** Done.

Line 325: "see result plot from ref.34": which plot? I'm confused.

– **Response:** This sentence has been removed.

Line 326: "in that date"  "on that date"

– **Response:** Done.

Line 328: "and only kept Stokes I"  "and only Stokes I was recorded"

.

– **Response:** Done.

Line 329: "the trial DM range is from": start new sentence.

– **Response:** Done.

Line 329: "we matched the pulse width by a boxcar search.": what range of widths was searched?

– **Response:** The range of width search is from 0.049 to 200 ms; this has been noted in the Methods.

Line 337: "The sub pulse is recognized if the profile peak does not fall behind the noise baseline.": this needs to be rewritten. It's awkwardly worded and unclear.

– **Response:** The revised sentence reads: "A sub pulse is recognized if the bridge between the two closely-spaced-in-time peaks drops more than 5 sigma below the higher peak."

Line 339: "The bandwidth of each burst is roughly estimated by its spectrum": it's obvious that the bandwidth of the burst is estimated from the spectrum, but *how*?

– **Response:** The revised sentence reads: "We roughly estimate the bandwidth of each burst by dividing the whole bandpass into 50 MHz subbands and identifying the subbands containing burst emission."

Line 345: "DDT project: 20A-557"  "DDT project 20A-557"

– **Response:** Done.

Line 373: "formed by realtime system"  "formed by the realtime system"

– **Response:** Done.

Line 377: "when the data is de-dispersed at a DM closer to the DM of the candidate": maybe better to say "closer to the true DM"

– **Response:** Done.

2 Localization of bursts:

Line 392: "data with Common"  "data using the Common"

– **Response:** Done.

Line 394: "each observation with the burst"  "each observation with a detected burst"

– **Response:** Done.

Line 395: "data using CASA task"  "data using the CASA task"

– **Response:** Done.

Line 395: "Observation of 3C 286 (before the FRB observation) was used"  "Observations of 3C 286 (before the FRB observations) were used"

– **Response:** Done.

Line 400: "Therefore, the burst positions on short timescales will have systematic errors of the same magnitude as the deep imaging.": I think that this is probably true, but I would point out that the uv-coverage is much worse for individual bursts than it is for the integrated image (see e.g. Nimmo et al. FRB 20201124A paper, where the situation is even worse).

– **Response:** Because the snapshot coverage of the VLA is good (and enhanced by the fact that both the FRB and PRS can incorporate multi-frequency synthesis), the ultimate effect on the difference in size of the synthesized beam in the integrated vs. burst image is negligible; this is furthermore supported by our short-timescale imaging described in Section~2.

Line 404: "For each burst, we search different spectral window ranges to generate the image with highest S/N": quantify how many trials are being done here.

– **Response:** We have reworded this statement to: "Most of the bursts are frequency modulated, so for each burst we select the spectral window range that produces the highest image S/N."

Line 425: "the best estimate for burst"  "the best estimate for the burst"

– **Response:** Done.

Line 437: "Due to this any temporal structure in the bursts (multiple components, scattering, sub-burst drift) would be resolved out and we would not be able to model it.": don't you mean the opposite? The burst structure is *unresolved*, as opposed to being "resolved out".

– **Response:** We have removed the fitting analysis and therefore this statement has also been removed from the text. The Methods have been clarified on lines 390-396.

Line 445: "Burst times are defined in barycentric MJD time": do you mean TCB or TDB? Be clear.

– **Response:** Fixed

3 Persistent Radio Source:

Line 496: "This can be accounted for if the source is more extended than the PRS for FRB 121102.": would be useful to quantify what lower-limit on physical scale this implies (>~ 1pc, right?).

– **Response:** It seems premature to discuss variability or lack thereof in the PRS spectrum given the sparse amount of data. At this stage, we cannot rule out any variability very strongly, so we have decided to not offer any interpretation on this aspect of the PRS spectrum (see revised Methods lines 442-445).

Line 505: "of a few tens mJy"  "of a few tens of mJy"

– **Response:** Done.

4 Galaxy Photometry and Redshift Determination:

Line 530: "was obtained with Double Spectrograph (DBSP)"  "was obtained with the Double Spectrograph (DBSP)"

– **Response:** Done.

Line 535: "The PRS location later is found to"  "The PRS location was later found to"

– **Response:** Done.

Line 538: "as reported in PanSTARRS DR1 catalog"  "as reported in the PanSTARRS DR1 catalog"

– **Response:** Done.

Line 540: "allowing both the coordinate of VLA persistent radio source and the M-star to fall into the slit"  "ensuring that both the VLA persistent radio source and the M-star fell within the slit"

– **Response:** Done.

Line 541: "under photometric sky condition"  "under photometric sky conditions"

– **Response:** Done.

Line 553: "indicating the extended"  "indicating that the extended"

– **Response:** Done.

Line 556: "coincidence probability of the galaxy (J160204.31-111718.5) to that of the burst"  "coincidence probability of the galaxy (J160204.31-111718.5) and burst source"

– **Response:** Done.

Line 559: "the half light radius of galaxy"  "the half light radius of the galaxy"

– **Response:** Done.

Line 560: "the localization error region": you mean the *burst* localisation region? Be clear.

– **Response:** Done.

Line 561: "the localization error on the FRB location"  "the localization error on the FRB position"

– **Response:** Done.

Line 562: "the size of the host"  "and the size of the host"

– **Response:** Done.

5 FAST Burst Sample Analysis:

Line 575: "We find the burst rate of FRB 190520B is": it's important to associate this with a fluence threshold, since the observed activity rate depends on the sensitivity of the telescope and brightness of the bursts.

– **Response:** We added the threshold of 9 mJy ms to the end of this sentence.

Line 582: "A high value means that the burst is concentrated in a small phase window, which indicates a possible periodicity pattern.": I would caution that this assumes that there is a single phase window. Many pulsars/magnetars/giant pulse sources show multiple phase windows of emission.

– **Response:** The revised sentence reads: "A high value means that the burst may concentrate in a small phase window, which indicates a possible periodicity pattern."

Line 585: "session, we then folded"  "session. We then folded"

– **Response:** Done.

Line 592: "which indicates the observation selection effects rather than true periodicity patterns"  "which indicates that this is dominated by observational selection effects as opposed to a true periodic behaviour related to the source"

– **Response:** Done.

Line 598: "the off-pulsed noise level"  "the off-pulse noise level"

– **Response:** Done.

Line 599: "zenith angle depended telescope gain"  "zenith angle dependent telescope gain"

– **Response:** Done.

Line 599: "The Kelvin unit was then converted to Jy"  "We then converted from units of Kelvin to Jy"

– **Response:** Done.

Line 604: "Assuming it is of flat broadband spectrum, We then" → "Assuming a flat, broadband spectrum, we then"

– **Response:** Done.

Line 608: "for each pulses at"  "for each pulse at"

– **Response:** Done.

Line 614: "The total of 79 bursts detected in 16 months show a faint trend in DM.": unclear what "faint trend" means.

– **Response:** This text has been removed.

Line 615: "The DMs of each day use the best-fit DM value of that day.": how was this determined.

– **Response:** The text has been clarified in Methods->Observations->FAST. We fit for DM using the DM_phase.py package. Since the weaker bursts could not be fitted well at higher time resolution, we used the best-fit DM of the highest S/N burst on a given day to de-disperse all other bursts detected on that same day.

Line 616: "More detection are needed for a detailed analysis.": for a detailed analysis of what?

– **Response:** This sentence has been removed.

Line 635: "gravitation constant G"  "gravitational constant G"

– **Response:** Done.

Line 642: "the uncertainty is due to the measurement": not sure what is meant by this. I don't think it adds anything to the sentence.

– **Response:** The sentence has been edited for clarity.

Line 693: "and showed some of"  "and show some of"

– **Response:** Done.

.

Line 693: "Averaged scintillation bandwidth over"  "By averaging scintillation bandwidth over"

– **Response:** N/A (scintillation bandwidth analysis was removed).

Line 723: "The geometric factor G is": I would avoid using "G", since that was previously used for the gravitational constant.

– **Response:** We have modified the notation so that G is referred to as G_scatt. The first corrected instance of G_scatt is highlighted in blue and all subsequent instances of the parameter have been modified.

Line 741: "Rotation measure (RM) is searched at L-band with FAST data.": why not also in the VLA data, which provide higher frequencies and large bandwidth?

– **Response:** This comment is no longer applicable because the polarization analysis has been removed. The authors determined it was beyond the scope of this paper (see general response to editor above). Full Stokes information was not available for the VLA data, so the RM could not be estimated for the VLA bursts regardless. Further analyses of the polarization are ongoing and make use of newer data that will make the picture more definitive.

Line 746: "The observed polarization could be due to an intrinsic low linear polarization.": or propagation effects in the local environment that go beyond the "normal" Faraday rotation effect.

– **Response:** N/A, this section was removed.

Line 755: "we place an lower limit"  "we place a lower limit"

– **Response:** N/A, this sentence was removed.

Line 756: "Such a large RM is even larger than that of FRB 121102, which suggests that FRB 190520B also resides in an extreme magneto-ionic environment": maybe, but it also bears mentioning that FRB 121102 is also depolarized at L-band, even when taking intra-channel rotation into account. It also bears mentioning that FRB 121102 has a highly variable RM, so be clear on whether *all* the FAST bursts were searched for an RM.

– **Response:** N/A, this section was removed.

Extended Data:

ED Fig. 1:

- Given that the 1.5 GHz and 3 GHz bursts are systematically offset from each other, I'm not sure that this earlier averaging of the positions between frequencies (Line 423: "Finally, we take a weighted average of the three burst positions at the three frequency bands, weighing each position by the inverse of the total error obtained at the respective frequency band.") makes sense. How can the overall uncertainty on the position be 0.1 arcsecond when the offset between the groups of bursts at 1.5 GHz and 3 GHz seen in this figure is ~ 0.2 arcsecond?

– **Response:** We have updated this figure. It shows that 1.5GHz and 3GHz burst localization regions overlap for all except one S band burst (the errors in this updated figure have changed as we are now estimating the systematic errors using a larger radius for cross-match). Therefore, we have assumed that the positions are independently distributed, without any real offset. So, we were able to use the weighted averaging method to estimate the final burst position and errors.

- Why were the visibilities of multiple bursts on the same day (and/or between epochs) not combined (cf. Nimmo et al. FRB 20201124A paper)? This would give a "coherent" summation of the burst data, as opposed to "incoherently" averaging the positions of multiple bursts.

– **Response:** The procedure we have used in this work allows us to localize individual bursts and include detailed analysis of systematic errors. We have demonstrated in the Methods section that the systematic errors are dominated by the minute-scale calibration errors that vary between bursts. We believe that this approach also leads to a robust estimate of the burst position and the corresponding errors.

ED Fig. 2:

- Would be better to label the x-axis "Frequency (GHz)" rather than " ν (GHz)".

– **Response:** Done.

ED Fig. 3:

- "Two emission lines are"  "These two emission lines are"

– **Response:** Done.

ED Fig. 4:

- Indicate what threshold fluence these rates correspond to.

– **Response:** Done.

ED Fig. 5:

- "P1 stands for pdot.": why define a variable that just means the same thing as another already defined variable?

- I'm not sure that this figure is very useful. The number of period trials in the left panel must be very high, and I'm not sure if any promising candidates would be visible within the forest of other lines.

– **Response:** This figure has been removed. A revised description of the burst periodicity search can be found in Section 5 of the Methods ("Short and long timescale periodicity search" on line 530).

ED Fig. 8:

- Given the large scatter in apparent DMs for epochs with MJD > 58900, it is not well motivated to fit a linear trend back to a single data point at MJD ~ 58620. Also, the large scatter in the DM values, compared to the uncertainties on each "best-fit value per epoch" is huge. Either the uncertainties are grossly underestimated, or there are huge DM variations on short timescales. This needs to be reconciled or interpreted astrophysically.

– **Response:** This figure has been removed.

ED Fig. 9:

- "for burst P5, P46, P42, P51 that for examples"  "for bursts P5, P46, P42, P51, as examples"

- "the blue line is the best fit result": using what kind of fitting function?

- Has the zero lag been removed? Indicate this.

– **Response:** We have decided to remove ED Fig. 9 and the scintillation analysis from this paper as the results are irrelevant to the main conclusions of the paper.

ED Fig. 10:

- "All 1D burst profiles have been normalized to a noise threshold of one.": I'm not sure what's meant by this, and the y-axis in the burst profile sub-panels is confusing. Why not plot S/N or flux density?

– **Response:** We have modified the y-axis to show the burst profiles in units of S/N.

- "frequency drift,"  "time-frequency drift,"

– **Response:** Done.

ED Tab. 3:

- Don't need to quote 4 digits for S/N.

– **Response:** Done.

ED Tab. 4:

- "MJD is referenced to the solar system barycenter": TCB or TDB?

– **Response:** TDB, the caption has been updated accordingly.

- "As mentioned in text, width values here should be considered as upper limits.": some of these widths are $\sim < 1$ ms, even though the sampling time is 10ms. I don't see how this can be a robust measurement. It's just a reflection of the fact that you're also fitting for scattering and that leads to an apparently very narrow "intrinsic width", right? I think you're over-fitting the data with a too complicated model and that the quoted widths are misleading. But in the description of the fitting (Line 429) there's no mention of fitting for scattering... so how can the fitting widths be much lower than the time resolution of the data?!

- You need to caution that the uncertainties on the DM values are underestimated; or, the paper should interpret these variations, if you consider them to be genuine.

– **Response:** We have removed the burst fitting analysis and have modified the burst properties in the table. Now we are using CASA imfit to obtain burst flux using only the spectral windows with the burst signal, and are visually determining the upper limits on burst width as the VLA data has poor time resolution. We have also determined the burst bandwidths visually and added that to the table. We report the S/N maximizing DM values in ED Table 4, as the data is not of sufficient resolution to obtain the structure maximizing DM.

Supplementary:

- "The P1 to P4 were detected"  "P1 to P4 were detected"

– **Response:** Done.

- "Arrival time of burst at the solar system barycenter.": TCB or TDB?

– **Response:** It's TDB. We add it in the note.

- "All the bursts detected on the same day are assigned the best fit DM value of the burst from that day.": of which burst?

– **Response:** The selected bursts are highlighted in bold now.

- "We only report scattering timescales for bursts with an obvious scattering tail.": how does one judge if there is "obviously" a scattering tail, given that the scattering time is for the most part, comparable to or less than the pulse width (and given time-frequency drifts, and uncertain DM).

– **Response:** The criteria used to report scattering times has been clarified both here in the Table caption and in the Methods section on scattering.

- The DMs appear to vary quite drastically between epochs. Either this should be interpreted from an astrophysical point of view, or the uncertainties should capture systematic effects.

– **Response:** At this stage, the apparent DM variations between epochs appear to be entirely consistent with variations in pulse shape from a combination of variable effects, including intrinsic pulse structure, scintillation drift, flux distribution in frequency, and scattering (which affects bursts chromatically). A brief statement to this effect has been added to the Supplementary Table footnote, and to the Methods (see line 284).

Reviewer Reports on the Second Revision:

Referees' comments:

Referee #1 (Remarks to the Author):

The authors have made detailed responses to all of the points raised in the second review. I think the paper is much improved as a result, especially in the Methods section. I have a couple of minor remaining points that the authors may wish to consider, which follow below.

1. Regarding nomenclature: I agree with the second reviewer that there is no reason to use two names, and I would encourage the use of the TNS name 20190520B rather than 190520B (I don't understand the objection raised that there is already a source named 20190520A - that is the reason for the alphabetical suffix?)

2. Regarding the fluences of the VLA bursts: it remains surprising to me that every single VLA detection, except S5, has a higher fluence than the *brightest* FAST burst. Imagine placing the VLA bursts onto ED Figure 6: a number of the VLA bursts would have energy $>10^{39}$ erg and be off the chart, highly inconsistent with the fitted lognormal function. In 11.4 hours of VLA time, 8 bursts were seen that exceeded the maximum fluence seen by FAST in 18 hours. Obviously, the source activity level may be changing as a function of time, but both the VLA and FAST observations spanned a wide range of dates. Accordingly, given that the authors have clarified that the burst properties were measured in an identical fashion, it would seem worthy of comment somewhere in the paper that a) the localisation was seemingly rather fortuitous, given that none of the FAST bursts seen in a period of 17 hours exceed the 0.4 Jy ms 8 sigma realfast detection limit that is quoted for the VLA, and yet the source was seen numerous times in 11.4 hours of VLA time, and b) this would seem to support variability in the activity levels (unsurprising, but worth noting).

3. Related to the above point, on line 323: the 8 sigma fluence limit of a 10 ms VLA image is quoted as 0.4 Jy ms - but this must depend on the observing band? Is this an L band number? Burst S4 and S5 fall below this limit, considerably so in the case of burst S5.

4. Not that it appreciably impacts the results, but I think there is a minor error in the calculation of the rms of DM_MW. I get 21 units, not 17 units, for the rms, given the two uniform ranges quoted for galactic and halo contributions.

5. Line 131: "Large DM_host may also imply..." -> "Large DM_host ****contributions**** may also imply..."

6. Everywhere the scattering time is quoted: the uncertainty should reach the same decimal place as the value. So it should be 10 +/- 2 ms, , or 9.8 +/- 2.0 ms (the latter is used in one place that I saw, but mostly it is given as 9.8 +/- 2).

7. Line 504 and line 607: why is the Halpha luminosity presented at line 504 (without having first presented the Halpha flux, which is not shown anywhere), and then the flux suddenly pops up at line

607? It would be much more logical to introduce the observable (flux) first at line 504, since the luminosity is derived from it.

Referee #2 (Remarks to the Author):

General comments:

The revised manuscript is significantly improved and addresses the vast majority of my previous comments.

I recommend the paper for publication, but would again like to ask the authors to sharpen a few things.

The most novel observational result is that this repeating FRB has a very large DM_{host} contribution, which means that inferring redshift via DM will in some cases be highly inaccurate. Considering the current sample of FRBs with host galaxies, it still appears that such DM_{host} outliers are rare (~< 10%), and I'd still encourage the authors to give the caveat that one-off and repeating FRBs could very well have distinct DM_{host} distributions (if so, this would most likely then be due to their local environment as opposed to the ISM of the host galaxy).

The authors say "The issue of inferring an FRB redshift from its DM is independent of whether the FRB repeats or not. It is a more conservative choice to recommend caution for all FRBs rather than just for the repeating sample." I don't fully agree with this. If one-off and repeating FRBs have different DM_{host} distributions, it is completely possible that the DMs of one-off FRBs are good proxies for redshift while those of repeaters are less reliable. That is still consistent with the data presented in Figure 3 of the paper.

Since the paper has been substantially reshaped, there is no longer any discussion of the Faraday rotation measure of this source. Given that the authors talk about the "dense environment" of FRB 190520B, and highlight its similarities with the original repeater FRB 121102, it is strange to remove the previous description of the polarimetric analysis.

Sincerely,

Jason Hessels
University of Amsterdam & ASTRON

Below are some remaining detailed comments, mostly minor.

Summary:

- Line 52: "The dense FRB environment and the association with a compact persistent radio source may point to a distinctive origin or an earlier evolutionary stage for this FRB source.": compared to what? The majority of other FRBs? This source doesn't seem distinct in origin or necessarily younger

compared to the first-known repeater, FRB 121102.

Main:

- Line 57: "Four bursts were detected during the initial scan."  "Four bursts were detected during the initial XXX-s scan." (indicate how long this scan was)
- Line 60: " $R(>7 \sigma) = \dots$ ": so is the 7 sigma equivalent to "a fluence lower limit of 9 mJy ms and a burst width of 1 ms": I'm confused about how the two are related and this should be more clear.
- Line 68: "position in International"  "position in the International"
- Line 69: "with position uncertainty"  "with positional uncertainty"
- Line 70: "dominated by systematic effects.": point to Methods here.
- Line 73: "has a flux density of 202 +/- 8 uJy at 3.0 GHz": also indicate at what epoch (or "averaged over span of epochs XXX - YYY").
- Line 75: "spectrum can be fit with a power-law index of": although it is fairly standard in astronomy, you should probably still define "spectral index" clearly, given the broader audience.
- Line 76: "was obtained by"  "was obtained using the"
- Line 83: "even before considering the radio counterpart association": I agree that the radio counterpart association with the FRB is clear, but it's less obvious to me how this strengthens the connection to the claimed host galaxy. I would leave this out.
- Line 84: "1.153-1.354"  "1.153-\$-1.354"
- Line 84: "shows the most likely stellar continuum emission": do you mean "most likely shows the stellar continuum emission of the host galaxy"?
- Line 92: "indicates the"  "indicates that the"
- Line 104: "a flat halo distribution from": would be more clear to say "uniform"?
- Line 126: "demonstrates that the distribution of DM_host values has a long tail"  "demonstrates that the distribution of DM_host values for the FRB population has a long tail"
- Line 126: In this paragraph it should be noted that the DM_host distribution might be different for repeating and non-repeating FRBs. In fact, despite the currently small sample of FRBs with hosts, this might already be demonstrable.
- Line 133: It would be good to quote the size constraint on the PRS, which I think is currently only

mentioned in the Methods, and to compare this to the constraint on FRB 121102's PRS (from Marcote et al. 2017).

- Line 148: "rotation measure variations of over 15% per year": it bears mentioning that FRB 121102 has an extremely large RM, and that RM variations are seen on both short (burst to burst, day to day, week to week) and long (year) timescales (see also Hilmarsson et al. 2021). There should also be some mention of what we know about the RM of FRB 190520B, since that is highly relevant to understanding whether it is also in an extreme magneto-ionic local environment.

- Line 157: "This hypothesis may however be modified by the ambiguity in the distribution of ionized gas responsible for the large DM_host from the host galaxy of FRB 190520B.": this sentence is hard to interpret. Do you mean that the DM_host isn't necessarily (all) local to the source and hence doesn't directly relate to the age/direct environment of the FRB? This could be worded much more clearly. If the DM_host were largely local to the FRB, one might also have expected to see a systematic DM evolution, right? (e.g. Piro paper)

Figure 1:

- "labeled in red patches"  "labeled using red patches"

- Indicate what DM (or multiple DMs?) were used to dedisperse the bursts as shown here.

Figure 2:

- The 0.1 arcsec radius localisation circle is so small that it might be better to show a 3-sigma circle instead. Or maybe I just need glasses.

Table 1:

- "Isotropic energy"  "Isotropic equivalent energy"

- "Flux at 3.0 GHz"  "Flux density at 3.0 GHz"

- For the PRS, add a table note to indicate the epoch of the flux density measurement (or range of epochs that are averaged over).

References:

- Line 188: "NAN, R.": is this name meant to be all caps?

Methods:

- Line 288: "sad trombone drift"  "time-frequency drift"

- Line 293: "by a boxcar search"  "using a boxcar search"

- Line 295: "follow-up burst search was taken with a narrower"  "follow-up burst search was done with a narrower"
- Line 296: "100-2000"  "100\$-\$2000"
- Line 302: "(see Section 5)": I'm not sure you can use section numbering in the Methods.
- Line 321: "for transient search"  "to search for transients"
- Line 362: "Most of the bursts are frequency modulated": I don't think you really mean "frequency modulated" (which has a specific meaning in engineering, e.g.), but rather "spectrally confined".
- Line 373: "R.A. error, Decl. error)": use the same abbreviations as earlier in the paper (do this throughout).
- Line 381: "of FRB position"  "of the FRB position"
- Line 416: "as 1.4" x 0.89", and 0.51" x 0.14", and 0.36" x 0.1""  "as 1.4" x 0.89", 0.51" x 0.14", and 0.36" x 0.1""
- Line 442: "The error on this position is estimated to be 0.10" and 0.05".": this should be written in a way to make it clear that you mean that these represent the uncertainties on the RA and Dec coordinates, respectively.
- Line 449: "yielding an average spectral index for the PRS of": define spectral index.
- Line 516: "a flat SED in": define "SED" (or just use "spectral energy density", since this is the only occurrence). Also define "nu" and "L_nu".
- Line 528: "obtained with Markov Chain"  "obtained with a Markov Chain"
- Line 553: "Assuming bursts have flat radio spectra": avoid jargon. Would be more clear to say "spectral index of about 0" (you will have also defined spectral index earlier in the text)
- Line 558: "the luminosity distance": here I'd add something like "using standard cosmological parameters" and give a reference.
- Line 569: "as the mean of a flat distribution": more clear to say "uniform"? (here and in the next sentence)
- Line 647: "such as the time-frequency drift of intensity islands (often referred to as the "sad trombone" effect).": give reference.
- Line 757: "for the technical support."  "for their technical support."

- Line 772: "helped on"  "helped with"

- Line 776: "on the present results"  "on the presented results"

Extended Data:

ED Table 2:

- "offsets in RA. and Dec."  "offsets in RA and Dec." (remove extraneous period)

ED Table 3:

- For row "S5", quote RA 16h02m04.250s to get the same alignment

ED Table 4:

- "Flux" should be "Flux density"

- For row "S3", quote Frequency 2.50-2.76 to get the same alignment

- "(TDB)"  "(in the TDB scale)"

- "DM that maximizes"  "The DM that maximizes"

- Add a single-sentence note that the apparent DMs may reflect time-frequency structure, as opposed to bona fide DM variations.

Supplementary Information:

Table 1:

- "of barycentric dynamical time"  "in barycentric dynamical time (TDB)"

- "the international celestial reference System (ICRS)"  "the International Celestial Reference System (ICRS)"

- "Epoch-to-epoch has been separated by single line"  "Epochs are separated by single horizontal lines"

- "and its DM variations may solely"  "and the apparent DM variations may solely"

Author Rebuttals to Second Revision:

Referees' comments:

Referee #1 (Remarks to the Author):

The authors have made detailed responses to all of the points raised in the second review. I think the paper is much improved as a result, especially in the Methods section. I have a couple of minor remaining points that the authors may wish to consider, which follow below.

1. Regarding nomenclature: I agree with the second reviewer that there is no reason to use two names, and I would encourage the use of the TNS name 20190520B rather than 190520B (I don't understand the objection raised that there is already a source named 20190520A – that is the reason for the alphabetical suffix?)

#Response

Thanks, all the names have been revised to FRB 20190520B.

2. Regarding the fluences of the VLA bursts: it remains surprising to me that every single VLA detection, except S5, has a higher fluence than the **brightest** FAST burst. Imagine placing the VLA bursts onto ED Figure 6: a number of the VLA bursts would have energy $>10^{39}$ erg and be off the chart, highly inconsistent with the fitted lognormal function. In 11.4 hours of VLA time, 8 bursts were seen that exceeded the maximum fluence seen by FAST in 18 hours. Obviously, the source activity level may be changing as a function of time, but both the VLA and FAST observations spanned a wide range of dates. Accordingly, given that the authors have clarified that the burst properties were measured in an identical fashion, it would seem worthy of comment somewhere in the paper that a) the localisation was seemingly rather fortuitous, given that none of the FAST bursts seen in a period of 17 hours exceed the 0.4 Jy ms⁸ sigma realfast detection limit that is quoted for the VLA, and yet the source was seen numerous times in 11.4 hours of VLA time, and b) this would seem to support variability in the activity levels (unsurprising, but worth noting).

#Response

We agree that VLA detections seem rather fortuitous, but we would also like to highlight the following that complicates this interpretation:

1. Extended data Figure 5 shows that the burst fluence distribution does not follow a power law. It seems to be clustered below the VLA L-band fluence limit of ~ 0.3 Jy ms. This makes it difficult to estimate the VLA detection rate from the FAST detection rate.
2. The burst fluence distribution is known to change in time (e.g., Li et al on FRB 121102). The VLA observations at L-band mostly fell in a gap of FAST observing, so FAST may not have seen the same activity as the VLA.
3. The FAST fluence is measured for sub-bursts, which may bias the fluence distribution. The VLA time resolution is 10 ms, so all sub-bursts are

accumulated in a single detection, leading to a seemingly higher fluence estimate.

3. Related to the above point, on line 323: the 8 sigma fluence limit of a 10 ms VLA image is quoted as 0.4 Jy ms – but this must depend on the observing band? Is this an L band number? Burst S4 and S5 fall below this limit, considerably so in the case of burst S5.

#Response

We thank the referee for pointing it out. We have added the sensitivities at all three bands to the text now.

4. Not that it appreciably impacts the results, but I think there is a minor error in the calculation of the rms of DM_MW. I get 21 units, not 17 units, for the rms, given the two uniform ranges quoted for galactic and halo contributions.

#Response

We've double checked the calculations and consistently get 17 units.

It can be calculated in two ways:

- (1) The two flat distributions when convolved (statistical independence assumed) yield a trapezoidal distribution. When the mean and rms are calculated using the usual integrals over this PDF we get 16.55 DM units that we round up to 17 DM units.
- (2) Since the two contributions are independent, their variances add. The variance of a flat distribution with width W is $W / \sqrt{12}$. Using $W_d = 0.4 \times \text{DM}_{\text{NE2001}} = 16.3$ for the disk with $\text{DM}_{\text{NE2001}} = 40.7$ DMunits and $W_h = 80 - 25 = 55$ for the halo, we get $\text{rms} = \sqrt{W_d^2 + W_h^2} / \sqrt{12} = 57.36 / \sqrt{12} = 16.55$ DM units => same answer, as expected.

5. Line 131: “Large DM_host may also imply…” -> “Large DM_host **contributions** may also imply…”

#Response

Thanks! Done

6. Everywhere the scattering time is quoted: the uncertainty should reach the same decimal place as the value. So it should be 10 +/- 2 ms, , or 9.8 +/- 2.0 ms (the latter is used in one place that I saw, but mostly it is given as 9.8 +/- 2).

#Response

The scattering time is now referred to as 10 +/- 2 ms throughout the text.

7. Line 504 and line 607: why is the Halpha luminosity presented at line 504 (without having first presented the Halpha flux, which is not shown anywhere), and then the flux suddenly pops up at line 607? It would be much more logical to introduce the observable (flux) first at line 504, since the luminosity is derived from it.

#Response

Done - the flux is now quoted before introducing the luminosity.

Referee #2 (Remarks to the Author):

General comments:

The revised manuscript is significantly improved and addresses the vast majority of my previous comments.

I recommend the paper for publication, but would again like to ask the authors to sharpen a few things.

#Response

Thanks!

The most novel observational result is that this repeating FRB has a very large DM_{host} contribution, which means that inferring redshift via DM will in some cases be highly inaccurate. Considering the current sample of FRBs with host galaxies, it still appears that such DM_{host} outliers are rare ($\sim 10\%$), and I'd still encourage the authors to give the caveat that one-off and repeating FRBs could very well have distinct DM_{host} distributions (if so, this would most likely then be due to their local environment as opposed to the ISM of the host galaxy).

The authors say "The issue of inferring an FRB redshift from its DM is independent of whether the FRB repeats or not. It is a more conservative choice to recommend caution for all FRBs rather than just for the repeating sample.". I don't fully agree with this. If one-off and repeating FRBs have different DM_{host} distributions, it is completely possible that the DM s of one-off FRBs are good proxies for redshift while those of repeaters are less reliable. That is still consistent with the data presented in Figure 3 of the paper.

#Response

In response to the comments above, we have added a caveat to the main text around line 130, stating that the \$DM_{\text{host}}\$ distributions may be different for repeating and non-repeating FRBs.

It has been revised to:

"It is conceivable that the \$DM_{\text{host}}\$ distribution may differ for repeating and non-repeating FRBs, which could make non-repeating FRB \$DM\$ s more accurate proxies for redshift."

Since the paper has been substantially reshaped, there is no longer any discussion of the Faraday rotation measure of this source. Given that the authors talk about the "dense environment" of FRB 190520B, and highlight its similarities with the original repeater FRB 121102, it is strange to remove the previous description of the polarimetric

analysis.

#Response

Thanks, we put the RM description back.

Sincerely,

Jason Hessels
University of Amsterdam & ASTRON

Below are some remaining detailed comments, mostly minor.

Summary:

- Line 52: "The dense FRB environment and the association with a compact persistent radio source may point to a distinctive origin or an earlier evolutionary stage for this FRB source.": compared to what? The majority of other FRBs? This source doesn't seem distinct in origin or necessarily younger compared to the first-known repeater, FRB 121102.

#Response

Thanks, we revised the sentence to:

"The dense FRB environment and the association with a compact persistent radio source may point to a distinctive origin or an earlier evolutionary stage for this type of source, namely FRB 20121102A and FRB 20190520B."

Main:

- Line 57: "Four bursts were detected during the initial scan."
 "Four bursts were detected during the initial XXX-s scan."
(indicate how long this scan was)

#Response

Thanks, The description has been revised to:

"Four bursts were detected during the initial 24s scan."

- Line 60: " $R(>7 \text{ sigma}) = \dots$ ": so is the 7 sigma equivalent to "a fluence lower limit of 9 mJy ms and a burst width of 1 ms": I'm confused about how the two are related and this should be more clear.

#Response

Thanks for pointing it out, we delete the sigma and change it to "R = ...".

- Line 68: "position in International"  "position in the International"

#Response

Done

- Line 69: "with position uncertainty"  "with positional uncertainty"

#Response

Done

- Line 70: "dominated by systematic effects.": point to Methods here.

#Response

Done

- Line 73: "has a flux density of 202 +/- 8 uJy at 3.0 GHz": also indicate at what epoch (or "averaged over span of epochs XXX - YYY").

#Response

Done. The sentence has been revised to "has a flux density of 202 +/- 8 uJy averaged over span of ~ 2 months from August 30 to November 16 of 2020 at 3.0 GHz".

- Line 75: "spectrum can be fit with a power-law index of": although it is fairly standard in astronomy, you should probably still define "spectral index" clearly, given the broader audience.

#Response

We have added definitions in the "Method".

- Line 76: "was obtained by"  "was obtained using the"

#Response

Done

- Line 83: "even before considering the radio counterpart association": I agree that the radio counterpart association with the FRB is clear, but it's less obvious to me how this strengthens the connection to the claimed host galaxy. I would leave this out.

#Response

Thanks, Done

- Line 84: "1.153-1.354"  "1.153\$-\$1.354"

#Response

Done

- Line 84: "shows the most likely stellar continuum emission": do you mean "most likely shows the stellar continuum emission of the host galaxy"?

#Response

Thank you for the suggestion. We have revised the description here to "most likely shows the stellar continuum emission of the host galaxy".

- Line 92: "indicates the"  "indicates that the"

#Response

Done

- Line 104: "a flat halo distribution from": would be more clear to say "uniform"?

#Response

Thanks, Done

- Line 126: "demonstrates that the distribution of DM_host values has a long tail"  "demonstrates that the distribution of DM_host values for the FRB population has a long tail"

#Response

Thanks, Done

- Line 126: In this paragraph it should be noted that the DM_host distribution might be different for repeating and non-repeating FRBs. In fact, despite the currently small sample of FRBs with hosts, this might already be demonstrable.

#Response

We have added a statement to this effect in this paragraph.

- Line 133: It would be good to quote the size constraint on the PRS, which I think is currently only mentioned in the Methods, and to compare this to the constraint on FRB 121102's PRS (from Marcote et al. 2017).

#Response

The PRS size limits for both FRBs are mentioned at the end of the paragraph.

- Line 148: "rotation measure variations of over 15% per year": it bears mentioning that FRB 121102 has an extremely large RM, and that RM variations are seen on both short (burst to burst, day to day, week to week) and long (year) timescales (see also Hilmarsson et al. 2021). There should also be some mention of what we know about the RM of FRB 190520B, since that is highly relevant to understanding whether it is also in an extreme magneto-ionic local environment.

#Response

We have modified the sentence to "FRB 20121102A also demonstrates a sporadically large burst rate (e.g. a peak burst rate of 122 hr⁻¹), a substantial rotation measure (RM) that varies over both short (burst to burst) and long (year) timescales and DMhost as large as ~300 pc cm⁻³". We have also cited the Hilmarsson paper here. Regarding RM of FRB 20190520B, we added a sentence to the first paragraph describing the burst properties observed at FAST: "We detected no linear polarization for FRB 20190520B, which could be due to Faraday depolarization with an $RM > 2 \times 10^5$ rad m⁻², or a depolarization process taking place within the source itself (see Methods)".

- Line 157: "This hypothesis may however be modified by the ambiguity

in the distribution of ionized gas responsible for the large DM_host from the host galaxy of FRB 190520B.”: this sentence is hard to interpret. Do you mean that the DM_host isn’t necessarily (all) local to the source and hence doesn’t directly relate to the age/direct environment of the FRB? This could be worded much more clearly. If the DM_host were largely local to the FRB, one might also have expected to see a systematic DM evolution, right? (e. g. Piro paper)

#Response

This sentence has been re-worded for clarity.

Figure 1:

For the plot: We change the colormap (**only**) of the plot to make it better look.

- “labeled in red patches”  “labeled using red patches”

#Response

Done

- Indicate what DM (or multiple DMs?) were used to dedisperse the bursts as shown here.

#Response

Thanks, we referred them in the supplementary information table 1.

Figure 2:

- The 0.1 arcsec radius localisation circle is so small that it might be better to show a 3-sigma circle instead. Or maybe I just need glasses.

#Response

Thanks for the concern. After the discussion of our co-authors, we decide to stay the same.

Table 1:

- “Isotropic energy”  “Isotropic equivalent energy”

#Response

Done

- “Flux at 3.0 GHz”  “Flux density at 3.0 GHz”

#Response

Done

- For the PRS, add a table note to indicate the epoch of the flux density measurement (or range of epochs that are averaged over).

#Response

We have added a table note to indicate the range of epochs that are averaged over.

- Line 188: “NAN, R.”: is this name meant to be all caps?

#Response

Thanks for pointing this out, the bibitem is taken from ADS, and the mistake has been corrected, Thanks!

Methods:

- Line 288: "sad trombone drift"  "time-frequency drift"

#Response

Done

- Line 293: "by a boxcar search"  "using a boxcar search"

#Response

Done

- Line 295: "follow-up burst search was taken with a narrower"  "follow-up burst search was done with a narrower"

#Response

Done

- Line 296: "100-2000"  "100\$-\$2000"

#Response

Done

- Line 302: "(see Section 5)": I'm not sure you can use section numbering in the Methods.

#Response

Thanks, we removed the number serials.

- Line 321: "for transient search"  "to search for transients"

#Response

Done

- Line 362: "Most of the bursts are frequency modulated": I don't think you really mean "frequency modulated" (which has a specific meaning in engineering, e.g.), but rather "spectrally confined".

#Response

Thanks, the description has been revised to:

"Most of the bursts are spectrally confined"

- Line 373: "R.A. error, Decl. error)": use the same abbreviations as earlier in the paper (do this throughout).

#Response

Thanks, all the items have been uniformed to "R.A. & Decl. "

- Line 381: "of FRB position"  "of the FRB position"

#Response

Done

- Line 416: "as 1.4'' x 0.89'', and 0.51'' x 0.14'', and 0.36'' x 0.1''"
 "as 1.4'' x 0.89'', 0.51'' x 0.14'', and 0.36'' x 0.1''"

#Response

Done

- Line 442: "The error on this position is estimated to be 0.10'' and

0.05''.": this should be written in a way to make it clear that you mean that these represent the uncertainties on the RA and Dec coordinates, respectively.

#Response

Done

- Line 449: "yielding an average spectral index for the PRS of": define spectral index.

#Response

Definition added.

- Line 516: "a flat SED in": define "SED" (or just use "spectral energy density", since this is the only occurrence). Also define "nu" and "L_nu".

#Response

Thanks, We use "spectral energy distribution" instead of "SED". The defined "nu" and "L_nu" as :

"where nu and L_nu are frequency and luminosity, respectively."

- Line 528: "obtained with Markov Chain"  "obtained with a Markov Chain"

#Response

Done

- Line 553: "Assuming bursts have flat radio spectra": avoid jargon. Would be more clear to say "spectral index of about 0" (you will have also defined spectral index earlier in the text)

#Response

Done

- Line 558: "the luminosity distance": here I'd add something like "using standard cosmological parameters" and give a reference.

#Response

Done.

- Line 569: "as the mean of a flat distribution": more clear to say "uniform"? (here and in the next sentence)

#Response

Done

- Line 647: "such as the time-frequency drift of intensity islands (often referred to as the "sad trombone" effect).": give reference.

#Response

Done.

- Line 757: "for the technical support."  "for their technical support."

#Response

Done

- Line 772: "helped on"  "helped with"

#Response

Done

- Line 776: "on the present results"  "on the presented results"

#Response

Done

Extended Data:

ED Table 2:

- "offsets in RA. and Dec."  "offsets in RA and Dec." (remove extraneous period)

#Response.

Done

ED Table 3:

- For row "S5", quote RA 16h02m04.250s to get the same alignment

#Response.

Done

ED Table 4:

- "Flux" should be "Flux density"

#Response.

Done

- For row "S3", quote Frequency 2.50-2.76 to get the same alignment

#Response.

Done

- "(TDB)"  "(in the TDB scale)"

#Response.

Done

- "DM that maximizes"  "The DM that maximizes"

#Response.

Done

- Add a single-sentence note that the apparent DMs may reflect time-frequency structure, as opposed to bona fide DM variations.

#Response.

Done, we add

"the apparent DMs may reflect time-frequency structure, as opposed to bona fide DM variations"

in the end.

Supplementary Information:

Table 1:

- "of barycentric dynamical time"  "in barycentric dynamical time (TDB)"

#Response.

Done

- "the international celestial reference System (ICRS)"  "the International Celestial Reference System (ICRS)"

#Response.

Done

- "Epoch-to-epoch has been separated by single line"  "Epochs are separated by single horizontal lines"

#Response.

Done

- "and its DM variations may solely"  "and the apparent DM variations may solely"

#Response.

Done